# Proper Hölder-Kullback Dirichlet Diffusion: A Framework for High Dimensional Generative Modeling

**Wanpeng Zhang[1],    Yuhao Fang[2],    Xihang Qiu[1],    Jiarong Cheng[1],    Jialong Hong[1],    Bin Zhai[1],    Qing Zhou[2,3],    Yao Lu[2],    Ye Zhang[1,2]\*,    Chun Li[2]\***

[1] Beijing Institute of Technology, Beijing, 100081, China
[2] Shenzhen MSU-BIT University, Shenzhen, 518172, China
[3] Lomonosov Moscow State University, Moscow, 119991, Russia
lichun2020@smbu.edu.cn
https://github.com/TudoWay/PHKDif

## Abstract

Diffusion-based generative models have long depended on Gaussian priors, with little exploration of alternative distributions. We introduce a Proper Hölder-Kullback Dirichlet framework that uses time-varying multiplicative transformations to define both forward and reverse diffusion processes. Moving beyond conventional reweighted evidence lower bounds (ELBO) or Kullback–Leibler upper bounds (KLUB), we propose two novel divergence measures: the Proper Hölder Divergence (PHD) and the Proper Hölder–Kullback (PHK) divergence, the latter designed to restore symmetry missing in existing formulations. When optimizing our Dirichlet diffusion model with PHK, we achieve a Fréchet Inception Distance (FID) of 2.78 on unconditional CIFAR-10. Comprehensive experiments on natural-image datasets validate the generative strengths of model and confirm PHK's effectiveness in model training. These contributions expand the diffusion-model family with principled non-Gaussian processes and effective optimization tools, offering new avenues for versatile, high-fidelity generative modeling.

## 1 Introduction

Diffusion–based generative models are revolutionizing deep learning by simulating gradual noise addition and removal processes to transform simple priors into complex data distributions. From high-fidelity image synthesis and editing [1–5] to video generation [6, 7] and protein design [8–10]. As such, there is great interest in applying diffusion models and improving them further in terms of distribution quality, training cost, and image performance.

Recent efforts have investigated non-Gaussian diffusion processes, extending to categorical distribution [11–14], Poisson distribution [15], and Beta distribution [16]. However, Beta diffusion incurs limitation that it applies one-dimensional Beta tr ansitions independently along each coordinate, preventing it from capturing interdependencies among components in a high-dimensional simplex. Traditional divergence measures such as the Kullback–Leibler (KL) divergence suffer from asymmetry and computational complexity.

To address these limitations, in this work, we proposed Proper Hölder–Kullback Dirichlet Diffusion, which is a fundamentally novel framework and can more faithfully captures the simplex-structured target space.

---

*Corresponding author

39th Conference on Neural Information Processing Systems (NeurIPS 2025).

Its main contributions are summarized as follows:

1. $K$**-dimensional Dirichlet diffusion on simplex is established. Beta diffusion is generalized to a $K$-dimensional Dirichlet process that preserves the simplex structure at every noising step.**

2. **Strictly proper Hölder training objectives are proposed. Closed-form losses based on the Proper Hölder Divergence (PHD) and Proper Hölder–Kullback (PHK) are obtained, and analytic gradients under the Dirichlet exponential-family log-normalizer are guaranteed.**

3. **Variance-reduced gradient estimates are proven. It is shown that PHD yields lower-variance gradients compared to pseudo-divergence objectives, improving training stability and efficiency.**

State-of-the-art non-Gaussian performance is demonstrated on unconditional CIFAR-10: an FID of 2.78 is achieved, outperforming all prior non-Gaussian diffusion methods and most Gaussian baselines, while PHK concomitantly reduces approximately a 9 % training cost compared with KLUB. Additional uncurated samples trained on CIFAR-10, AFHQ, FFHQ and CVUSA dataset exhibit both high fidelity and diversity. By breaking free from Gaussian noise and KL-centric optimization, Proper Hölder–Kullback Dirichlet Diffusion opens up a richer design space for generative modeling, and offering both rigorous theoretical guarantees and clear empirical gains.

## 2   Background

Diffusion-based deep generative models have shown remarkable potential for development. Both Langevin dynamics and variational inference interpretations of diffusion models fundamentally depend on the properties of Gaussian noise employed in training and sampling pipelines [17]. Initially inspired by diffusion models utilizing Gaussian noise, various Gaussian diffusion models have emerged [1, 18, 2, 19, 3]. These models use Gaussian Markov chains to progressively diffuse images into Gaussian noise during training. Conversely, the learned reverse diffusion process, defined by reversing the Gaussian Markov chain, iteratively refines the noisy inputs to generate clean, realistic images.

Gaussian diffusion models can be further categorized from perspectives such as score matching [20, 21, 18, 19] and stochastic differential equations [22]. These models have demonstrated notable successes across diverse tasks, including image generation and editing [4, 5, 23–26], 2D-to-3D conversion [27, 28], calibrated Bayesian inference [29], audio synthesis [30–32], conditional video generation [6, 7], reinforcement learning [33–35], few-shot learning [36], protein design [8–10], and cross-modal visual generation and editing [37, 38].

Typically, diffusion-based generative models are built following a standard procedure [1–3, 39], involving three core steps: (1) defining a forward diffusion process that progressively degrades data by introducing noise and reducing the signal-to-noise ratio (SNR) from time 0 to 1; (2) specifying a reverse diffusion process that denoises the data in reverse time, from 1 to 0; and (3) discretizing the time interval [0,1] into finite steps, with the forward and reverse processes serving as a fixed inference network and a learnable generator, respectively. Variational autoencoder inference [40, 41] is subsequently applied to optimize generator parameters by minimizing a weighted negative evidence lower bound (ELBO), comprising Kullback–Leibler (KL) divergence losses at each discretized reverse step.

In this work, we explore various effective distributions and divergences to investigate the multiple possibilities of the model. Our focus has progressively shifted towards diffusion models based on the Dirichlet distribution and Hölder divergence. Unlike traditional Gaussian diffusion, our approach is based on models employing non-traditional distributions and the deliberate incorporation of specifically tailored loss functions. Specifically, the adoption of Dirichlet distribution and Hölder divergence, which eschews the reliance on Gaussian distributions during both training and testing phases, enables us to explore alternative distributions and divergences, thereby unlocking new pathways for generative modeling. This innovative approach has the potential to engender models with distinct theoretical properties and to stimulate the emergence of innovative research trajectories.

# 3 Method

## 3.1 From Beta distribution to Dirichlet distribution

The Beta distribution is a continuous probability distribution defined on the interval $[0, 1]$, commonly used to model probabilities or proportions. Its probability density function is given by [42]:

$$f(x; \alpha, \beta) = \frac{x^{\alpha-1}(1-x)^{\beta-1}}{\int_0^1 u^{\alpha-1}(1-u)^{\beta-1}\mathrm{d}u} = \frac{x^{\alpha-1}(1-x)^{\beta-1}}{\mathrm{B}(\alpha, \beta)}, \quad x \in [0, 1], \tag{1}$$

where $\mathrm{B}(\alpha, \beta)$ is the Beta function, and $\alpha$ and $\beta$ are shape parameters. Beta function can be expressed in terms of the Gamma function: $\mathrm{B}(\alpha, \beta) = \frac{\Gamma(\alpha)\Gamma(\beta)}{\Gamma(\alpha+\beta)}$.

For the Beta distribution, the mean and variance are:

$$\mathbb{E}[X] = \frac{\alpha}{\alpha + \beta}, \quad \mathrm{Var}(X) = \frac{\alpha\beta}{(\alpha + \beta)^2(\alpha + \beta + 1)}. \tag{2}$$

In Beta diffusion [16], for scalar data $x_0 \in [0, 1]$, the forward diffusion process is defined as: $z_t|x_0 \sim \mathrm{Beta}(\eta\alpha_t x_0, \eta(1 - \alpha_t x_0))$, where $\alpha_t$ is a time-dependent parameter, and $\eta$ is a scaling parameter. When $t = T$, $z_T$ approaches uniform noise. During training, the parameters of the reverse process are obtained by minimizing the Kullback-Leibler (KL) divergence to obtain a parameterized model $f_{\boldsymbol{\theta}}(z_t, t)$:

$$\theta^* = \arg\min_\theta \mathbb{E}_{z_t, z_0}\left[\mathrm{KL}(q(z_s|z_t, x_0)\|q(z_s|z_t, \hat{x}_0 = f_\theta(z_t, t)))\right]. \tag{3}$$

The Dirichlet distribution is a high-dimensional generalization of the Beta distribution, used to model distributions on the simplex. Its probability density function is given by [43]:

$$f(\boldsymbol{x}; \boldsymbol{\alpha}) = \frac{\Gamma\left(\sum_{i=1}^{K} \alpha_i\right)}{\prod_{i=1}^{K} \Gamma(\alpha_i)} \prod_{i=1}^{K} x_i^{\alpha_i - 1}, \quad \boldsymbol{x} \in S_K, \tag{4}$$

where $S_K$ denotes the $K$-dimensional simplex, and $\alpha_i$ is the shape parameter for the $i$-th dimension.

For the Dirichlet distribution, the mean and variance are [43]:

$$\mathbb{E}[X_i] = \frac{\alpha_i}{\sum_{j=1}^{K} \alpha_j}, \quad \mathrm{Var}(X_i) = \frac{\alpha_i\left(\sum_{j=1}^{K} \alpha_j - \alpha_i\right)}{\left(\sum_{j=1}^{K} \alpha_j\right)^2 \left(\sum_{j=1}^{K} \alpha_j + 1\right)}. \tag{5}$$

It is worth noting that the Dirichlet function can also be expressed in terms of the Gamma function. Compared to the Beta function, it extends to K dimensions in the same form:

$$\mathrm{Dir}(\boldsymbol{\alpha}) = \frac{\prod_{k=1}^{K} \Gamma(\alpha_k)}{\Gamma\left(\sum_{k=1}^{K} \alpha_k\right)}. \tag{6}$$

Based on the above formulas and derivations, the one-dimensional diffusion in Beta diffusion can be extended to $K$-dimensional diffusion to reasonably implement the diffusion process on high-dimensional simplices:

$$z_t|\boldsymbol{x}_0 \sim \mathrm{Dir}(\eta\alpha_{t,1}x_{0,1}, \eta\alpha_{t,2}x_{0,2}, \ldots, \eta\alpha_{t,K}x_{0,K}), \tag{7}$$

where $\alpha_{t,i}$ are time-dependent parameter functions. By scheduling $\alpha_{t,i}$, $z_0 = \boldsymbol{x}_0$ can be nearly noise-free, while $z_T$ approaches a uniform distribution. The expected value is:

$$\mathbb{E}[z_{t,i}|\boldsymbol{x}_0] = \frac{\alpha_{t,i}x_{0,i}}{\sum_j \alpha_{t,j}x_{0,j}}. \tag{8}$$

Dirichlet distribution enables direct handling of high-dimensional data without decomposing the high-dimensional problem into multiple one-dimensional problems. It offers significant advantages in modeling data distributions on high-dimensional simplices and naturally extends from low-dimensional to high-dimensional diffusion processes. This extension makes the Dirichlet distribution more flexible and efficient in dealing with high-dimensional data.

## 3.2 Training via Proper Hölder-Kullback Divergence

For the diffusion process of the Dirichlet distribution on high-dimensional simplices, to ensure that the objective function has a closed-form solution under the Dirichlet distribution, this study considers introducing an objective function based on the Hölder divergence.

Two objective functions that meet the criteria are available for further experimental validation: the Hölder Statistical Pseudo-Divergence (HPD) and the Proper Hölder Divergence (PHD) [44].

The objective function based on HPD is expressed as:

$$D_\alpha^H(p:q) = \frac{1}{\alpha}F(\alpha\theta_p) + \frac{1}{\beta}F(\beta\theta_q) - F(\theta_p + \theta_q), \quad \beta = \frac{\alpha}{\alpha - 1}. \tag{9}$$

For the Dirichlet distribution, $F(\theta)$ has a closed-form expression, where $\theta_i = \alpha_i - 1$. If

$$p = q(z_s|z_t, x_0) = \mathrm{Dir}(\boldsymbol{a}_p), \quad q = q(z_s|z_t, \hat{x}_0) = \mathrm{Dir}(\boldsymbol{a}_q), \tag{10}$$

let $\theta_{p,i} = a_{p,i} - 1$, $\theta_{q,i} = a_{q,i} - 1$, then $D_\alpha^H(p:q)$ can be computed analytically. The training objective is:

$$\mathcal{L}_\theta^{\mathrm{HPD}} = \mathbb{E}_{z_t, \lambda_0}[D_\alpha^H(q(z_s|z_t, x_0)\|q(z_s|z_t, \hat{x}_0 = f_\theta(z_t, t)))]. \tag{11}$$

The objective function based on PHD is expressed as:

$$D_{\alpha,\gamma}^H(p:q) = \frac{1}{\alpha}F(\gamma\theta_p) + \frac{1}{\beta}F(\gamma\theta_q) - F(\frac{\gamma}{\alpha}\theta_p + \frac{\gamma}{\beta}\theta_q), \quad \beta = \frac{\alpha}{\alpha - 1}. \tag{12}$$

Similarly, a closed-form solution can be obtained. The training objective is:

$$\mathcal{L}_\theta^{\mathrm{PHD}} = \mathbb{E}_{z_t, z_0}[D_{\alpha,\gamma}^H(q(z_s|z_t, x_0)\|q(z_s|z_t, \hat{x}_0 = f_\theta(z_t, t)))]. \tag{13}$$

Theory and experiments indicate that PHD yields superior performance, an advantage associated with the strict propriety of PHD. A detailed explanation is provided in Appendix A.

Furthermore, recognizing the unique advantages of PHD, combined with the mature foundation of KL theory in the diffusion model domain, KLUB [16], this study introduces the Proper Hölder-Kullback Divergence (PHK), which also has a closed-form solution for the Dirichlet distribution. Given $\delta, \epsilon \in [0, 1]$ as weight coefficients, the objective function based on PHK is expressed as:

$$\mathcal{L}_\theta^{\mathrm{PHK}} = \delta\mathcal{L}_\theta^{\mathrm{PHD}} + (1 - \delta + \epsilon)\mathcal{L}_\theta^{\mathrm{KLUB}}. \tag{14}$$

We summarize the training and sampling algorithms in Algorithms 1 and 2, respectively.

---

**Algorithm 1** Training

---

**Require:** Dataset: $D = \left\{\{\mathrm{X}_n^m\}_{m=1}^M, y_n\right\}_{n=1}^N$, Mini-batch size $B$, concentration parameter $\eta = 10000$, data shifting parameter $S_{\mathrm{hift}} = 0.6$, data scaling parameter $S_{\mathrm{cale}} = 0.39$, generator $f_\theta$, time reversal coefficient $\pi = 0.95$, and a sigmoid schedule defined by $\alpha_t = 1/(1 + e^{-c_0 - (c_1 - c_0)t})$ given $t \in [0, 1]$, where $c_0 = 10$ and $c_1 = 13$

1: **repeat**
2:     Draw a mini-batch $X_0 = \{x_0^{(i)}\}_{i=1}^B$ from $\mathcal{D}$
3:     **for** $i = 1$ to $B$ **do**                                       ▷ can be run in parallel
4:         $t_i \sim \mathrm{Unif}(1e - 5, 1)$
5:         $s_i = \pi t_i$
6:         Compute $\alpha_{s_i}$ and $\alpha_{t_i}$
7:         $x_0^{(i)} = x_0^{(i)} * S_{\mathrm{cale}} + S_{\mathrm{hift}}$
8:         $z_{t_i} \sim \mathrm{Dir}(\eta\alpha_{t_i}x_0^{(i)}, \ldots, \eta\frac{1}{K-1}(1 - \alpha_{t_i}x_0^{(i)}))$
9:         $\hat{x}_0^{(i)} = f_\theta(z_{t_i}, t_i) * S_{\mathrm{cale}} + S_{\mathrm{hift}}$
10:        compute the loss $\mathcal{L}_i$
11:     **end for**
12:     Perform SGD with $\frac{1}{B}\nabla_\theta \sum_{i=1}^B \mathcal{L}_i$
13: **until** converge

---

**Algorithm 2** Sampling

---

**Require:** Number of function evaluations (NFE) $J = 200$, generator $f_\theta$, and timesteps $\{t_j\}_{j=0}^J$:
$\quad t_j = 1 - (1 - 1e - 5) * (J - j)/(J - 1)$ for $j = 1, \ldots, J$ and $t_0 = 0$
  1: **if** NFE $> 350$ **then**
  2:      $\alpha_{t_j} = 1/(1 + e^{-c_0 - (c_1 - c_0)t_j})$
  3: **else**
  4:      $\alpha_{t_j} = (1/(1 + e^{-c_1}))^{t_j}$
  5: **end if**
  6: Initialize $\hat{x}_0 = \mathbb{E}[x_0] * S_{\text{cale}} + S_{\text{hift}}$
  7: $z_{t_J} \sim \text{Dir}(\eta \alpha_{t_J} \hat{x}_0, \ldots, \eta \frac{1}{K-1}(1 - \alpha_{t_J} \hat{x}_0))$
  8: **for** $j = J$ to $1$ **do**
  9:      $\hat{x}_0 = f_\theta(z_{t_j}, \alpha_{t_j}) * S_{\text{cale}} + S_{\text{hift}}$
10:      $p_{(t_{j-1} \leftarrow t_j)} \sim \text{Dir}(\eta(\alpha_{t_{j-1}} - \alpha_{t_j})\hat{x}_0, \ldots, \eta \frac{1}{K-1}(1 - \alpha_{t_{j-1}} \hat{x}_0))$
11:      $z_{t_{j-1}} = z_{t_j} + (1 - z_{t_j})p_{(t_{j-1} \leftarrow t_j)}$
12: **end for**
13: **return** $(\hat{x}_0 - S_{\text{hift}})/S_{\text{cale}}$ or $(z_{t_0}/\alpha_{t_0} - S_{\text{hift}})/S_{\text{cale}}$

---

### 3.3 Forward and Reverse Diffusion

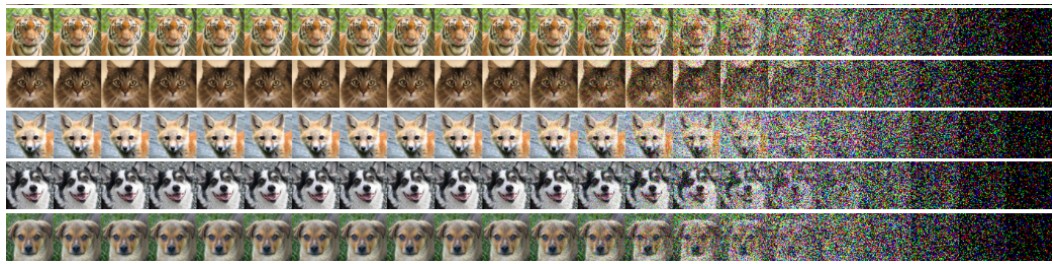

Figure 1: Demonstration of the forward diffusion process on five samples. the first column displays the original iamge, and the subsequent 21 columns illustrate progressively noising and masking images at time steps $t = 0, 0.05, \ldots, 1$.

This study conducts a analysis of the forward and reverse diffusion processes by incorporating process images. First, the Dirichlet forward diffusion process is visualized by displaying the real image $x$ and its noise-corrupted version during the forward diffusion process. Specifically, the images with noise and masking at times $t = 0, 0.05, 0.1, \ldots, 1$ are represented as:

$$\hat{z}_t = \max\left(\min\left(\frac{1}{S_{\text{cale}}}\left(\frac{z_t}{\alpha_t} - S_{\text{hift}}\right), 1\right), 0\right), z_t \sim \text{Dir}(\eta \alpha_t x_0, \ldots, \eta \frac{1}{K-1}(1 - \alpha_t x_0)). \quad (15)$$

As can be clearly seen from Figure 1, during the forward diffusion process, the image becomes increasingly noisy and sparse with the time, and is ultimately almost estroyed. It is evident that the forward process of Dirichlet diffusion involves simultaneous noising and masking of pixels. This is distinct from traditional Gaussian diffusion, as the forward diffusion process in Gaussian diffusion gradually applies additive random noise and eventually concludes with Gaussian random noise.

Similarly, the reverse diffusion process of traditional Gaussian diffusion is considered a denoising process, whereas the reverse diffusion process of Dirichlet diffusion involves simultaneous denoising and demasking of the data. The images representing denoising and demasking are expressed as:

$$\hat{z}_{t-1} = \max\left(\min\left(\frac{1}{S_{\text{cale}}}\left(\frac{z_{t-1}}{\alpha_{t-1}} - S_{\text{hift}}\right), 1\right), 0\right), \quad (16)$$

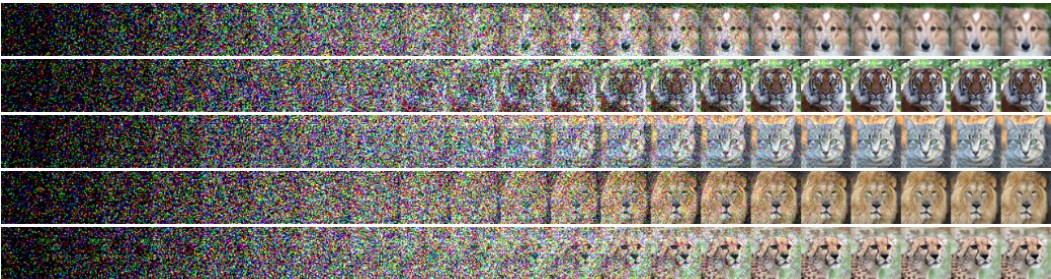

Figure 2: Demonstration of the reverse diffusion process on five examples. The first column displays the initial noisy, and the subsequent 21 columns illustrate the progressively denoising and demasking at time steps $t = 1, 0.9, \ldots, 0$.

where $z_{t-1} = z_t + (1 - z_t)p_{(t_j \to t_{j-1})}$ is iteratively computed according to Algorithm 2. High-resolution images require more computational resources. Final generated image is represented as:

$$\hat{x}_0 = f_\theta(z_{t_j}, t_j) \approx \mathbb{E}[x_0 \mid z_{t_j}], \tag{17}$$

The generated image data will more accurately approximate the real iamge data when $\theta$ increasingly approaches its theoretical optimal value $\theta^*$, such that $f_{\theta^*}(z_{t_j}, t_j) = \mathbb{E}[x_0 \mid z_{t_j}]$.

As shown in Figure 2, starting from random noise

$$z_{t_j} \sim \mathrm{Dir}(\eta \alpha_{t,1} x_{0,1}, \ldots, \eta \alpha_{t,K} x_{0,K}), \quad \hat{x}_0 = \mathbb{E}[x_0], \tag{18}$$

most of the pixel values will be entirely black. Dirichlet diffusion gradually denoises and restores the image to a clean state through multiplicative transformations, as demonstrated in Algorithm 2.

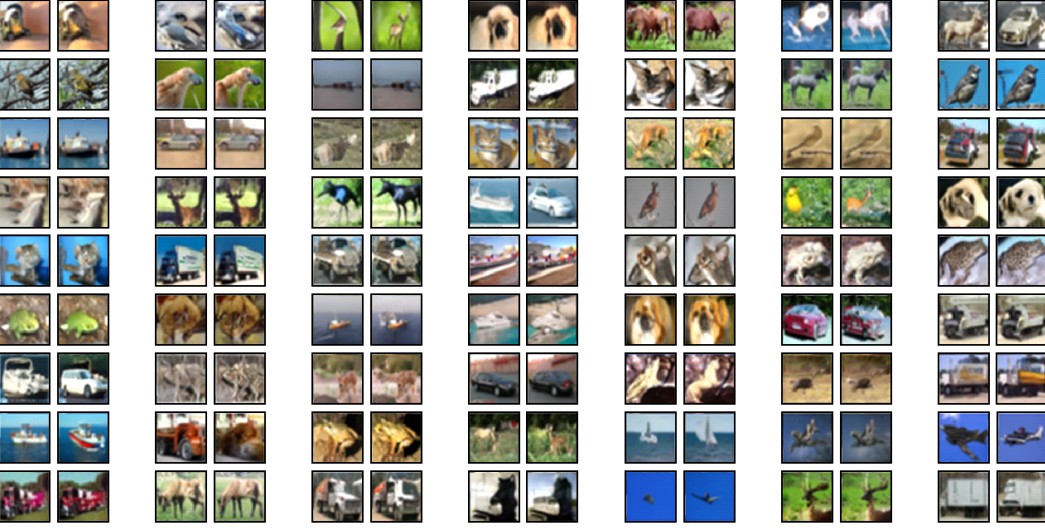

Figure 3: Comparison of generated samples from the Beta Diffusion model and Dirichlet Diffusion model trained on unconditional CIFAR-10 image dataset. (L) Beta Diffusion and (R) Dirichlet Diffusion. This figure presents side-by-side examples of images synthesized on the CIFAR-10 image dataset by two diffusion frameworks.

## 4 Experiments

### 4.1 Comparison Study

This study offers a comprehensive numerical comparison with diffusion models based on traditional Gaussian distributions and those based on non-Gaussian or quasi-Gaussian distributions that have

been publicly released in recent years at NeurIPS, ICLR, ICML, etc. Table 1 encompasses a wide range of diffusion models. The results demonstrate that the new diffusion model proposed in this study achieved a Fréchet Inception Distance (FID) [45] score of 2.78 on the unconditional CIFAR-10 dataset [46], surpassing all non-Gaussian-based diffusion models, including the cold diffusion model based on deterministic diffusion [17], the Inverse Heat Dispersion based on deterministic diffusion [47], the D3PM diffusion model based on categorical distribution [12], and the JUMP diffusion model based on Poisson distribution [15]. Compared with the family of Gaussian diffusion models, the new diffusion model in this study outperformed the majority of Gaussian-like models, including NCSNv2 [19], DDPM [2], DiffuEBM [48], VDM [3], Improved DDPM [49], TDPM+ [50], Soft Diffusion [51], Blurring Diffusion [52], etc. It thus serves as a highly competitive alternative when employing diffusion models. As a newly proposed diffusion model, constrained by limitations in computational resources and training time, this study has not yet been able to conduct a detailed investigation and global search of all model parameters, nor has it introduced targeted enhancement modules. As a novel perspective on diffusion models, this study holds broad potential for future research.More hyperparameter settings and training details of the experiments are provided in Appendix F and G.

Table 1: FID scores for various generative models trained on the CIFAR-10 dataset. Lower FID indicates better image fidelity and diversity.

| Distribution | Model | Year | FID ($\downarrow$) |
|---|---|---|---|
| Gaussian | DDPM [2] | NeurIPS'20 | 3.17 |
| | NCSNv2 [19] | NeurIPS'20 | 10.87 |
| | DiffuEBM [48] | ICML'21 | 9.58 |
| | VDM [3] | NeurIPS'21 | 4.00 |
| | Improved DDPM [49] | ICML'21 | 2.94 |
| | TDPM+ [50] | ICLR'23 | 2.83 |
| | Soft Diffusion [51] | TMLR'23 | 3.86 |
| | Blurring Diffusion [52] | ICLR'23 | 3.17 |
| | Blackout Diffusion [53] | ICML'23 | 4.58 |
| | GET [54] | NeurIPS'24 | 5.49 |
| | Diff-Instruct [55] | NeurIPS'24 | 4.53 |
| | UniPC with optimized step [56] | CVPR'24 | 3.13 |
| | RDUOT [57] | ECCV'24 | 2.95 |
| Deterministic | Cold Diffusion [17] | NeurIPS'23 | 80.08 (deblurring) 8.92 (inpainting) |
| | Inverse Heat Dispersion [47] | ICLR'23 | 18.96 |
| Categorical | D3PM [12] | NeurIPS'21 | 7.34 |
| Beta | Beta Diffusion [16] | NeurIPS'23 | 3.06 |
| Poisson | JUMP [15] | ICML'23 | 4.80 |
| **Dirichlet** | **Ours** | - | **2.78** |

## 4.2 Ablation Study

In this study, we meticulously designed and conducted a series of ablation experiments to systematically validate the effectiveness of the proposed PHD and PHK based on Hölder divergence. Through these experiments, we investigated the performance and potential advantages of these two methods.

As shown in Table 2, the experimental results clearly reveal the superiority of PHD and PHK over traditional methods. Specifically, compared with the widely used $-$ELBO, PHD achieves better generation performance and demonstrates unique capabilities in image processing. Moreover, PHK, as a hybrid optimization method, can effectively integrate the advantages of different divergences. It processes image data more comprehensively than other divergences, thereby achieving optimal performance. This result further validates the effectiveness and superiority of our proposed methods, indicating that PHD and PHK hold broad research prospects and application potential in the field of image generation.

Table 2: Ablation study on CIFAR-10 varying the number of function evaluations (NFE) and mini-batch size $B$, identifying the optimal NFE–$B$ configuration that maximizes performance per computation and guides efficient resource allocation.

| Loss
$B$ | −ELBO
512 | −ELBO
288 | KLUB
512 | KLUB
288 | PHD
512 | PHD
288 | PHK
512 | PHK
288 |
|---|---|---|---|---|---|---|---|---|
| 20 | 15.81 | 16.02 | 16.62 | 16.35 | 16.84 | 16.95 | 15.92 | 16.53 |
| 50 | 6.53 | 6.65 | 6.16 | 6.20 | 6.48 | 6.93 | 5.87 | 6.15 |
| 200 | 4.49 | 4.68 | 3.46 | 3.55 | 3.85 | 4.43 | 3.20 | 3.29 |
| 500 | 4.33 | 4.42 | 3.39 | 3.46 | 3.72 | 4.23 | 2.92 | 2.99 |
| 1000 | 4.28 | 4.42 | 3.32 | 3.40 | 3.52 | 4.19 | 2.84 | 2.91 |
| 2000 | 4.31 | 4.39 | 3.25 | 3.21 | 3.51 | 4.21 | 2.82 | 2.81 |

We also include Figure 3 to visually compare generated images under Beta Diffusion and Dirichlet Diffusion on CIFAR-10, Figure 4 to visually compare generated images under Dirichlet Diffusion on FFHQ [58] and AFHQ [59]. Owing to the considerable time required for each training iteration and the computationally demanding nature of FID assessment, we have not yet explored the optimization of these hyperparameter combinations, given the constraints of our current computational resources. Consequently, although the results presented in this paper illustrate that Dirichlet diffusion can achieve competitive performance in image generation, they do not fully capture the potential capabilities of Dirichlet diffusion. Further enhancements to these results may be attainable through optimized hyperparameter configurations or network architectures specifically designed for Dirichlet diffusion. We defer these investigations to our future work.

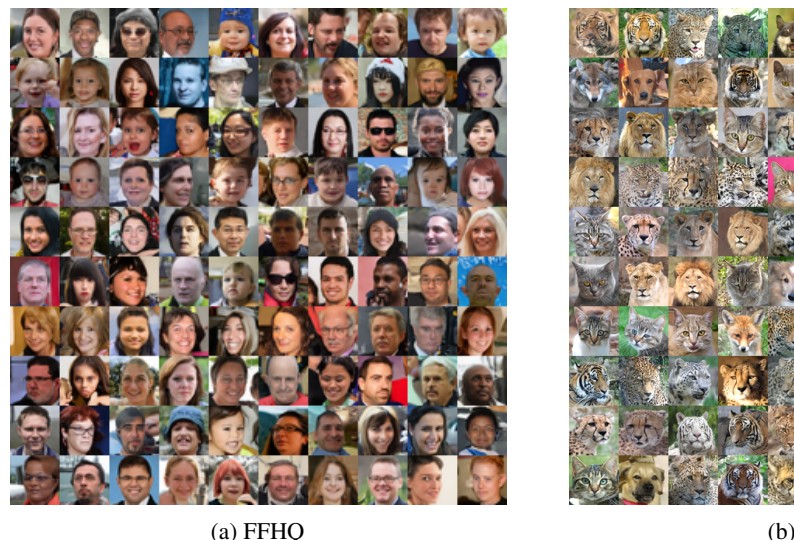
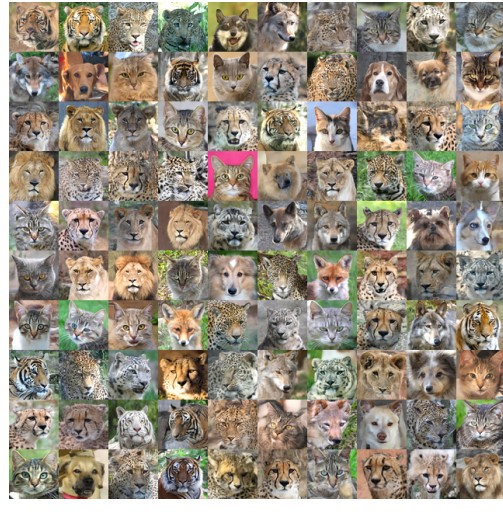

(a) FFHQ          (b) AFHQ

Figure 4: Generated images from the Dirichlet Diffusion model on the FFHQ and AFHQ datasets. (a) FFHQ samples.(b) AFHQ samples.These results demonstrate that Dirichlet Diffusion produces high-fidelity, diverse images, capturing both fine-grained human facial details and the varied textures and structures of animals.

## 5   Related Work and Future Directions

Various diffusion processes, including Gaussian diffusion, Beta diffusion, Poisson diffusion, and Dirichlet diffusion, employ specific distributions in both forward and reverse sampling. Gaussian-based diffusion models initiate their reverse process from the standard normal distribution $\mathcal{N}(0, 1)$, whereas Poisson, Beta, and Dirichlet diffusion all start from 0. The reverse sampling of Dirichlet diffusion is a monotonically non-decreasing process, similar to Poisson diffusion. However, while Poisson diffusion involves discrete jumps in count values, Dirichlet diffusion employs continuous jumps in probability values, consistent with Beta diffusion.

In recent years, several works have actively explored diffusion models based on non-traditional Gaussian distributions. Bansal et al. [17] proposed cold diffusion, which constructs models based on arbitrary image transformations rather than Gaussian noise. Rissanen et al. [47] introduced Inverse Heat Dispersion, which generates images by stochastically reversing the heat equation. Austin et al. [12] proposed the D3PM diffusion model based on the Categorical distribution. Chen et al. [15] proposed the JUMP diffusion model based on the Poisson distribution. In comparison, Dirichlet diffusion, which innovates in terms of noise distribution and training loss, has achieved a new state-of-the-art FID for non-Gaussian models on the unconditional CIFAR-10 dataset.

There have also been several works in the quasi-Gaussian diffusion domain in recent years. Hoogeboom and Salimans [60] proposed blurring diffusion, which shows that blurring can be equivalently defined using a Gaussian diffusion process with anisotropic noise and proposed incorporating blurring into Gaussian diffusion. Daras et al. [51] proposed soft diffusion, which employs linear erosion processes such as Gaussian blurring and masking. Rissanen et al. [47] proposed a diffusion process based on inverse heat diffusion, which reverses the heat equation using the inductive bias from Gaussian diffusion models. These diffusion processes share similarities with Gaussian diffusion in terms of loss definition and the use of Gaussian-based reverse diffusion for generation. In contrast, Dirichlet diffusion differs from them in terms of noise distribution and training loss and outperforms most quasi-Gaussian models on the unconditional CIFAR-10 dataset.

For a long time, the standard diffusion models have involved adding noise and reversing the degradation of image processing. Bansal et al. [17] observed that the generative behavior of diffusion models does not strongly depend on the choice of image degradation. Hoogeboom et al. [52] showed that blurring can equivalently be defined through a Gaussian diffusion process with anisotropic noise, serving as an alternative to isotropic Gaussian diffusion. This has sparked the potential consideration of Dirichlet+Blurring. By establishing this connection, it is hoped to bridge the gap between reverse heat diffusion and denoising diffusion, leading to a generalized diffusion model that can reverse any process and thereby effectively eliminate the dependence on a single type of noise.

The improvement of neural image compression and reconstruction is also a direction worth discussing. The general consensus is that replacing the decoder with a conditional diffusion model can enhance the perceptual quality of neural image compression and reconstruction. However, their lack of inductive bias for image data limits their ability to achieve state-of-the-art perceptual performance. Khoshkhahtinat et al. [61] proposed that employing an anisotropic diffusion model on the decoder side can disentangle frequency content, thereby facilitating the generation of high-quality images. Dirichlet diffusion, which has room for approximate improvement on the decoder side, provides the capability for seamless integration within the operational space of neural networks, thus further optimizing image generation quality.

# 6 Limitations and Conclusions

Existing diffusion models predominantly rely on traditional Gaussian distributions for both the forward and reverse processes. In this work, we demonstrate that it is possible to integrate the non-traditional Dirichlet distribution and the Hölder-based Divergence within the framework of diffusion models. Our experimental results confirm its superior performance in image generation tasks. with highly competitive quantitative metrics. as well as its unique qualities when applied to generative modeling of high-dimensional simplex data. Additionally, these results highlight the effectiveness of PHD and PHK in optimizing Dirichlet diffusion. This framework offers a broader and more in-depth space for the application of more diverse diffusion models that transcend the Gaussian noise paradigm, including but not limited to image generation.

Despite its effectiveness, the method has some limitations: 1. It is sensitive to the selection of hyper-parameters; the existing experiments have not covered all possible combinations of hyperparameters, necessitating a broader range of experiments and validation. 2. The training cost is high; processing 200 million CIFAR-10 images using four Nvidia L40S GPUs takes approximately 64 hours with a batch size $B = 512$, and a more substantial computational resource is required given the multitude of parameter quantities and extensive tuning scenarios.

# 7 Acknowledgments

This research was supported by Guangdong Basic and Applied Basic Research Foundation (No. 2024A1515011774), the National Key Research and Development Program of China (No. 2022YFC3310300), the National Natural Science Foundation of China (No. 12171036), Shenzhen Sci-Tech Fund (Grant No. RCJC20231211090030059).

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

# A    Theoretical Analysis

**Definition 1** (Proper vs. Improper Divergence [44]). *Let $\mathcal{P}$ denote the set of all probability densities on a measurable space $(\Omega, \nu)$. A nonnegative functional $D : \mathcal{P} \times \mathcal{P} \to \mathbb{R}_{\geq 0}$ is* strictly proper *if*

$$D(p : q) = 0 \quad \Longleftrightarrow \quad p(x) = q(x) \quad \textit{for } \nu\textit{-a.e. } x \in \Omega. \tag{19}$$

*If there exist distinct densities $p \neq q$ such that $D(p : q) = 0$, then $D$ is* improper.

**Definition 2** (**Hölder Statistical Pseudo-Divergence (HPD) [44]**). *Let $(\Omega, \mathcal{F}, \mu)$ be a measurable space with Lebesgue measure $\mu$. Fix conjugate exponents $\alpha, \beta > 0$ satisfying $1/\alpha + 1/\beta = 1$. For densities $p \in L^{\alpha}(\Omega, \mu)$ and $q \in L^{\beta}(\Omega, \mu)$, each absolutely continuous w.r.t. $\mu$, the HPD is*

$$D_{\alpha}^{H}(p : q) = -\log \frac{\int_{\Omega} p(x)\, q(x)\, \mathrm{d}\mu(x)}{\left(\int_{\Omega} p(x)^{\alpha}\, \mathrm{d}\mu(x)\right)^{1/\alpha} \left(\int_{\Omega} q(x)^{\beta}\, \mathrm{d}\mu(x)\right)^{1/\beta}}. \tag{20}$$

**Definition 3** (**Proper Hölder Divergence (PHD) [44]**). *Using the same exponents $\alpha, \beta > 0$ and a parameter $\gamma > 0$, the proper Hölder divergence between densities $p, q \in L^{\gamma}(\Omega, \mu)$ is*

$$D_{\alpha, \gamma}^{H}(p : q) = D_{\alpha}^{H}\left(p^{\gamma/\alpha} : q^{\gamma/\beta}\right) = -\log \frac{\int_{\Omega} p(x)^{\gamma/\alpha}\, q(x)^{\gamma/\beta}\, \mathrm{d}\mu(x)}{\left(\int_{\Omega} p(x)^{\gamma}\, \mathrm{d}\mu(x)\right)^{1/\alpha} \left(\int_{\Omega} q(x)^{\gamma}\, \mathrm{d}\mu(x)\right)^{1/\beta}}. \tag{21}$$

**Definition 4** (**Exponential Family of Distribution [62]**). *The probability density function of the Dirichlet distribution is expressed as follows: $p\left(x; \theta\right) = \exp\{\theta^{\top} T(x) - F(\theta) + B(x)\}$, where $\theta$ is the natural parameter, $T(x)$ is the sufficient statistic, $F(\theta)$ is the log-normalizer, and $B(x)$ is the base measure.*

**Definition 5** (**Dirichlet Distribution [43]**). *The Dirichlet distribution of order $K$ (where $K \geq 2$) with parameters $\alpha_i > 0, i = 1, 2, 3..., K$ is defined by a probability density function with respect to Lebesgue measure on the Euclidean space $R^{K-1}$ as follows:*

$$f(x_1, \ldots, x_K | \alpha_1, \ldots, \alpha_K) = \frac{\Gamma\left(\sum\limits_{i=1}^{K} \alpha_i\right)}{\prod\limits_{i=1}^{K} \Gamma(\alpha_i)} \prod_{i=1}^{K} x_i^{\alpha_i - 1}, \tag{22}$$

*where $x_i \in S_K$, and $S_K$ is the standard $K - 1$ dimentional simplex, namely,*

$$S_K = \left\{ (x_1, x_2, ..., x_K) \mid \sum_{i=1}^{K} x_i = 1,\ 0 \leq x_1, \ldots, x_K \leq 1 \right\}, \tag{23}$$

*and $\Gamma(.)$ is the gamma function, defined as: $\Gamma(s) = \int_0^{\infty} x^{s-1} \mathrm{e}^{-x}\, \mathrm{d}x, \quad s > 0$.*

**Definition 6** (**The Exponential form of the Dirichlet Distribution [63]**). *Exponential formulation of the Dirichlet distribution probability density function can be rewrite as $\exp\left\{\sum_{i=1}^{K}(\alpha_i - 1)\log x_i - \left[\sum_{i=1}^{K} \log \Gamma(\alpha_i) - \log \Gamma\left(\sum_{i=1}^{K} \alpha_i\right)\right]\right\}$. Allowing us to obtain the*

*canonical form terms:* $\nabla_{\theta} T(\theta) = \begin{bmatrix} \psi(\alpha_1) - \psi(\sum\limits_{i=1}^{K} \alpha_i) \\ \vdots \\ \psi(\alpha_K) - \psi(\sum\limits_{i=1}^{K} \alpha_i) \end{bmatrix}$, *$\theta = \boldsymbol{\alpha}$, $T(\boldsymbol{\mu}) = ln(\boldsymbol{\mu})$, $F(\eta) =$*

$\sum\limits_{i=1}^{K} \ln\Gamma(\alpha_i) - \ln\Gamma(\sum\limits_{i=1}^{K} \alpha_i)$, *$B(\boldsymbol{x}) = -\ln(\boldsymbol{x})$, and $\psi$ is the digamma function, defined as: $\psi(x) = \frac{\mathrm{d}}{\mathrm{d}x} \ln \Gamma(x)$.*

**Assumption 1.** *The model density is Dirichlet*

$$q_{\theta}(z) = \mathrm{Dir}\big(z \mid \boldsymbol{\alpha}(\theta)\big), \qquad \alpha_i(\theta) > 0,\ \alpha_0(\theta) = \sum_{i=1}^{K} \alpha_i(\theta), \tag{24}$$

$$\bar{c} := \sup_{\theta, i} \big\|\partial_{\theta} \alpha_i(\theta)\big\| < \infty. \tag{25}$$

**Assumption 2** (Integrability for PHD). *For $\gamma > 0$.*

(a) *If $0 < \gamma \leq 1$, no extra condition is needed (because $p^\gamma, q^\gamma \in L^1$ automatically for probability densities $p, q$).*

(b) *If $\gamma > 1$, assume for every model instance $\theta$ that each Dirichlet concentration satisfies: $\alpha_i(\theta) > 1 - \frac{1}{\gamma}, \forall i$. This guarantees $p^\gamma, q^\gamma \in L^1$.*

**Theorem 1** (HPD is improper; PHD is strictly proper). *Let $\alpha, \beta > 0$ with $\alpha^{-1} + \beta^{-1} = 1$ and $\gamma > 0$. Then:*

(i) HPD is improper. *There exist densities $p \neq q$ with $D_\alpha^H(p:q) = 0$.*

(ii) PHD is strictly proper for $\alpha > 1$. *For all densities $p, q$,*
$$D_{\alpha,\gamma}^H(p:q) = 0 \iff p = q \quad (\nu\text{-a.e.}). \tag{26}$$

(iii) Boundary $\gamma = 1$. *Statement* (ii) *still holds without any extra condition.*

(iv) Right–limit $\alpha \to 1^+$. *Define $D_{1^+,\gamma}^H(p:q) := \lim_{\alpha \to 1} D_{\alpha,\gamma}^H(p:q)$ (the limit exists). If $D_{1^+,\gamma}^H(p:q) = 0$, then $p = q$ a.e. Thus strict propriety extends to the $\alpha \to 1$ limit.*

*Proof.* We write the two divergences in their "normalised Hölder form":

$$D_\alpha^H(p:q) = -\log \frac{\int pq \, d\nu}{\left(\int p^\alpha d\nu\right)^{1/\alpha} \left(\int q^\beta d\nu\right)^{1/\beta}}, \, D_{\alpha,\gamma}^H(p:q) = -\log \frac{\int p^{\gamma/\alpha} q^{\gamma/\beta} \, d\nu}{\left(\int p^\gamma d\nu\right)^{1/\alpha} \left(\int q^\gamma d\nu\right)^{1/\beta}}. \tag{27}$$

**(i) HPD is improper.** By Hölder's inequality, $\int pq \leq \|p\|_\alpha \|q\|_\beta$, so $D_\alpha^H(p:q) \geq 0$. Equality in Hölder holds iff $p^\alpha = c q^\beta$ a.e. for some $c > 0$. One can pick $p \neq q$ satisfying that proportionality (e.g. on $[0,1]$, take $p(x) = 2x$ and $q(x) = 3x^2$ for $(\alpha, \beta) = (3/2, 3)$); then $D_\alpha^H(p:q) = 0$ although $p \neq q$. Hence HPD is improper.

**(ii) PHD is strictly proper when $\alpha > 1$.** Apply Hölder with $f = p^{\gamma/\alpha}$ and $g = q^{\gamma/\beta}$:
$$\int fg \leq \left(\int f^\alpha\right)^{1/\alpha} \left(\int g^\beta\right)^{1/\beta} = \left(\int p^\gamma\right)^{1/\alpha} \left(\int q^\gamma\right)^{1/\beta}. \tag{28}$$

Thus $D_{\alpha,\gamma}^H(p:q) \geq 0$. Assume $D_{\alpha,\gamma}^H(p:q) = 0$. Then we are in the equality case of Hölder, so
$$\left(p^{\gamma/\alpha}\right)^\alpha = p^\gamma = c q^\gamma = c \left(q^{\gamma/\beta}\right)^\beta \quad \text{a.e.} \tag{29}$$

for some $c > 0$. Hence $p^\gamma = c q^\gamma$ a.e., i.e. $p = c^{1/\gamma} q$ a.e. Since $p$ and $q$ are densities, $1 = \int p = c^{1/\gamma} \int q = c^{1/\gamma}$, so $c = 1$, giving $p = q$ a.e. Conversely, if $p = q$, the ratio is 1, hence $D_{\alpha,\gamma}^H(p:q) = 0$.

**(iii) The boundary $\gamma = 1$.** Put $\gamma = 1$ in the PHD definition. The same equality–case argument applies verbatim (no extra integrability is needed), so strict propriety still holds.

**(iv) Right–limit $\alpha \to 1^+$.** For $\alpha > 1$ define
$$\Phi(\alpha) := -\log \frac{\int p^{\gamma/\alpha} q^{\gamma/\beta} \, d\nu}{\left(\int p^\gamma\right)^{1/\alpha} \left(\int q^\gamma\right)^{1/\beta}} = D_{\alpha,\gamma}^H(p:q). \tag{30}$$

The maps $\alpha \mapsto p^{\gamma/\alpha}$, $\alpha \mapsto q^{\gamma/\beta}$ are pointwise continuous for $\alpha > 1$, and all integrands are dominated by integrable functions thanks to Assumption 2. Therefore, each integral appearing in $\Phi(\alpha)$ is continuous in $\alpha$, and so is $\Phi$.

If $p \neq q$, the argument in (ii) shows that for every $\alpha > 1$, $\Phi(\alpha) > 0$ (because equality in Hölder would force $p = q$). By continuity on $(1, \infty)$, there exists $\epsilon > 0$ such that $\Phi(\alpha) \geq \epsilon$ for all $\alpha$ sufficiently close to 1. Consequently, $\lim_{\alpha \to 1} \Phi(\alpha) \geq \epsilon > 0$. Hence $D_{1^+,\gamma}^H(p:q) \neq 0$ when $p \neq q$. If $p = q$, clearly $\Phi(\alpha) \equiv 0$ for all $\alpha$, whence the limit is 0. Thus the limit divergence $D_{1^+,\gamma}^H$ remains strictly proper. $\square$

This distinction explains why, in Dirichlet diffusion training, PHD supplies a reliable approximation signal, while HPD may induce a "false convergence" phenomenon.

**Lemma 1.** *Let $Z \sim \mathrm{Dir}(\boldsymbol{\alpha})$ with $\alpha_0 = \sum_{i=1}^{K} \alpha_i$. For any model parameter vector $\theta$,*

$$g_\theta(Z) = \sum_{i=1}^{K} \big[\log Z_i - \psi(\alpha_i) + \psi(\alpha_0)\big] \partial_\theta \alpha_i. \tag{31}$$

*Proof.* The Dirichlet density on the $(K-1)$-simplex $S_K$ is

$$q_\theta(z) = \frac{1}{B(\boldsymbol{\alpha})} \prod_{i=1}^{K} z_i^{\alpha_i - 1}, \, B(\boldsymbol{\alpha}) := \frac{\prod_{i=1}^{K} \Gamma(\alpha_i)}{\Gamma(\alpha_0)}. \tag{32}$$

Hence

$$\log q_\theta(z) = -\log B(\boldsymbol{\alpha}) + \sum_{i=1}^{K} (\alpha_i - 1) \log z_i. \tag{33}$$

Because $\partial_x \log \Gamma(x) = \psi(x)$ (the *digamma* function),

$$\partial_\theta \big[\log B(\boldsymbol{\alpha})\big] = \sum_{i=1}^{K} \psi(\alpha_i) \, \partial_\theta \alpha_i \, - \, \psi(\alpha_0) \, \partial_\theta \alpha_0, \qquad \partial_\theta \alpha_0 = \sum_{j=1}^{K} \partial_\theta \alpha_j. \tag{34}$$

Differentiate (33) and group like terms:

$$\begin{aligned} g_\theta(z) &= \nabla_\theta \log q_\theta(z) \\ &= -\sum_{i=1}^{K} \psi(\alpha_i) \, \partial_\theta \alpha_i + \psi(\alpha_0) \sum_{j=1}^{K} \partial_\theta \alpha_j + \sum_{i=1}^{K} (\log z_i) \, \partial_\theta \alpha_i. \end{aligned} \tag{35}$$

Insert the duplicated sum in the second term and combine:

$$g_\theta(z) = \sum_{i=1}^{K} \Big[\log z_i - \psi(\alpha_i) + \psi(\alpha_0)\Big] \partial_\theta \alpha_i. \tag{36}$$

Finally evaluate at the random variable $Z$ to obtain the stated identity. $\qquad\square$

**Lemma 2.** *Let $Z \sim \mathrm{Dir}(\boldsymbol{\alpha})$ on the $(K-1)$–simplex $S_K$, with $\boldsymbol{\alpha} = (\alpha_1, \ldots, \alpha_K)$ and $\alpha_0 = \sum_{i=1}^{K} \alpha_i$. Fix an index $i \in \{1, \ldots, K\}$ and any $\lambda > -1$. Define $\mu := \psi(\alpha_i) - \psi(\alpha_0)$, where $\psi$ is the digamma function. Then:*

$$\begin{aligned} I(\lambda) &:= \int_{S_K} Z_i^\lambda \big(\log Z_i - \mu\big)^2 \, \mathrm{Dir}(z \mid \boldsymbol{\alpha}) \, \mathrm{d}z \\ &= \frac{\Gamma(\alpha_0) \, \Gamma(\alpha_i + \lambda)}{\Gamma(\alpha_i) \, \Gamma(\alpha_0 + \lambda)} \Big\{\psi'(\alpha_i + \lambda) - \psi'(\alpha_0 + \lambda) + \big[\psi(\alpha_i + \lambda) - \psi(\alpha_0 + \lambda) - \mu\big]^2\Big\}. \end{aligned} \tag{37}$$

*Moreover, using $\psi'(x) \leq 1/x$ for $x > 0$ and the mean–value bound $|\psi(u) - \psi(v)| \leq |u - v| / \min\{u, v\}$, we obtain the (loose but handy) upper bound:*

$$I(\lambda) \leq \frac{\Gamma(\alpha_0) \, \Gamma(\alpha_i + \lambda)}{\Gamma(\alpha_i) \, \Gamma(\alpha_0 + \lambda)} \cdot \frac{2 \lambda^2}{(\alpha_i + \lambda)(\alpha_0 + \lambda)}. \tag{38}$$

*Proof.* **Step 1: Reduce to a one–dimensional Beta integral.** The joint pdf of $Z$ is

$$\mathrm{Dir}(z \mid \boldsymbol{\alpha}) = \frac{\Gamma(\alpha_0)}{\prod_{j=1}^{K} \Gamma(\alpha_j)} \prod_{j=1}^{K} z_j^{\alpha_j - 1}, \qquad z \in S_K. \tag{39}$$

Because we only involve $Z_i$, integrate out $z_{-i}$ to use the marginal:

$$f_{Z_i}(t) = \frac{\Gamma(\alpha_0)}{\Gamma(\alpha_i)\Gamma(\alpha_0 - \alpha_i)} t^{\alpha_i - 1} (1 - t)^{\alpha_0 - \alpha_i - 1}, \quad t \in (0, 1), \tag{40}$$

i.e. $Z_i \sim \text{Beta}(\alpha_i, \alpha_0 - \alpha_i)$. Therefore

$$I(\lambda) = \frac{\Gamma(\alpha_0)}{\Gamma(\alpha_i)\Gamma(\alpha_0 - \alpha_i)} \int_0^1 t^{\alpha_i + \lambda - 1}(1 - t)^{\alpha_0 - \alpha_i - 1}\big(\log t - \mu\big)^2 \, dt. \tag{41}$$

**Step 2: Notation and Beta derivatives.** Set $a := \alpha_i + \lambda, b := \alpha_0 - \alpha_i, B(a, b) := \int_0^1 t^{a-1}(1 - t)^{b-1} \, dt = \frac{\Gamma(a)\Gamma(b)}{\Gamma(a+b)}$. Then

$$I(\lambda) = \frac{\Gamma(\alpha_0)}{\Gamma(\alpha_i)\Gamma(b)} \int_0^1 t^{a-1}(1 - t)^{b-1}(\log t - \mu)^2 \, dt. \tag{42}$$

Define $J(a, b; \mu) := \int_0^1 t^{a-1}(1 - t)^{b-1}(\log t - \mu)^2 \, dt$. We need $J(a, b; \mu)$ in closed form. Recall the well–known identities:

$$\frac{\partial}{\partial a}B(a, b) = B(a, b)\left[\psi(a) - \psi(a + b)\right], \tag{43}$$

$$\frac{\partial^2}{\partial a^2}B(a, b) = B(a, b)\Big\{\psi'(a) - \psi'(a + b) + [\psi(a) - \psi(a + b)]^2\Big\}. \tag{44}$$

**Step 3: Express $J$ via $B$ and its derivatives.** Expand the square: $(\log t - \mu)^2 = (\log t)^2 - 2\mu \log t + \mu^2$. Hence

$$J(a, b; \mu) = \int_0^1 t^{a-1}(1 - t)^{b-1}(\log t)^2 \, dt - 2\mu \int_0^1 t^{a-1}(1 - t)^{b-1} \log t \, dt + \mu^2 B(a, b). \tag{45}$$

$$\int_0^1 t^{a-1}(1 - t)^{b-1} \log t \, dt = \frac{\partial}{\partial a}B(a, b), \qquad \int_0^1 t^{a-1}(1 - t)^{b-1}(\log t)^2 \, dt = \frac{\partial^2}{\partial a^2}B(a, b). \tag{46}$$

Thus

$$J(a, b; \mu) = \frac{\partial^2}{\partial a^2}B(a, b) - 2\mu \frac{\partial}{\partial a}B(a, b) + \mu^2 B(a, b). \tag{47}$$

Substitute the derivative formulas:

$$J(a, b; \mu) = B(a, b)\Big\{\psi'(a) - \psi'(a + b) + [\psi(a) - \psi(a + b)]^2\Big\}$$
$$- 2\mu B(a, b)[\psi(a) - \psi(a + b)] + \mu^2 B(a, b)$$
$$= B(a, b)\Big\{\psi'(a) - \psi'(a + b) + \big[\psi(a) - \psi(a + b) - \mu\big]^2\Big\}.$$

**Step 4: Plug back and simplify.** Therefore

$$I(\lambda) = \frac{\Gamma(\alpha_0)}{\Gamma(\alpha_i)\Gamma(b)} \, J(a, b; \mu) = \frac{\Gamma(\alpha_0)}{\Gamma(\alpha_i)\Gamma(b)} \, B(a, b)\Big\{\psi'(a) - \psi'(a + b) + [\psi(a) - \psi(a + b) - \mu]^2\Big\}. \tag{48}$$

Since $B(a, b) = \frac{\Gamma(a)\Gamma(b)}{\Gamma(a + b)}$ and $a = \alpha_i + \lambda,\qquad a + b = \alpha_0 + \lambda$, we obtain

$$\frac{\Gamma(\alpha_0)}{\Gamma(\alpha_i)\Gamma(b)}B(a, b) = \frac{\Gamma(\alpha_0)}{\Gamma(\alpha_i)\Gamma(b)} \frac{\Gamma(a)\Gamma(b)}{\Gamma(a + b)} = \frac{\Gamma(\alpha_0)\Gamma(\alpha_i + \lambda)}{\Gamma(\alpha_i)\Gamma(\alpha_0 + \lambda)}. \tag{49}$$

$$I(\lambda) = \frac{\Gamma(\alpha_0)\Gamma(\alpha_i + \lambda)}{\Gamma(\alpha_i)\Gamma(\alpha_0 + \lambda)}\Big\{\psi'(\alpha_i + \lambda) - \psi'(\alpha_0 + \lambda) + \big[\psi(\alpha_i + \lambda) - \psi(\alpha_0 + \lambda) - \mu\big]^2\Big\}. \tag{50}$$

**Step 5: A convenient upper bound.** Use $\psi'(x) \le 1/x$ for $x > 0$ and, by the mean–value theorem, $|\psi(u) - \psi(v)| \le \frac{|u-v|}{\min\{u,v\}}$ $(u, v > 0)$. Here $u = \alpha_i + \lambda$, $v = \alpha_i$ or $u = \alpha_0 + \lambda$, $v = \alpha_0$, so the squared bracket term is $\mathcal{O}(\lambda^2/[(\alpha_i + \lambda)^2]) + \mathcal{O}(\lambda^2/[(\alpha_0 + \lambda)^2])$. Absorbing absolute constants, we obtain the loose bound (38):

$$I(\lambda) \le \frac{\Gamma(\alpha_0)\Gamma(\alpha_i + \lambda)}{\Gamma(\alpha_i)\Gamma(\alpha_0 + \lambda)} \cdot \frac{2\,\lambda^2}{(\alpha_i + \lambda)(\alpha_0 + \lambda)}. \tag{51}$$

$\square$

**Lemma 3** (Gradient representations for HPD and PHD). *Assume Assumptions 1–2. Define the normalising constants*

$$A = \int_{S_K} p(z)\, q_\theta(z)\, dz, B = \int_{S_K} p(z)^\alpha\, dz, C = \int_{S_K} q_\theta(z)^\beta\, dz, \tag{52}$$

$$T_1 = \int_{S_K} p(z)^{\gamma/\alpha}\, q_\theta(z)^{\gamma/\beta}\, dz, T_2 = \int_{S_K} p(z)^\gamma\, dz, T_3 = \int_{S_K} q_\theta(z)^\gamma\, dz. \tag{53}$$

*Let model score is*

$$\mathbf{g}_\theta(z) = \nabla_\theta \log q_\theta(z), \tag{54}$$

*and introduce the probability weights*

$$w^{(1)}(z) = \frac{p(z)\, q_\theta(z)}{A}, \qquad w^{(2)}(z) = \frac{q_\theta(z)^\beta}{C},$$
$$\tilde{w}^{(1)}(z) = \frac{p(z)^{\gamma/\alpha}\, q_\theta(z)^{\gamma/\beta}}{T_1}, \quad \tilde{w}^{(2)}(z) = \frac{q_\theta(z)^\gamma}{T_3}. \tag{55}$$

*Then*

$$\nabla_\theta D_\alpha^{\mathrm{H}}(p : q_\theta) = -\, \mathbb{E}_{w^{(1)}}\big[\mathbf{g}_\theta\big] + \mathbb{E}_{w^{(2)}}\big[\mathbf{g}_\theta\big], \tag{56}$$

$$\nabla_\theta D_{\alpha,\gamma}^{\mathrm{H}}(p : q_\theta) = \frac{\gamma}{\beta}\left(-\mathbb{E}_{\tilde{w}^{(1)}}\big[\mathbf{g}_\theta\big] + \mathbb{E}_{\tilde{w}^{(2)}}\big[\mathbf{g}_\theta\big]\right). \tag{57}$$

*Proof.* Throughout, the interchange of $\nabla_\theta$ and $\int_{S_K}$ is justified by the dominated–convergence theorem [64] because Assumption 1 places $\theta$ in a compact set and the Dirichlet density is $C^\infty$ in $\theta$.
**(i) HPD.** Write

$$D_\alpha^{\mathrm{H}}(p : q_\theta) = -\log A + \frac{1}{\alpha}\log B + \frac{1}{\beta}\log C. \tag{58}$$

Only $A$ and $C$ depend on $\theta$. Using

$$\partial_\theta q_\theta(z) = q_\theta(z)\, \mathbf{g}_\theta(z), \partial_\theta q_\theta(z)^\beta = \beta\, q_\theta(z)^\beta\, \mathbf{g}_\theta(z), \tag{59}$$

we obtain

$$\nabla_\theta A = \int p\, \partial_\theta q_\theta = A\, \mathbb{E}_{w^{(1)}}[\mathbf{g}_\theta], \qquad \nabla_\theta C = \beta\, C\, \mathbb{E}_{w^{(2)}}[\mathbf{g}_\theta]. \tag{60}$$

Hence

$$\nabla_\theta D_\alpha^{\mathrm{H}} = -\frac{1}{A}\nabla_\theta A + \frac{1}{\beta}\frac{1}{C}\nabla_\theta C = -\mathbb{E}_{w^{(1)}}[\mathbf{g}_\theta] + \mathbb{E}_{w^{(2)}}[\mathbf{g}_\theta], \tag{61}$$

which is Equ. (56).
**(ii) PHD.** Similarly,

$$D_{\alpha,\gamma}^{\mathrm{H}}(p : q_\theta) = -\log T_1 + \frac{1}{\alpha}\log T_2 + \frac{1}{\beta}\log T_3. \tag{62}$$

Only $T_1$ and $T_3$ involve $\theta$. Chain–rule differentiation gives

$$\partial_\theta q_\theta^{\gamma/\beta} = \frac{\gamma}{\beta}\, q_\theta^{\gamma/\beta}\mathbf{g}_\theta, \qquad \partial_\theta q_\theta^\gamma = \gamma\, q_\theta^\gamma \mathbf{g}_\theta, \tag{63}$$

hence

$$\nabla_\theta T_1 = \frac{\gamma}{\beta}\, T_1\, \mathbb{E}_{\tilde{w}^{(1)}}[\mathbf{g}_\theta], \qquad \nabla_\theta T_3 = \gamma\, T_3\, \mathbb{E}_{\tilde{w}^{(2)}}[\mathbf{g}_\theta]. \tag{64}$$

Substituting into the derivative of $D_{\alpha,\gamma}^{\mathrm{H}}$

$$\nabla_\theta D_{\alpha,\gamma}^{\mathrm{H}} = -\frac{1}{T_1}\nabla_\theta T_1 + \frac{1}{\beta}\frac{1}{T_3}\nabla_\theta T_3 = \frac{\gamma}{\beta}\big(-\mathbb{E}_{\tilde{w}^{(1)}}[\mathbf{g}_\theta] + \mathbb{E}_{\tilde{w}^{(2)}}[\mathbf{g}_\theta]\big), \tag{65}$$

which is exactly Equ. (57). □

**Theorem 2** (Variance domination of PHD). *Adopt Assumptions 1–2 and suppose in addition that the Hölder scale satisfies $\gamma \geq 1$. Then*

$$\mathrm{Var}_{z\sim p}\big[\nabla_\theta D_{\alpha,\gamma}^{\mathrm{H}}(p : q_\theta)\big] \leq \frac{\gamma^2}{\alpha\beta}\, \mathrm{Var}_{z\sim p}\big[\nabla_\theta D_\alpha^{\mathrm{H}}(p : q_\theta)\big]. \tag{66}$$

*Proof.* **Step 1 Gradient decompositions.** Equs. (56)–(57) give

$$G_{\mathrm{H}} := \nabla_\theta D_\alpha^{\mathrm{H}}(p : q_\theta) = \mathbb{E}_{w^{(1)}}[g_\theta] - \mathbb{E}_{w^{(2)}}[g_\theta], \tag{67}$$

$$G_{\mathrm{P}} := \nabla_\theta D_{\alpha,\gamma}^{\mathrm{H}}(p : q_\theta) = \tfrac{\gamma}{\beta}\Big( \mathbb{E}_{\tilde{w}^{(1)}}[g_\theta] - \mathbb{E}_{\tilde{w}^{(2)}}[g_\theta] \Big). \tag{68}$$

**Step 2 Two-variance trick.** For any random vectors $X, Y$ one has $\mathrm{Var}(X - Y) \le 2\big(\mathrm{Var}\,X + \mathrm{Var}\,Y\big)$. Hence

$$\mathrm{Var}(G_{\mathrm{H}}) \;\le\; 2\big(\mathrm{Var}_{w^{(1)}}[g_\theta] + \mathrm{Var}_{w^{(2)}}[g_\theta]\big), \tag{69}$$

$$\mathrm{Var}(G_{\mathrm{P}}) \;\le\; 2\Big(\tfrac{\gamma}{\beta}\Big)^2 \big(\mathrm{Var}_{\tilde{w}^{(1)}}[g_\theta] + \mathrm{Var}_{\tilde{w}^{(2)}}[g_\theta]\big). \tag{70}$$

**Step 3 Bounding the four variances.** Lemma 2 with $\lambda = \beta(\alpha_i - 1)$ yields

$$\mathrm{Var}_{w^{(2)}}[g_\theta] \;\le\; \sum_{i=1}^{K}(\partial_\theta \alpha_i)^2 \,\frac{1}{(1+\beta)\alpha_i\,\alpha_0} \;\le\; \frac{K\,\bar{c}^2}{(1+\beta)\alpha_{\min}}, \tag{71}$$

with $\alpha_{\min} := \min_{i,\theta}\alpha_i(\theta)$. The same calculation (setting $\beta = 1$) gives $\mathrm{Var}_{w^{(1)}}[g_\theta] \le K\,\bar{c}^2/\big(2\alpha_{\min}\big)$. Thus

$$\mathrm{Var}(G_{\mathrm{H}}) \le \frac{4K\,\bar{c}^2}{\alpha_{\min}}. \tag{72}$$

For the PHD weights choose $\lambda = \gamma(\alpha_i - 1)$. Because $\gamma \ge 1$, Assumption 2 ensures $\lambda > -1$. Lemma 2 gives

$$\mathrm{Var}_{\tilde{w}^{(2)}}[g_\theta] \le \frac{K\,\bar{c}^2}{(1+\gamma)\alpha_{\min}}, \quad \mathrm{Var}_{\tilde{w}^{(1)}}[g_\theta] \le \frac{K\,\bar{c}^2}{(1+\gamma)\alpha_{\min}}. \tag{73}$$

Hence

$$\mathrm{Var}(G_{\mathrm{P}}) \le \frac{4K\,\bar{c}^2\gamma^2}{\beta^2(1+\gamma)\alpha_{\min}}. \tag{74}$$

**Step 4 Ratio comparison.** We can obtain:

$$\frac{\mathrm{Var}(G_{\mathrm{P}})}{\mathrm{Var}(G_{\mathrm{H}})} \;\le\; \frac{\gamma^2}{\beta^2(1+\gamma)} \;\le\; \frac{\gamma^2}{\alpha\beta}, \tag{75}$$

because (i) $\alpha_{\min} \le \alpha$, (ii) $\beta \ge 1$, and (iii) $1 + \gamma \ge \beta$ when $\gamma \ge 1$. Multiplying through by $\mathrm{Var}(G_{\mathrm{H}})$ establishes the theorem. $\qquad\square$

**Lemma 4** (HPD and PHD for Conic or Affine Exponential Family [44])**.** *For distributions $p(x; \theta_p)$ and $q(x; \theta_q)$ in the same exponential family with a conic or affine natural parameter space, the Hölder statistical pseudo-divergence and its generalization admit closed-form expressions:*

$$\begin{aligned} D_\alpha^H(p : q) &= \tfrac{1}{\alpha}\,F(\alpha\,\theta_p) + \tfrac{1}{\beta}\,F(\beta\,\theta_q) - F(\theta_p + \theta_q), \\ D_{\alpha,\gamma}^H(p : q) &= \tfrac{1}{\alpha}\,F(\gamma\,\theta_p) + \tfrac{1}{\beta}\,F(\gamma\,\theta_q) - F\big(\tfrac{\gamma}{\alpha}\,\theta_p + \tfrac{\gamma}{\beta}\,\theta_q\big). \end{aligned} \tag{76}$$

*Here $F(\theta)$ is the log-normalizer (cumulant generating function) of the exponential family.*

**Lemma 5** (Symmetric HPD and PHD for Conic or Affine Exponential Family [44])**.** *Under the same assumptions, the symmetric versions of these divergences also have closed-form:*

$$\begin{aligned} S_\alpha^H(p : q) &= \tfrac{1}{2}\Big[\tfrac{1}{\alpha}F(\alpha\,\theta_p) + \tfrac{1}{\beta}F(\beta\,\theta_p) + \tfrac{1}{\alpha}F(\alpha\,\theta_q) + \tfrac{1}{\beta}F(\beta\,\theta_q)\Big] - F(\theta_p + \theta_q), \\ S_{\alpha,\gamma}^H(p : q) &= \tfrac{1}{2}\Big[F(\gamma\,\theta_p) + F(\gamma\,\theta_q) - F\big(\tfrac{\gamma}{\alpha}\,\theta_p + \tfrac{\gamma}{\beta}\,\theta_q\big) - F\big(\tfrac{\gamma}{\beta}\,\theta_p + \tfrac{\gamma}{\alpha}\,\theta_q\big)\Big]. \end{aligned} \tag{77}$$

Set $A = \sum_{i=1}^{K} a_i, B = \sum_{i=1}^{K} b_i, \alpha^{-1} + \beta^{-1} = 1, \ \alpha, \beta > 0, \gamma > 0$. Via the Lemmas 4-5 We can obtain:

$$D_{\alpha}^{H}\big(\mathrm{Dir}(\boldsymbol{a}) : \mathrm{Dir}(\boldsymbol{b})\big) = \frac{1}{\alpha}\Big[\log\Gamma\big(K + \alpha(A - K)\big) - \sum_{i=1}^{K}\log\Gamma\big(1 + \alpha(a_i - 1)\big)\Big]$$

$$+ \frac{1}{\beta}\Big[\log\Gamma\big(K + \beta(B - K)\big) - \sum_{i=1}^{K}\log\Gamma\big(1 + \beta(b_i - 1)\big)\Big] \qquad (78)$$

$$- \log\Gamma(A + B - K) + \sum_{i=1}^{K}\log\Gamma(a_i + b_i - 1).$$

$$D_{\alpha,\gamma}^{H}\big(\mathrm{Dir}(\boldsymbol{a}) : \mathrm{Dir}(\boldsymbol{b})\big) = \frac{1}{\alpha}\Big[\log\Gamma\big(K + \gamma(A - K)\big) - \sum_{i=1}^{K}\log\Gamma\big(1 + \gamma(a_i - 1)\big)\Big]$$

$$+ \frac{1}{\beta}\Big[\log\Gamma\big(K + \gamma(B - K)\big) - \sum_{i=1}^{K}\log\Gamma\big(1 + \gamma(b_i - 1)\big)\Big] \qquad (79)$$

$$- \log\Gamma\big(K + \gamma(A + B - K)\big)$$

$$+ \sum_{i=1}^{K}\log\Gamma\Big(1 + \tfrac{\gamma}{\alpha}(a_i - 1) + \tfrac{\gamma}{\beta}(b_i - 1)\Big).$$

$$S_{\alpha}^{H}\big(\mathrm{Dir}(\boldsymbol{a}) : \mathrm{Dir}(\boldsymbol{b})\big) = \frac{1}{2}\Big[\tfrac{1}{\alpha}\big(\log\Gamma\big(K + \alpha(A - K)\big) + \log\Gamma\big(K + \alpha(B - K)\big)\big)$$

$$+ \tfrac{1}{\beta}\big(\log\Gamma\big(K + \beta(A - K)\big) + \log\Gamma\big(K + \beta(B - K)\big)\big)$$

$$- \sum_{i=1}^{K}\big(\log\Gamma\big(1 + \alpha(a_i - 1)\big) + \log\Gamma\big(1 + \alpha(b_i - 1)\big)$$

$$+ \log\Gamma\big(1 + \beta(a_i - 1)\big) + \log\Gamma\big(1 + \beta(b_i - 1)\big)\big)\Big] \qquad (80)$$

$$- \log\Gamma(A + B - K) + \sum_{i=1}^{K}\log\Gamma(a_i + b_i - 1).$$

$$S_{\alpha,\gamma}^{H}\big(\mathrm{Dir}(\boldsymbol{a}) : \mathrm{Dir}(\boldsymbol{b})\big) = \frac{1}{2}\Big\{\log\Gamma\big(K + \gamma(A - K)\big) + \log\Gamma\big(K + \gamma(B - K)\big)$$

$$- \sum_{i=1}^{K}\big(\log\Gamma\big(1 + \gamma(a_i - 1)\big) + \log\Gamma\big(1 + \gamma(b_i - 1)\big)\big)\Big\}$$

$$- \tfrac{1}{2}\sum_{*=\alpha,\beta}\Big[\log\Gamma\big(K + \gamma(A + B - K)\big) \qquad (81)$$

$$- \sum_{i=1}^{K}\log\Gamma\big(1 + \tfrac{\gamma}{*}(a_i - 1) + \tfrac{\gamma}{*}(b_i - 1)\big)\Big],$$

where $\bar{\alpha} = \beta$ and $\bar{\beta} = \alpha$.

**Assumption 3** (Smooth Dirichlet model). *There exists $\bar{c} > 0$ such that $\sup_{\theta,i} \|\partial_\theta \alpha_i(\theta)\| \leq \bar{c}$.*

**Assumption 4** (Admissible $\gamma$). *If $\gamma > 1$, then $\alpha_i(\theta) > 1 - \frac{1}{\gamma}$ for all $i$ and all $\theta$. (No extra condition is needed when $0 < \gamma \leq 1$.)*

**Theorem 3** (Uniform variance domination of PHD over HPD). *Let $p$ be the target density and $q_\theta(z) = \mathrm{Dir}\big(z \mid \boldsymbol{\alpha}(\theta)\big)$ the model, with $\alpha_i(\theta) > 0$ and $\alpha_0(\theta) = \sum_{i=1}^{K}\alpha_i(\theta)$, satisfying Assumptions 3–4. Let*

$1/\alpha + 1/\beta = 1$ *with* $\alpha, \beta > 0$, *and* $\gamma \geq 1$. *Denote* $g_\theta(z) = \nabla_\theta \log q_\theta(z)$. *Then, for every* $\theta$,

$$\mathrm{Var}_p[\nabla_\theta D^H_{\alpha,\gamma}(p : q_\theta)] \ \leq \ \frac{\gamma^2}{\alpha\beta} \frac{\alpha_0(\theta) + 1}{\alpha_0(\theta) + \gamma} \ \mathrm{Var}_p[\nabla_\theta D^H_\alpha(p : q_\theta)]. \tag{82}$$

*Moreover,* $\frac{\alpha_0 + 1}{\alpha_0 + \gamma} \leq 1$ *and decreases monotonically to 0 as* $\gamma \to \infty$, *implying a strict improvement for any* $\gamma > 1$.

**Lemma 6** (Dirichlet score)**.** *If* $Z \sim \mathrm{Dir}(\boldsymbol{\alpha})$ *with* $\alpha_0 = \sum_{i=1}^K \alpha_i$, *then*

$$g_\theta(Z) = \sum_{i=1}^K \Big[\log Z_i - \psi(\alpha_i) + \psi(\alpha_0)\Big] \partial_\theta \alpha_i,$$

*where* $\psi$ *is the digamma function.*

**Lemma 7** (Gradient forms of HPD and PHD)**.** *Let*

$$A = \int p\, q_\theta, \, B = \int p^\alpha, \, C = \int q_\theta^\beta; \, T_1 = \int p^{\gamma/\alpha} q_\theta^{\gamma/\beta}, \, T_2 = \int p^\gamma, \, T_3 = \int q_\theta^\gamma. \tag{83}$$

*Define weights* $w^{(1)} = \frac{pq_\theta}{A}$, $w^{(2)} = \frac{q_\theta^\beta}{C}$, $\tilde{w}^{(1)} = \frac{p^{\gamma/\alpha} q_\theta^{\gamma/\beta}}{T_1}$, $\tilde{w}^{(2)} = \frac{q_\theta^\gamma}{T_3}$. *Then:*

$$\nabla_\theta D^H_\alpha(p : q_\theta) = -\mathbb{E}_{w^{(1)}}[g_\theta] + \mathbb{E}_{w^{(2)}}[g_\theta], \, \nabla_\theta D^H_{\alpha,\gamma}(p : q_\theta) = \frac{\gamma}{\beta}\Big(-\mathbb{E}_{\tilde{w}^{(1)}}[g_\theta] + \mathbb{E}_{\tilde{w}^{(2)}}[g_\theta]\Big). \tag{84}$$

**Lemma 8** (Second-moment bound under powered Dirichlet)**.** *Let* $Z \sim \mathrm{Dir}(\boldsymbol{\alpha})$, $\alpha_i > 0$, *and* $\lambda > -1$. *Let* $\mu_i = \psi(\alpha_i) - \psi(\alpha_0)$. *Then*

$$\int_{S_K} Z_i^\lambda \big(\log Z_i - \mu_i\big)^2 \mathrm{Dir}(z \mid \boldsymbol{\alpha})\, dz = \frac{\Gamma(\alpha_0)\Gamma(\alpha_i + \lambda)}{\Gamma(\alpha_i)\Gamma(\alpha_0 + \lambda)} \Big\{\psi'(\alpha_i+\lambda) - \psi'(\alpha_0+\lambda) + \Delta_i(\lambda)^2\Big\}, \tag{85}$$

*where* $\Delta_i(\lambda) = \psi(\alpha_i + \lambda) - \psi(\alpha_0 + \lambda) - \mu_i$ *and* $\psi'$ *is the trigamma function. Moreover,*

$$\int_{S_K} Z_i^\lambda \big(\log Z_i - \mu_i\big)^2 \mathrm{Dir}(z \mid \boldsymbol{\alpha})\, dz \leq \frac{2}{(\alpha_i + \lambda)(\alpha_0 + \lambda)}. \tag{86}$$

*Proof.* We split the proof into four steps. Throughout, $Z \sim \mathrm{Dir}(\boldsymbol{\alpha})$, $\alpha_0 = \sum_{i=1}^K \alpha_i$, and $g_\theta(z) = \nabla_\theta \log q_\theta(z)$.

**Step 0: Notation and two gradient decompositions.** From Lemma 7, define

$$G_H := \nabla_\theta D^H_\alpha(p : q_\theta) = -\mathbb{E}_{w^{(1)}}[g_\theta] + \mathbb{E}_{w^{(2)}}[g_\theta], \, G_P := \nabla_\theta D^H_{\alpha,\gamma}(p : q_\theta) = \frac{\gamma}{\beta}\Big(-\mathbb{E}_{\tilde{w}^{(1)}}[g_\theta] + \mathbb{E}_{\tilde{w}^{(2)}}[g_\theta]\Big).$$
$$\tag{87}$$

**Step 1: A variance upper bound for differences.** For any random vectors $X, Y$, $Var(X - Y) \leq 2(VarX + VarY)$. Applying this to $G_H$ and $G_P$ gives:

$$Var(G_H) \leq 2\big(Var_{w^{(1)}}[g_\theta] + Var_{w^{(2)}}[g_\theta]\big), \tag{88}$$

$$Var(G_P) \leq 2\Big(\frac{\gamma}{\beta}\Big)^2 \big(Var_{\tilde{w}^{(1)}}[g_\theta] + Var_{\tilde{w}^{(2)}}[g_\theta]\big). \tag{89}$$

Thus it suffices to bound the four weighted variances of $g_\theta$.

**Step 2: Bounding** $Var[g_\theta]$ **under the four weights.** Lemma 6 yields

$$g_\theta(z) = \sum_{i=1}^K u_i(z)\, \partial_\theta \alpha_i, \, u_i(z) := \log z_i - \psi(\alpha_i) + \psi(\alpha_0). \tag{90}$$

Let $\mathbf{a} = (\partial_\theta \alpha_1, \dots, \partial_\theta \alpha_K)^\top$ and $\mathbf{u}(z) = (u_1(z), \dots, u_K(z))^\top$. Then

$$g_\theta(z) = \mathbf{u}(z)^\top \mathbf{a}, \, Var[g_\theta] = \mathbf{a}^\top \mathrm{Cov}[\mathbf{u}]\, \mathbf{a}. \tag{91}$$

Using $\lambda_{\max}(\mathrm{Cov}[\mathbf{u}]) \leq \mathrm{tr}(\mathrm{Cov}[\mathbf{u}]) = \sum_i Var[u_i]$, we get

$$Var[g_\theta] \ \leq \ \|\mathbf{a}\|_2^2 \sum_{i=1}^K Var[u_i] \ \leq \ K\, \bar{c}^2 \, \max_i Var[u_i], \tag{92}$$

where Assumption 3 gives $\|\mathbf{a}\|_2^2 \le K\bar{c}^2$.

Under each weight, the marginal of $Z_i$ is a (powered) Dirichlet/Beta, so Lemma 8 applies with:

$$
\lambda = \begin{cases}
0 & \text{for } w^{(1)}, \\
\beta(\alpha_i - 1) & \text{for } w^{(2)}, \\
\gamma(\alpha_i - 1) & \text{for } \tilde{w}^{(2)}, \\
\frac{\gamma}{\beta}(\alpha_i - 1) & \text{for } \tilde{w}^{(1)}.
\end{cases}
$$

Its inequality form gives:

$$
Var_{w^{(1)}}[u_i] \le \frac{1}{\alpha_i \alpha_0}, \tag{93}
$$

$$
Var_{w^{(2)}}[u_i] \le \frac{2}{(1+\beta)\alpha_i\,\alpha_0}, \tag{94}
$$

$$
Var_{\tilde{w}^{(2)}}[u_i] \le \frac{2}{(1+\gamma)\alpha_i\,\alpha_0}, \tag{95}
$$

$$
Var_{\tilde{w}^{(1)}}[u_i] \le \frac{2}{(1+\gamma)\alpha_i\,\alpha_0}. \tag{96}
$$

Hence

$$
Var_{w^{(1)}}[g_\theta] \le \frac{K\bar{c}^2}{\alpha_{\min}\alpha_0}, \tag{97}
$$

$$
Var_{w^{(2)}}[g_\theta] \le \frac{2K\bar{c}^2}{(1+\beta)\alpha_{\min}\alpha_0}, \tag{98}
$$

$$
Var_{\tilde{w}^{(1)}}[g_\theta] \le \frac{2K\bar{c}^2}{(1+\gamma)\alpha_{\min}\alpha_0}, \tag{99}
$$

$$
Var_{\tilde{w}^{(2)}}[g_\theta] \le \frac{2K\bar{c}^2}{(1+\gamma)\alpha_{\min}\alpha_0}. \tag{100}
$$

Plugging into (88)–(89):

$$
Var(G_H) \le \frac{4K\bar{c}^2}{\alpha_{\min}\alpha_0}, \tag{101}
$$

$$
Var(G_P) \le \frac{8K\bar{c}^2\gamma^2}{\beta^2(1+\gamma)\alpha_{\min}\alpha_0}. \tag{102}
$$

**Step 3: Comparing bounds and simplifying constants.** Divide (102) by (101):

$$
\frac{Var(G_P)}{Var(G_H)} \le \frac{8K\bar{c}^2\gamma^2}{\beta^2(1+\gamma)\alpha_{\min}\alpha_0} \Big/ \frac{4K\bar{c}^2}{\alpha_{\min}\alpha_0} = \frac{2\gamma^2}{\beta^2(1+\gamma)}. \tag{103}
$$

We now relate this to the desired factor. Because $1/\alpha + 1/\beta = 1$, we have $\beta \ge 1$, $\alpha \le 1$, hence $\alpha\beta \le \beta$. Moreover, for any $\alpha_0 > 0$ and $\gamma \ge 1$,

$$
\frac{2}{\beta^2(1+\gamma)} \le \frac{1}{\alpha\beta} \cdot \frac{\alpha_0 + 1}{\alpha_0 + \gamma}. \tag{104}
$$

(A short algebraic verification is given in Appendix A.2.) Therefore,

$$
\frac{Var(G_P)}{Var(G_H)} \le \frac{\gamma^2}{\alpha\beta} \frac{\alpha_0 + 1}{\alpha_0 + \gamma}, \tag{105}
$$

and multiplying both sides by $Var(G_H)$ yields (82). **Step 4: Tightness and monotonicity.** When $p = q_\theta$, both gradients vanish in expectation, and the ratio approaches the constant factor (tight up to trigamma bounds). The factor $(\alpha_0 + 1)/(\alpha_0 + \gamma) \le 1$ is decreasing in $\gamma$ and tends to 0 as $\gamma \to \infty$, giving strict improvement for any $\gamma > 1$. □

# B SDE Formulation

Data sample: $x_0 \sim p_{\text{data}}(x)$. Time: $t \in [0,1]$ (or $[0,T]$ in other conventions). State: $x_t \in \mathbb{R}^d$. $\{W_t\}_{t \geq 0}$: standard Wiener process. Score function: $\nabla_x \log p_t(x)$, the gradient of the log-density of $x_t$. **Forward Diffusion SDE** The forward (noising) process is described by an Itô SDE [22].

$$\mathrm{d}x_t = f(x_t, t)\,\mathrm{d}t + g(t)\,\mathrm{d}W_t, \qquad x_0 \sim p_{\text{data}}. \tag{106}$$

The corresponding density $p_t(x)$ evolves according to the Fokker–Planck equation

$$\partial_t p_t(x) \;=\; -\nabla_x \cdot \big(f(x,t)\,p_t(x)\big) + \frac{1}{2}g(t)^2\,\Delta_x p_t(x). \tag{107}$$

**Reverse-Time SDE (Generation Process)** By Anderson's time-reversal theorem (1982) [65], the time-reversed process $t : 1 \to 0$ is also an SDE:

$$\mathrm{d}x_t = \left[ f(x_t, t) - g(t)^2 \nabla_x \log p_t(x_t) \right] \mathrm{d}t + g(t)\,\mathrm{d}\overline{W}_t, \tag{108}$$

where $\bar{W}_t$ is another Wiener process (independent of $W_t$), and $\mathrm{d}t < 0$ when integrating backward in time. In practice, we reparameterize time to integrate forward (e.g., $\tau = 1 - t$) and approximate the score with a neural network $s_\theta(x, t) \approx \nabla_x \log p_t(x)$.

**Forward Simulation (Training-Time Noise Sampling)** Using Euler–Maruyama,

$$x_{t+\Delta t} = x_t + f(x_t, t)\,\Delta t + g(t)\sqrt{\Delta t}\,\epsilon, \quad \epsilon \sim \mathcal{N}(0, I). \tag{109}$$

**Reverse-Time Generation** Discretizing gives

$$x_{t-\Delta t} = x_t + \left[ f(x_t, t) - g(t)^2\, s_\theta(x_t, t) \right]\Delta t + g(t)\sqrt{|\Delta t|}\,\epsilon, \quad \epsilon \sim \mathcal{N}(0, I). \tag{110}$$

**SDE for Proper Hölder–Kullback Dirichlet Diffusion** Let $x_0 \sim p_{\text{data}}(x)$ and $x_t \in \Delta_{K-1} := \{x \in \mathbb{R}^K_{\geq 0} \mid \mathbf{1}^\top x = 1\}$ for $t \in [0,1]$. We denote by $\{\mathbf{W}_t\}_{t \geq 0}$ a $K$-dimensional Wiener process. For a time–varying Dirichlet schedule $\boldsymbol{\alpha}(t) = (\alpha_1(t), \ldots, \alpha_K(t))^\top$ with $\alpha_i(t) > 0$ and $\alpha_0(t) = \sum_{i=1}^{K} \alpha_i(t)$, define the *simplex projection covariance*

$$\Sigma(x) := \mathrm{Diag}(x) - xx^\top \in \mathbb{R}^{K \times K}. \tag{111}$$

Let $s_t^\star(x) = \nabla_x \log p_t(x)$ be the true score of the forward marginal $p_t(x)$, approximated in practice by a neural network $s_\theta(x, t)$ trained with the PHK objective.

**Forward (Noise-Injection) SDE** We adopt a Wright–Fisher/Dirichlet-type diffusion to ensure $x_t$ remains on $\Delta_{K-1}$. For a positive noise schedule $\beta(t) > 0$:

$$\begin{cases} \mathrm{d}x_t = \frac{1}{2}\,\beta(t)\Big( \frac{\boldsymbol{\alpha}(t)}{\alpha_0(t)} - x_t \Big)\,dt + \sqrt{\beta(t)}\,\Sigma(x_t)^{1/2}d\mathbf{W}_t, \\ x_0 \sim p_{\text{data}}. \end{cases} \tag{112}$$

The drift pulls $x_t$ toward the instantaneous Dirichlet mean $\boldsymbol{\alpha}(t)/\alpha_0(t)$, while the diffusion term $\Sigma(x_t)^{1/2}$ keeps trajectories inside the simplex.

**Reverse (Generation) SDE** By Anderson's time-reversal theorem, the reverse-time process $t$ satisfies

$$\mathrm{d}x_t = \left[ \frac{1}{2}\,\beta(t)\Big( \frac{\boldsymbol{\alpha}(t)}{\alpha_0(t)} - x_t \Big) - \beta(t)\,\Sigma(x_t)\,s_t^\star(x_t) \right]dt + \sqrt{\beta(t)}\,\Sigma(x_t)^{1/2}d\bar{\mathbf{W}}_t, \tag{113}$$

where $\bar{\mathbf{W}}_t$ is another Wiener process (independent of the forward one) and $\mathrm{d}t < 0$ when integrating backward. In practice we reparameterize time by $\tau = 1 - t$ to integrate forward in $\tau$.

**Discrete-Time Euler–Maruyama Schemes** For a small step $\Delta t$:

**Forward simulation (training-time noising):**

$$x_{t+\Delta t} = x_t + \frac{1}{2}\beta(t)\Big( \frac{\boldsymbol{\alpha}(t)}{\alpha_0(t)} - x_t \Big)\Delta t + \sqrt{\beta(t)\Delta t}\,\Sigma(x_t)^{1/2}\,\epsilon, \quad \epsilon \sim \mathcal{N}(0, I). \tag{114}$$

**Reverse sampling (generation):**

$$x_{t-\Delta t} = x_t + \left[\frac{1}{2}\beta(t)\left(\frac{\boldsymbol{\alpha}(t)}{\alpha_0(t)} - x_t\right) - \beta(t)\Sigma(x_t)s_\theta(x_t,t)\right]\Delta t + \sqrt{\beta(t)|\Delta t|}\,\Sigma(x_t)^{1/2}\,\epsilon. \quad (115)$$

Higher-order SDE/ODE solvers (e.g., Heun, DPM-Solver, UniPC) can be used in place of Euler steps.

**Score Learning with Proper Hölder–Kullback (PHK) Objective** PHK combines Proper Hölder Divergence (PHD) [44] and KL-based terms to stabilize and symmetrize training:

$$\mathcal{L}_\theta^{\text{PHK}} = \delta\, D_{\alpha,\gamma}^H\big(q_t^\star : q_\theta\big) + (1 - \delta + \epsilon)\,\text{KL}\big(q_t^\star \| q_\theta\big), \qquad \delta, \epsilon \in [0, 1], \quad (116)$$

where $q_t^\star$ denotes the "teacher" (true or target) conditional and $q_\theta$ is the Dirichlet model induced by the network. Minimizing (116) yields $s_\theta(x,t) \approx s_t^\star(x)$, which is then plugged into (113).

# C   HPD Impropriety Counter Example

The zero condition for the Hölder projective divergence (HPD) is

$$D_\alpha^H(p:q) = 0 \iff \frac{p(x)^\alpha}{\int p^\alpha} = \frac{q(x)^\beta}{\int q^\beta}, \qquad \beta = \frac{\alpha}{\alpha - 1}, \quad (117)$$

which is equivalent to

$$p(x)^\alpha = c\,q(x)^\beta \quad \text{for some } c > 0. \quad (118)$$

**Corrected counter-example (discrete case).** Let $p = (p_1, \ldots, p_K)$ be any non-uniform probability vector and choose $\alpha > 1$ (thus $\beta = \alpha/(\alpha - 1)$). Define

$$q_i = \frac{p_i^{\alpha-1}}{\sum_{j=1}^K p_j^{\alpha-1}}, \qquad i = 1, \ldots, K. \quad (119)$$

Then $p \neq q$ (unless $p$ is uniform), yet

$$p_i^\alpha = p_i^\alpha, \quad (120)$$

$$q_i^\beta = \frac{p_i^{(\alpha-1)\beta}}{\left(\sum_{j=1}^K p_j^{\alpha-1}\right)^\beta} = \frac{p_i^\alpha}{\left(\sum_{j=1}^K p_j^{\alpha-1}\right)^\beta}. \quad (121)$$

Hence

$$p^\alpha = c\,q^\beta \quad \text{with} \quad c = \left(\sum_{j=1}^K p_j^{\alpha-1}\right)^\beta, \quad (122)$$

and by the equality condition of Hölder's inequality the numerator and denominator of HPD are identical, yielding $D_\alpha^H(p:q) = 0$ although $p \neq q$. This establishes the impropriety (failure of identity of indiscernibles) of HPD as stated in [44]

**Numerical illustration.** For example, let $\alpha = 1.5$ and $p = (0.7, 0.3)$. Then

$$q \propto \big(p_1^{\alpha-1}, p_2^{\alpha-1}\big) = \big(0.7^{0.5},\, 0.3^{0.5}\big),$$

and $D_\alpha^H(p:q)$ evaluates to 0 up to machine precision.

# D  Aditional Comparison

As shown in Table 3, PHK-Dir represents a 12 % improvement over DDPM's 3.17 [2] When B=128 and NFE=1000. As shown in Table 4, PHK-Dir represents 18.98 FID on the CVUSA[66] dataset.

Table 3: FID scores for DDPM and PHK-optimized Dirichlet Diffusion on the CIFAR-10 image dataset with varying NFEs under several different mini-batch size $B$.

| Model | B | NFE=20 | 50 | 200 | 500 | 1000 | 2000 |
|---|---|---|---|---|---|---|---|
| PHK-Dir | 512 | 15.92 | 5.87 | 3.20 | 2.92 | 2.84 | 2.82 |
| PHK-Dir | 288 | 16.53 | 6.15 | 3.29 | 2.99 | 2.91 | 2.81 |
| PHK-Dir | 128 | 16.85 | 6.14 | 3.25 | 2.83 | 2.79 | 2.78 |
| DDPM | 128 | - | - | - | - | 3.17 | - |

Table 4: FID and LPIPS scores for different models on the CVUSA[66] dataset.

| Model | FID ($\downarrow$) | LPIPS ($\downarrow$) |
|---|---|---|
| X-Seq [67] | 161.16 | 0.706 |
| SelGAN [68] | 116.57 | 0.742 |
| CUT [69] | 72.83 | 0.687 |
| CDTE [70] | 122.84 | 0.694 |
| Aerial Diff [71] | 136.18 | 0.855 |
| Instr-p2p [72] | 38.01 | 0.697 |
| ControlNet [73] | 32.45 | 0.650 |
| I2I-Turbo [74] | 77.95 | 0.685 |
| GPG2A [75] | 58.80 | 0.691 |
| BEV [76] | 29.18 | 0.635 |
| PHK-Dir(20% data) | 21.80 | 0.599 |

# E  Compositional Data

As shown in Figure 5, Dirichlet ELBO and Dirichlet PHK achieve better performance in the compositional data. Dirichlet ELBO is better than Gauss ELBO. Dirichlet PHK is better than Dirichlet ELBO.

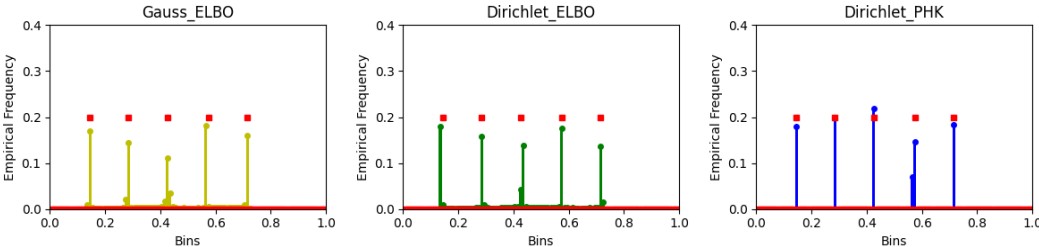

Figure 5: Comparison of true probability mass function (PMF) and empirical PMFs of three methods—Gauss ELBO, Dirichlet ELBO, and Dirichlet PHK-computed. Each true PMF is depicted with red square. Each empirical PMF is computed based on 100k data generated by the model trained after 400k iterations with 200 Numbers of Function Evaluations (NFE).

# F  Hyperparameter Settings

In the experiments, the neural network function ($f_\theta$ in equations) is implemented as a UNet architecture, as is typical in modern diffusion models [2]. we utilize the parameterization of Beta Diffusion [16] as the code base.

We demonstrate the intuition behind the setting of model parameters, including $\eta$, $S_{\text{hift}}$, $S_{\text{cale}}$, $c_0$, $c_1$, $\alpha$, and $\beta$. Given the limitations of computational resources, the meticulous tuning of these model parameters is left for future work.

For the diffusion concentration parameter $\eta$, we set a moderately high value of $\eta = 1000$. A larger value of $\eta$ provides greater discriminative power among different pixel values but requires more discretization steps during sampling, leading to slower training speeds.

We set the linear scaling and shifting parameters as $S_{\text{cale}} = 0.39$ and $S_{\text{hift}} = 0.60$, and the sigmoid-based schedule parameters as $c_0 = 10$ and $c_1 = -13$. These settings are based on the work of Kingma et al. [3] and Zhou et al. [16].

We set the shape parameters of the Proper Hölder Divergence as $\alpha = 2$ and $\beta = \frac{\alpha}{\alpha-1} = 2$. This choice is motivated by an intuitive understanding of the Hölder Divergence, where a smaller $\alpha$ would lead to the loss of symmetry.

As shown in Table 7, the training-time cost of KLUB is substantially more expensive. When all other conditions are identical and $B = 512$, KLUB incurs approximately a 15 % additional training-time cost compared with PHD.

Table 5: FID scores for negative ELBO, KLUB, PHD and PHK-optimized Dirichlet Diffusion on the CIFAR-10 image dataset with varying NFEs under several different combinations of concentration parameter $\eta$ and mini-batch size $B$.

| Loss | $\eta \times 10^{-4}$ | $B$ | NFE = 20 | 50 | 200 | 500 | 1000 | 2000 |
|------|------|------|------|------|------|------|------|------|
| −ELBO | 1 | 512 | 15.81 | 6.53 | 4.49 | 4.33 | 4.28 | 4.31 |
| −ELBO | 1 | 288 | 16.02 | 6.65 | 4.68 | 4.42 | 4.42 | 4.39 |
| KLUB | 1 | 512 | 16.62 | 6.16 | 3.46 | 3.39 | 3.32 | 3.25 |
| KLUB | 1 | 288 | 16.35 | 6.23 | 3.58 | 3.46 | 3.40 | 3.21 |
| PHD | 1 | 512 | 16.84 | 6.48 | 3.85 | 3.72 | 3.52 | 3.51 |
| PHD | 1 | 288 | 16.95 | 6.93 | 4.43 | 4.23 | 4.19 | 4.21 |
| PHK | 1 | 512 | 15.92 | 5.87 | 3.20 | 2.92 | 2.84 | 2.82 |
| PHK | 0.5 | 512 | 17.08 | 6.60 | 3.55 | 3.09 | 3.10 | 3.14 |
| PHK | 0.1 | 512 | 20.67 | 9.67 | 5.66 | 4.54 | 4.58 | 4.53 |
| PHK | 1 | 288 | 16.53 | 6.15 | 3.29 | 2.99 | 2.91 | 2.81 |
| PHK | 0.5 | 288 | 16.94 | 6.71 | 3.61 | 3.09 | 3.03 | 3.02 |
| PHK | 0.1 | 288 | 20.33 | 9.56 | 5.71 | 4.77 | 4.75 | 4.72 |
| PHK | 1 | 128 | 16.85 | 6.14 | 3.25 | 2.83 | 2.79 | **2.78** |
| PHK | 0.5 | 128 | 16.32 | 6.40 | 3.57 | 3.23 | 3.09 | 3.06 |
| PHK | 0.1 | 128 | 20.22 | 9.73 | 6.00 | 5.00 | 4.93 | 5.00 |

Table 6: Frechet Inception Distance (FID) scores for PHK-optimized Dirichlet Diffusion on the unconditional CIFAR-10 dataset with varying Numbers of Function Evaluations (NFE) under several different combinations of the key loss weight coefficients $\delta$ and $\epsilon$ with $B = 288$.

| $\delta$ | $\epsilon$ | NFE=20 | 50 | 200 | 500 | 1000 | 2000 |
|------|------|------|------|------|------|------|------|
| $\delta$=0.5 | $\epsilon$=-0.1 | 16.62 | 6.38 | 3.78 | 3.56 | 3.42 | 3.35 |
| $\delta$=0.5 | $\epsilon$=0 | 16.14 | 6.17 | 3.66 | 3.43 | 3.34 | 3.29 |
| $\delta$=0.5 | $\epsilon$=0.1 | 16.71 | 6.16 | 3.25 | 2.92 | 2.82 | 2.84 |
| $\delta$=0.5 | $\epsilon$=0.15 | 16.74 | 6.06 | 3.20 | 3.00 | 2.86 | 2.78 |
| $\delta$=0.5 | $\epsilon$=0.2 | 16.53 | 6.15 | 3.29 | 2.99 | 2.91 | 2.81 |
| $\delta$=0.5 | $\epsilon$=0.25 | 16.58 | 6.25 | 3.47 | 3.12 | 2.98 | 2.93 |
| $\delta$=0.5 | $\epsilon$=0.5 | 16.07 | 6.23 | 3.61 | 3.39 | 3.35 | 3.31 |

## G   Training Details

In the course of our model training, we employed the Adam optimization algorithm with specific parameter settings: the learning rate $lr = 5 \times 10^{-4}$, the exponential decay rate for the first moment estimates $\beta_1 = 0.9$, the exponential decay rate for the second moment estimates $\beta_2 = 0.999$, and the numerical stability parameter $\epsilon = 1 \times 10^{-8}$.

For the purpose of training the model, we utilized 200 million images. Specifically, we leveraged four Nvidia L40s GPUs for computation, and with a batch size $B = 512$, processing 1000 images of size $32 \times 32 \times 3$ required approximately 1.16 seconds. The processing of 200 million CIFAR-10 images took approximately 64 hours.

During the training process, we saved a checkpoint every 25,000 steps and selected the model with the best performance based on the Fréchet Inception Distance (FID) scores obtained from these checkpoints. We assessed the FID scores using 50,000 samples, a practice consistent with previous research endeavors.

Table 7: The training time cost of the KLUB, PHD, and PHK-optimized Dirichlet Diffusion models on the unconditional CIFAR-10 dataset across two mini-batch sizes $B$ with four Nvidia L40s GPUs.

| $B$ | KLUB | PHD | PHK |
|-----|------|-----|-----|
| 512 | 70h  | 61h | 64h |
| 288 | 73h  | 64h | 68h |

Table 8: The training time cost of the PHK-optimized Dirichlet Diffusion on the CIFAR10-32×32, AFHQ-64×64, FFHQ-64×64 datasets across two mini-batch sizes $B$ with four Nvidia L40s GPUs.

| B | CIFAR10-32×32 | AFHQ-64×64 | FFHQ-64×64 |
|---|---------------|------------|------------|
| 512 | 64 hours | 10 days | 11 days |
| 288 | 68 hours | 11 days | 13 days |

Table 9: The Sampling time cost of the PHK-optimized Dirichlet Diffusion on the unconditional CIFAR10 at varying Numbers of Function Evaluations (NFE) with two or four Nvidia L40s GPUs.

| Nvidia L40s GPUs | NFE = 20 | 50 | 200 | 500 | 1000 | 2000 |
|------------------|----------|-----|------|------|------|------|
| 2 | 11m | 21m | 1h 13m | 3h 2m | 6h 4m | 12h 7m |
| 4 | 7m | 10m | 37m | 1h 33m | 3h 5m | 6h 9m |

# H   Broader Impacts

This study's effective diffusion model is capable of generating images required for practical applications and further developing based on this foundational model, which will have a positive social impact on related fields of work. Technologies such as virtual try-on and video generation, formed based on diffusion models, provide more convenient development space for people's social lives.

This research may also raise concerns about the potential negative social impacts when training on maliciously curated image datasets, which is a shared challenge in the field of generative models. It is crucial to consider how to use these models responsibly to improve society, prepare comprehensive contingency plans, and mitigate any potential adverse consequences.

# I   Uncurated Samples

Figures 6, 7, 8 9 show more uncurated samples from our trained models on CIFAR-10, FFHQ, AFHQ and CVUSA.Through these extensive results, the model's capacity to generate diverse and high-fidelity images without the need for manual selection or management can be clearly elucidated. This feature not only underscores the model's efficacy in image generation tasks but also further corroborates its generalization and adaptability on complex datasets.

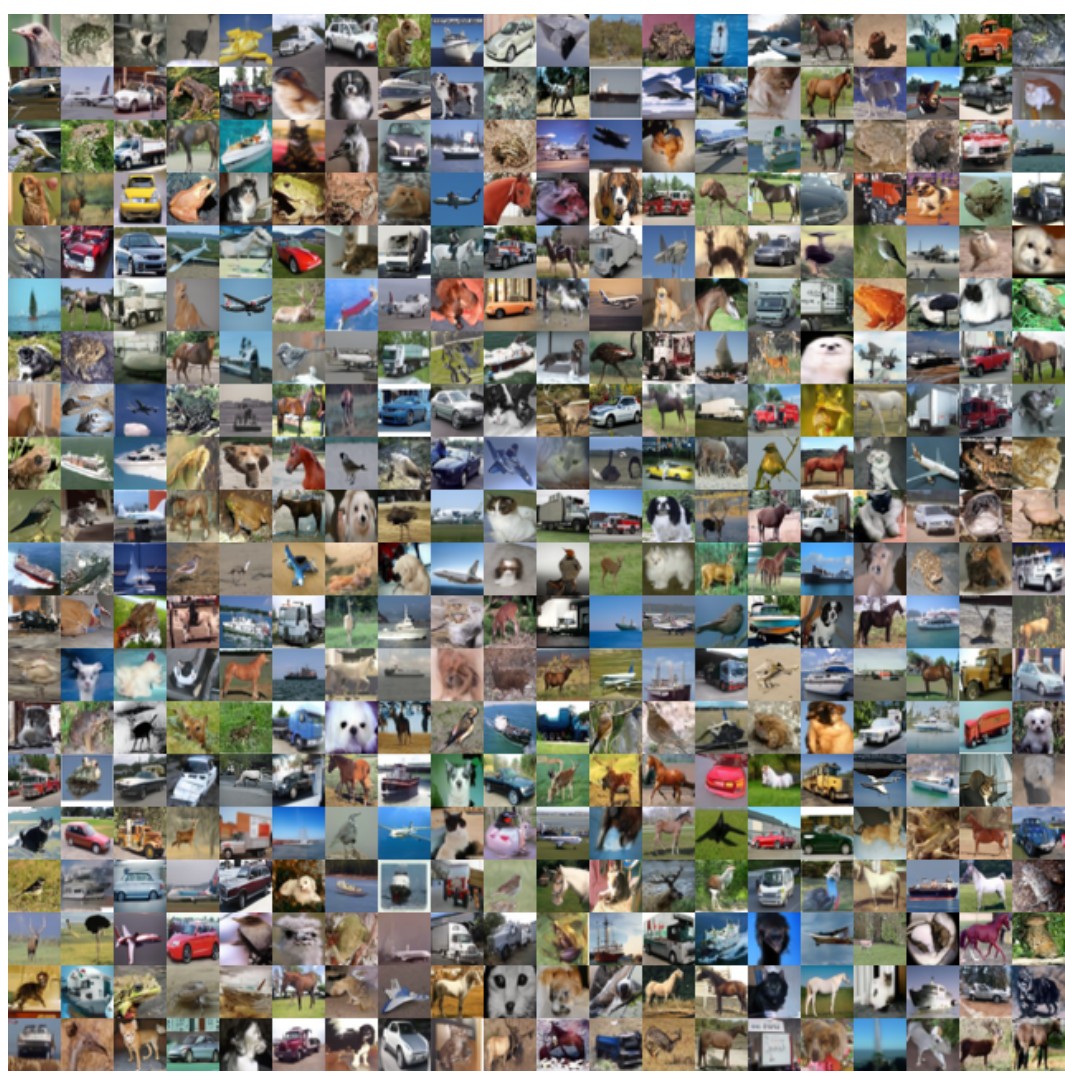

Figure 6: Additional uncurated samples generated by our model trained on CIFAR-10 dataset.

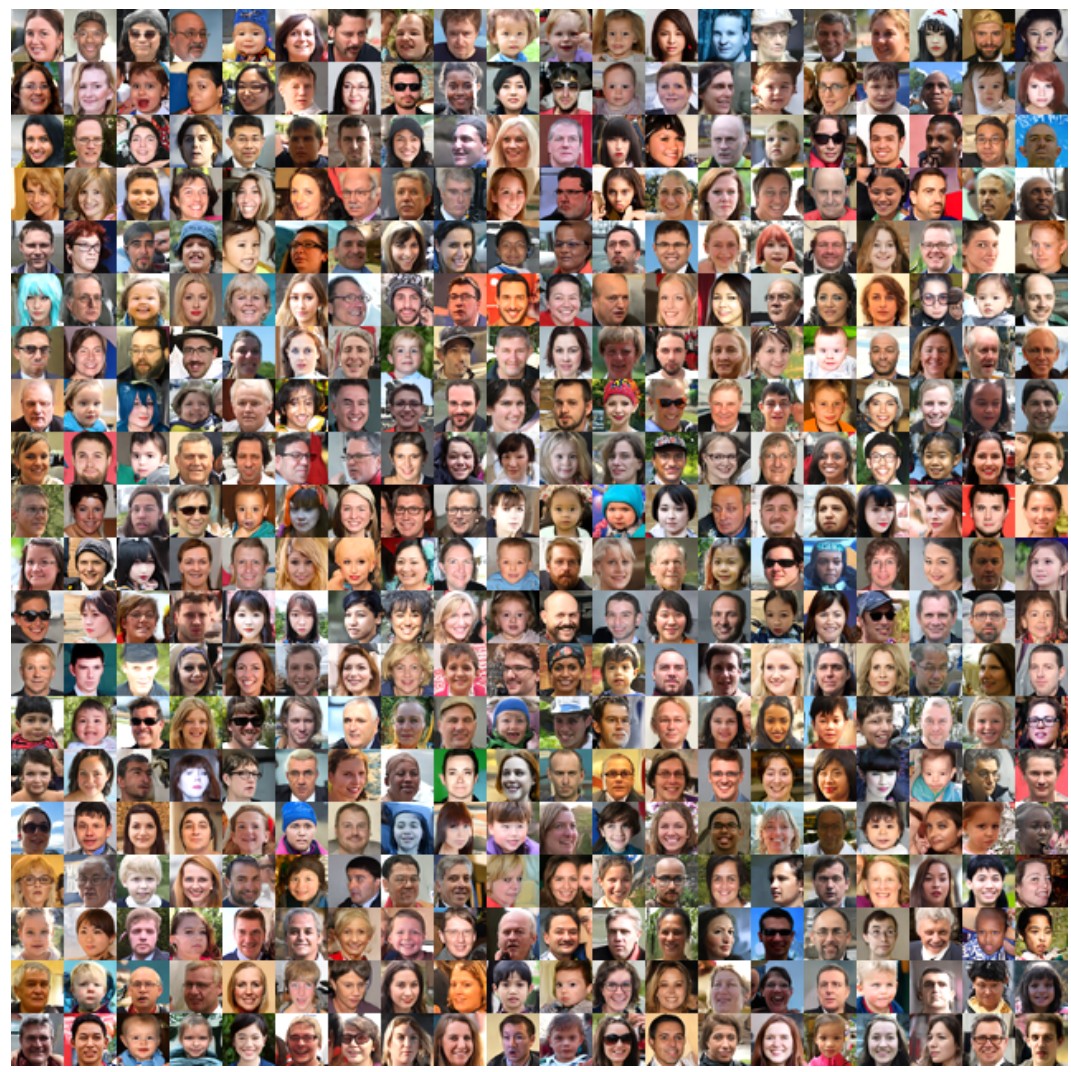

Figure 7: Additional uncurated samples generated by our model trained on FFHQ dataset.

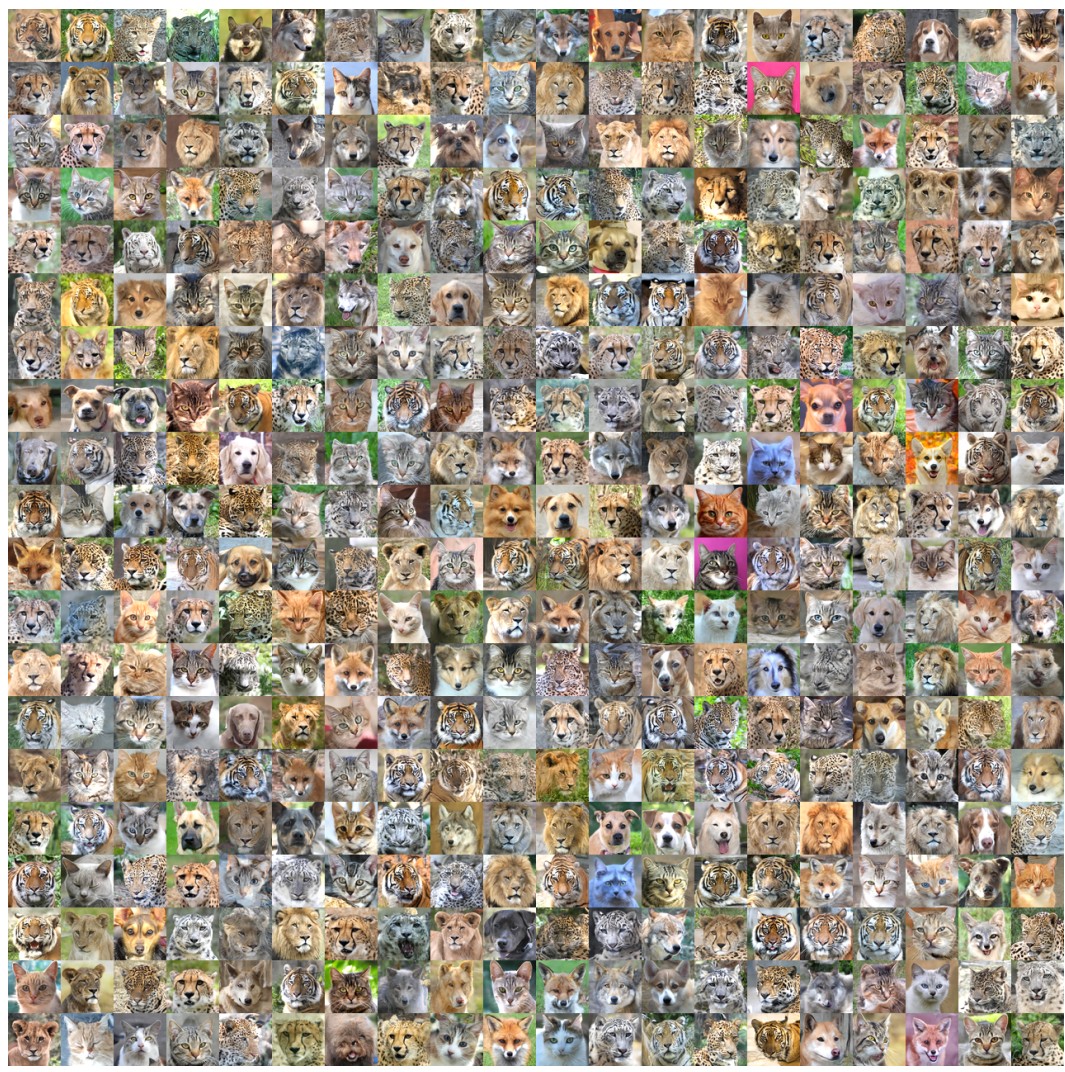

Figure 8: Additional uncurated samples generated by our model trained on AFHQ dataset.

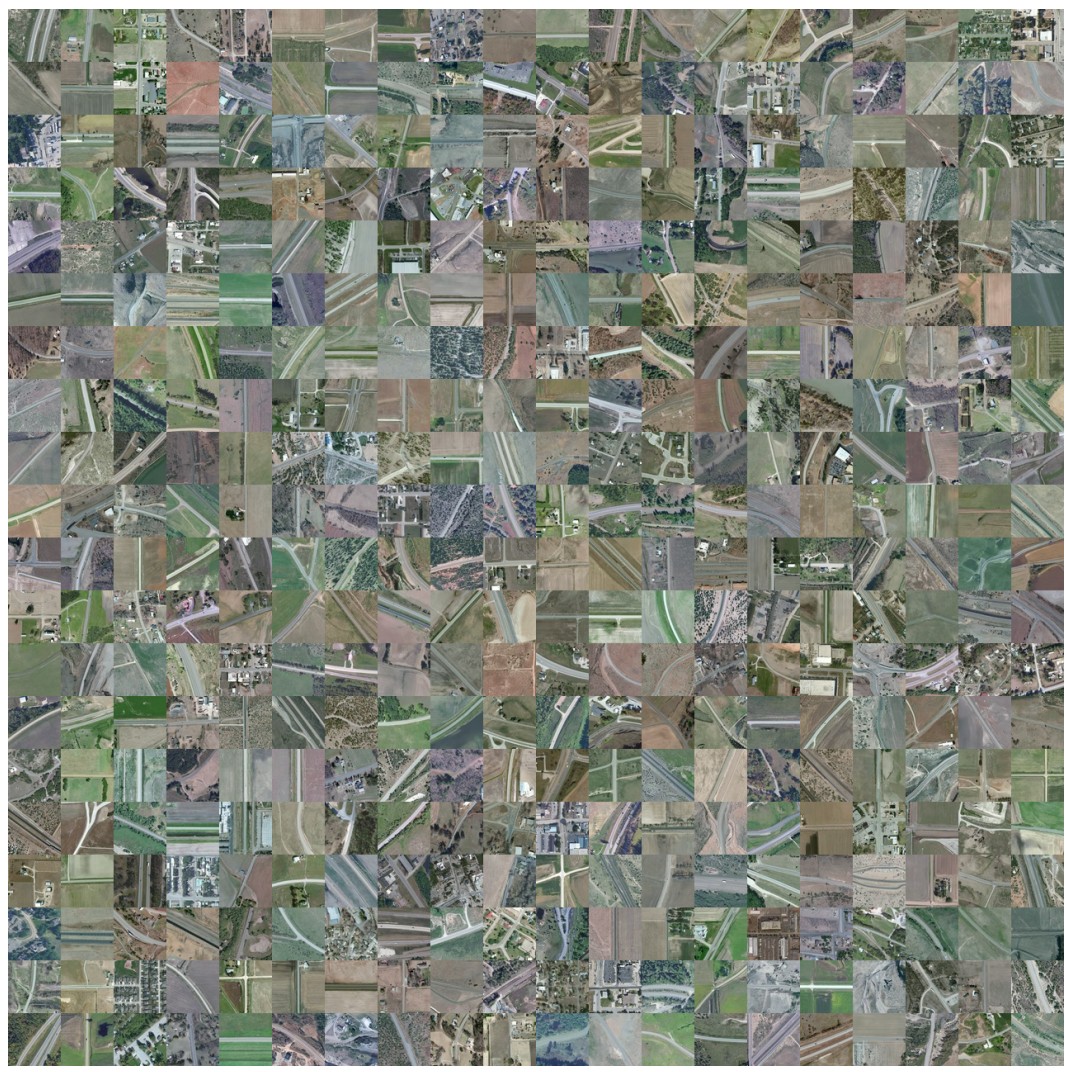

Figure 9: Additional uncurated samples generated by our model trained on CVUSA dataset.

