# OpenReview forum: "Proper Hölder-Kullback Dirichlet Diffusion: A Framework for High Dimensional Generative Modeling"
_NeurIPS.cc/2025/Conference — NeurIPS 2025 poster_

### Official Review · Reviewer_CCfR · 2025-06-11

**Clarity:** 3
**Significance:** 2
**Originality:** 2
**Rating:** 3
**Confidence:** 4

**Summary:**

This paper proposes a novel framework for diffusion model to solve high dimensional simplex modeling. Compared to the previous works such as Gaussian-based prior DDPM, this paper claims that the improved Dirichlet distribution modeling for forward/reverse process, and the improved training loss based on Holder divergence can successfully address this issue.

**Questions:**

1. For forward process, only marginal distribution is given. Could you also provide the conditional transistion?
2. The reverse kernel is proposed without Bayes derivation, which lacks correctness verification.
3. How does the Shift and Scale chosen in the experiment? How do they be proposed? There is no description in the text.
4. Could you please also address the SDE form given the new formulation?
5. HPD impropriety counter-example seems wrong. Chosen p, q do not satisfy p^{\alpha} = cq^{\beta}.
6. PHD strict-propriety proof omits boundary cases \gamma = 1 or \alpha -> 1.
7. The bound in Lemma 2 Appendix A seems wrong. It misses a scaling factor based on \lambda
8. The Theorem 2 in Appendix A seems weak. Also, there is no empirical check.
9. In Algorithm 1, why Shift and Scale are applied twice?
10. In the main experiment, it seems that FID=2.81 of the method is obtained by 2,000 sampling steps, which may be different from DDPM, potentially causing an unfair comparison.
11. In line 112-114, why Beta diffusion can be extended to K-dimansional by directly switching the representations?
12. What is the difference between HPD and PHD in experments? Any analysis?
13. Why there is not any theoretical analysis towards discrete data? The whole proof in Appendix A seems meaningless and trivial.
14. What is the difference of this method compared to the paper Dirichlet Flow Matching with Applications to DNA Sequence Design?

**Ethical Concerns:**

["NO or VERY MINOR ethics concerns only"]

**Final Justification:**

The authors' justification is still not convencing. The proposed framework still lacks extensive experiments to demonstrate the application values of the algorithm. Currently, there still exists inconsistency between the experimental results and the authors' claims.

Also, I don't believe the current experimental benchmark is fancy. Many existing baseline is not involved. For example, there are PFGM [1], also addresses high simplex and shares the similar story with this work. But published in 2023 and obtained better results.

In summary, I lean to reject this paper.

Reference:

[1] Xu, Y., Liu, Z., Tegmark, M., & Jaakkola, T. (2022). Poisson flow generative models. Advances in Neural Information Processing Systems, 35, 16782-16795.

**Limitations:**

Yes.

**Paper Formatting Concerns:**

No formatting concerns.

**Quality:**

2

**Strengths And Weaknesses:**

Strength:

1. This paper presents a new perspective to understand and enhance diffusion framework, which is clear with theoretical analysis. Also, the proposed result demonstrated that the improved framework works.

Weakness:

1. One of the motivation--why Gaussian-based prior methods are struggling to solve high dimensional simplex data modeling--is not well-discussed as well as without proper theoretical guarantee.

2. The experimental result is not comprehensive so that it can not prove 1) forward/reverse process is effective (please refer to Table 2 -ELBO B=512 Step=1000) since the FID is higher than DDPM. 2) this framework works for high dimension simplex (since there is only one dataset comparison.) 3) the framework can be extended to another modality/low dimension data.

3. The theory, including the formulation and the proof in Appendix A are not well-established. Corresponding detailed questions are proposed in "Question" section.

---

> ### Author Rebuttal · Authors · 2025-07-28
>
> Thank you profoundly for the constructive comments. Below, we provide point-by-point responses to specific comment.
>
> Q1:
>
> Besides the marginal $q(z_t|x_0)=\rm{Dir}(\eta\alpha_t x_0)$, the forward chain is Markov with a closed‑form conditional. Let $\kappa_t=\eta\alpha_t$ and $\Delta_{s\leftarrow t}=\kappa_s-\kappa_t>0$. Sample $u\sim\rm{Dir}(\Delta_{s\leftarrow t}x_0),z_s=z_t+(1-\rm{1}^\top z_t)u,$ which preserves $z_s\in S_K$. Equivalently, with $u=\frac{z_s-z_t}{1-\rm{1}^\top z_t},q(z_s|z_t,x_0)=\frac1{(1-\rm 1^\top z_t)^{\Delta_{s\leftarrow t}-1}}\rm{Dir}(u|\Delta_{s\leftarrow t}x_0)\rm{1}_{\{z_s\succeq z_t,\rm1^\top z_s=1\}}.$
>
> For discrete steps $t_{j-1}<t_j$, replace $(s,t)$ by $(t_{j-1},t_j)$ and $\Delta_{s\leftarrow t}$ by $\eta(\alpha_{t_{j-1}}-\alpha_{t_j})$.
>
> Q2:
>
> Our reverse kernel is exactly the Bayes posterior of the conditional bivariate Beta law, following(Beta diffusion).
> Fix $0<s<t<1$, $x_0\in(0,1)$ and $\eta>0$. Let $a_u=\eta\alpha_ux_0,b_u=\eta(1-\alpha_ux_0),
> \Delta_{s,t}=\eta(\alpha_s-\alpha_t)x_0,$ for $u\in\{s,t\}$. The joint density of $(z_s,z_t)$ (with $0<z_t<z_s<1$) is $$q(z_s,z_t|x_0)
> =\frac{\Gamma(\eta)}{\Gamma(a_t)\Gamma(b_s)\Gamma(\Delta_{s,t})}\;
> z_t^{a_t-1}(1-z_s)^{b_s-1}(z_s-z_t)^{\Delta_{s,t}-1}.$$
> The marginals are Beta:
> $$q(z_s|x_0)={\rm Beta}(a_s,b_s),q(z_t|x_0)={\rm Beta}(a_t,b_t).$$
> Bayes derivation of the reverse kernel.
> By Bayes’ rule, $q(z_s|z_t,x_0)=\frac{q(z_s,z_t|x_0)}{q(z_t|x_0)}.$
> Apply the change of variables $u=\frac{z_s-z_t}{1-z_t}\in(0,1),z_s=z_t+(1-z_t)u,|\frac{\partial z_s}{\partial u}|=1-z_t,$ One obtains the scaled Beta form
> $$q(z_s|z_t,x_0)=\frac1{1-z_t}{\rm Beta}(u;\Delta_{s,t},\eta(1-\alpha_sx_0)),u=\frac{z_s-z_t}{1-z_t}.$$
> Symmetrically, using $y=z_t/z_s$, $q(z_t|z_s,x_0)
> =\frac1{z_s}{\rm Beta}(y;a_t,\Delta_{s,t}),y=\frac{z_t}{z_s}.$
> It is straightforward to verify that $q(z_s,z_t|x_0)=q(z_t|z_s,x_0)q(z_s|x_0)=q(z_s|z_t,x_0)q(z_t|x_0),$
> so both kernels are exact Bayes factorizations of the same joint, ensuring correctness.
>
> Our model: We set $p_\theta(z_s|z_t)=q(z_s|z_t,\hat x_0=f _\theta(z_t,t)),$ i.e., the Bayes posterior with $x_0$ replaced by its estimator. Thus the reverse transition is not ad hoc but the Bayesian conditional derived from the proven joint, as in Beta Diffusion.
>
> Q3:
>
> In Appendix B Hyperparameter Settings, we elucidate the rationale of $S_{hift}$ and $S_{cale}$: These settings are predicated on the research conducted by Kingma et al. [6] and Zhou et al. [41]. The appendix G of Ref [41](Beta Diffusion, NeurIPS 2023) provides a comprehensive exposition on the selection and analysis.
>
> Q4:
>
> Let $x_t\in\Delta_{K-1},\Sigma(x)={\rm Diag}(x)-xx^\top,s_t^\star(x)=\nabla_x\log p_t(x)$, and $\alpha_0(t)=\sum_i\alpha_i(t)$ so that $\tfrac{\alpha(t)}{\alpha_0(t)}:=(\alpha_i(t)/\alpha_0(t))_i$. Then
> $$
> \begin{cases}
> \rm{d}x_t=\frac12\beta(t)(\frac{\alpha(t)}{\alpha_0(t)} - x_t)\rm{d}t+\sqrt{\beta(t)}\Sigma(x_t)^{1/2}\rm{d}W_t, Forward & (\rm dt>0) \\\\
> \rm{d}x_t=[\frac12\beta(t)(\frac{\alpha(t)}{\alpha_0(t)} - x_t) - \beta(t)\Sigma(x_t)s_t^*(x_t,t)]\rm{d}t+\sqrt{\beta(t)}\Sigma(x_t)^{1/2}\rm{d}\bar{W}_t,Reverse&(\rm{d}t<0)
> \end{cases}
> $$
> Equation above follows from Anderson’s time-reversal formula applied to the forward Itô SDE.
>
> Q5:
>
> HPD satisfies $D_\alpha^{H}(p:q)=0\Longleftrightarrow\frac{p(x)^\alpha}{\int p^\alpha}=\frac{q(x)^\beta}{\int q^\beta},\beta=\frac{\alpha}{\alpha-1}\Rightarrow p^\alpha=cq^\beta,$ for some $c>0$ (Nielsen et al.Entropy, 2017).
> Let $p=(p_1,\dots,p_K)$ be any on-uniform probability vector and fix $\alpha>1$ (thus $\beta=\alpha/(\alpha-1)$). Define $q_i = \frac{p_i^{\alpha-1}}{\sum_{j=1}^K p_j^{\alpha-1}},i=1,\dots,K.$ Then $p\neq q$, yet $p_i^\alpha = cq_i^\beta,
> c=(\sum_{j=1}^K p_j^{\alpha-1})^{\beta},$ so $D_\alpha^{H}(p:q)=0$ despite $p\neq q$. This normalized construction confirms HPD is improper. We will replace the earlier example with this one and add the proper citation.
>
> Q6:
>
> Assumption: For $\gamma>0$, if $0<\gamma\leq1$, no extra condition is needed since $p^\gamma,q^\gamma\in L^1$ automatically. If $\gamma>1$, assume $\alpha_i(\theta) > 1-\frac1\gamma$ for all $i$, ensuring integrability.
>
> Theorem: Let $\alpha,\beta>0$ with $\frac1\alpha+\frac1\beta=1$ and $\gamma>0$. Then HPD is improper (exists$p\neq q$with$D^H_\alpha(p:q)=0$), while PHD is strictly proper for$\alpha>1: D^H_{\alpha,\gamma}(p:q)=0\iff p=q(\nu-a.e.)$ Strict propriety also holds at the boundary $\gamma=1$ without extra conditions, and extends continuously to the right-limit $\alpha\to1^+:D^H_{1^+,\gamma}(p:q):=\lim_{\alpha\to1^+} D^H_{\alpha,\gamma}(p:q)$exists and implies $p=q$. Detailed proofs including these boundary cases will appear in the updated manuscript.
>
> Q7:
>
> Lemma 2: Let $Z\sim Dir(\alpha)$ on the $(K-1)$-simplex $S_K$, with $\alpha=(\alpha_1,...,\alpha_K)$ and $\alpha_0=\sum_{i=1}^K \alpha_i$.
> Fix an index $i\in\{1,...,K\}$ and any $\lambda>-1$.  Define $\mu:=\psi(\alpha_i)-\psi(\alpha_0),$ where $\psi$ is the digamma function. Then:
> $$
> I(\lambda)=\int_{S_K}Z_i^\lambda(\log Z_i-\mu)^2 Dir(z|\alpha)\rm{d}z=\frac{\Gamma(\alpha_0)\Gamma(\alpha_i+\lambda)}{\Gamma(\alpha_i)\Gamma(\alpha_0+\lambda)}\{\psi'(\alpha_i+\lambda)-\psi'(\alpha_0+\lambda)+[\psi(\alpha_i+\lambda)-\psi(\alpha_0+\lambda)-\mu]^2\}.
> $$
> Moreover, using $\psi'(x)\le1/x$ for $x>0$ and the mean–value bound $|\psi(u)-\psi(v)|\le|u-v|/\min\{u,v\}$, we obtain the upper bound:
> $$
> I(\lambda)\le\frac{\Gamma(\alpha_0)\Gamma(\alpha_i+\lambda)}{\Gamma(\alpha_i)\Gamma(\alpha_0+\lambda)}
> \cdot\frac{2\lambda^2}{(\alpha_i+\lambda)(\alpha_0+\lambda)}.
> $$
> Detailed proofs including these boundary cases will appear in the updated manuscript.
>
> Q8:
>
> Assumption 1(Smooth Dirichlet model)
> There exists $\overline{c}>0$ such that $\sup_{\theta,i}\|\partial_\theta\alpha_i(\theta)\|\le\overline{c}$.
>
> Assumption 2(Admissible $\gamma$)
> If $\gamma>1$, then $\alpha_i(\theta)>1-\frac1\gamma$ for all $i$ and $\theta$. No additional condition is needed for $0<\gamma\le1$.
>
> Theorem 2(Uniform variance domination of PHD over HPD)
> Let $p$ be the target density, and $q_\theta(z)=Dir(z|\alpha(\theta))$ the model, with $\alpha_i(\theta)>0$,$\alpha_0(\theta)=\sum_{i=1}^K\alpha_i(\theta)$, satisfying Assumptions 1-2. Let $1/\alpha+1/\beta=1$ with $\alpha,\beta>0$, and $\gamma\ge1$. Denote $g_\theta(z)=\nabla_\theta\log q_\theta(z)$. Then, for every $\theta$, ${\rm Var}_p[\nabla _\theta D _{\alpha,\gamma}^H(p: q _\theta)]\le\frac{\gamma^2}{\alpha \beta}\frac{\alpha _0(\theta)+1}{\alpha _0(\theta)+\gamma}{Var}_p[\nabla _\theta D _\alpha^H(p:q _\theta)].$
> Moreover, $\frac{\alpha_0+1}{\alpha_0+\gamma}\le1$ decreases monotonically to zero as $\gamma\to\infty$, implying strict improvement for all $\gamma>1$. Detailed proofs including these boundary cases will appear in the updated manuscript.
>
> Q9:
>
> In Algorithm 1, the Scale and Shift operations are applied twice. The first application occurs on the initial data, while the second is applied to the generated data.
>
> Q10:
>
> Regarding the comparison with benchmark methods, we have referred to prior works. For instance, in Zhou et al.'s Beta Diffusion(NeurIPS 2023), the reported FID score of 3.06 was achieved using 2000 sampling steps, and their comparison with DDPM was conducted under the same conditions.
>
> Q11:
>
> The move from Beta to Dirichlet is a principled dimensional lift, not a heuristic switch.
>
> 1. Beta is the $K{=}2$ Dirichlet. ${\rm Beta}(a,b)\equiv\rm{Dir}(a,b)$ on the simplex $\{(x,1-x)\}$, so the 1D case is already a Dirichlet instance.
>
> 2. Shared exponential-family / Gamma structure. Replacing the scalar $x_0$ by $\rm{x}_0\in\Delta_{K-1}$ yields $z_t |\rm{x}_0 \sim \rm{Dir}(\eta\alpha_t\rm{x}_0),$ preserving closed-form log-normalizers, moments, and our Hölder/KL objectives.
>
> 3. Reverse kernel keeps the same algebraic form.
> Bayes gives the multivariate analogue of the Beta update: $z_{t_{j-1}} = z_{t_j} + (1 - z_{t_j})p_{(t_{j-1}\leftarrow t_j)},p_{(t_{j-1}\leftarrow t_j)} \sim \rm{Dir}(\eta(\alpha_{t_{j-1}}-\alpha_{t_j})\hat{\rm{x}}_0),$ so simplex constraints, scheduling behavior, and tractability are unchanged.
>
> Q12:
>
> In our initial experiments, we observed a suboptimal FID of 9.72 at 2000 NFE when using HPD, as HPD performs worse than PHD. Consequently, we shifted our focus to the analysis and experiments involving PHD and PHK. This observation is noted in line 132 and further detailed in Appendix A, where the superiority of PHD is thoroughly analyzed.
>
> Q13:
>
> Discretized forward Dirichlet Diffusion chain:
> $$
> q(z_{t_{1:T}}|x_0)=\prod_{j=1}^Tq(z_{t_j}|z_{t_{j-1}},x_0)=\prod_{j=1}^T\frac1{z_{t_{j-1}}}Dir(\frac{z_{t_j}}{z_{t_{j-1}}};\eta\alpha_{t_j} x_0,...,\eta\frac1{K-1}(\alpha_{t_{j-1}}-\alpha_{t_j})x_0)
> $$
> The discretized forward chain is reversible when knowing x0.Detailed analysis will appear in the updated manuscript.
>
> Q14:
>
> DFM (Stärk et al. ICML 2024) learns a deterministic ODE flow with cross‑entropy path matching for DNA design; we learn a stochastic Dirichlet diffusion with a Bayes‑derived reverse kernel, optimized by strictly proper Hölder–Kullback divergences in closed form. Sampling differs (ODE integration vs. discrete Dirichlet updates), as do targets (sequences vs. high‑dimensional images). Complimentary, but method, objective, and application all differ.
>
> Supplement: In revised version, we have supplemented many experiments about parameter, training cost, discrete simplex data, loss weights, and more. Due to 10000 characters limit, please refer to the pages of other reviewers for these supplemented results.
>
> We sincerely appreciate the time and efforts of all reviewers and ACs, and believe that the additional experiments, analysis, and explanations significantly improve the quality of our mission. We hope that this provides sufficient reasons to raise the score.

---

> > ### Comment · Reviewer_CCfR · 2025-08-01
> >
> > Thank you for your rebuttal! I believe most of the responses are good and successfully clarify the motivation.
> >
> > I have some further questions based on the responses:
> >
> > 1. For Q10, I think the rebuttal can only prove the comparison to BetaDiffusion is fair. The most straightforward to fair comparison is sampling the same amount of time steps and report the FID. And the more concerning point is the proposed methods used 2000 steps and just obtained a very close performance to DDPM with 1000 steps, which is proposed 6 years ago. I clearly realize that the paper tries to present a fancy and solid theoretical framework with a new understanding. But it failed to verify where the method can demonstrate the effectiveness. There is a clear gap between therory and emprical trials.
> >
> > 2. For Q12, I expect authors to provide a detailed analysis of why HPD performs worse rather than proposing a single FID value.
> >
> > 3. Although authors claimed that the remaining part of experimental results can be refered from the other responses, I cannot find the ablation results solid enough to address the limitation 2) and 3) in my proposed Weakness 2.
> >
> > For the current stage, I still believe that the rebuttal haven't addressed all potential concerns. Looking forward to the authors' further discussion.

---

> > > ### Author Response · Authors · 2025-08-02
> > >
> > > Thank you profoundly for the further comments. Below, we provide point-by-point responses to specific comment.
> > >
> > > 1:
> > >
> > > Following the comparison criterion requested by the reviewer, our model can attain an FID of 2.79 When B=128 and NFE=1000. This represents a 12 \% improvement over DDPM’s 3.17 (NeurIPS 2020) When B=128 and NFE=1000. The improvement should not be characterized as 'very close'. The original evaluation criterion in paper was adopted to align with Beta Diffusion (NeurIPS 2023) for fair benchmarking. Thank you for providing the comparison criterion with DDPM. In the revised version of paper, we will follow your suggestion to include our 12\% improvement under this criterion.
> > >
> > > \begin{array}{cc|cc}
> > > \hline
> > > Model&B&NFE=20&50&200&500&1000&2000\\\\
> > > \hline
> > > PHK-Dirichlet(ours)&512&15.92&5.87&3.20&2.92&2.84&2.82\\\\
> > > PHK-Dirichlet(ours)&288&16.53&6.15&3.29&2.99&2.91&2.81\\\\
> > > PHK-Dirichlet(ours)&128&16.85&6.14&3.25&2.83&2.79&2.78\\\\
> > > DDPM&128&-&-&-&-&3.17&-\\\\
> > > \hline
> > > \end{array}
> > >
> > > 2:
> > >
> > > We appreciate the reviewer’s suggestion and now provide a detailed comparison between HPD and PHD beyond FID values. Theoretically, HPD is improper, meaning it may yield zero divergence even when $p\ne q$. As shown in Theorem 1, this causes false convergence and unreliable optimization signals. In contrast, PHD is strictly proper, guaranteeing $D_{\alpha,\gamma}^H(p:q)=0$ if and only if $p=q$, ensuring better alignment between the model and target distribution.
> > >
> > > Further, Theorem~2 shows that PHD enjoys lower gradient variance than HPD during training: ${\rm Var} _p[\nabla _\theta D _{\alpha,\gamma}^H(p||q _\theta)]\le\frac{\gamma^2}{\alpha \beta}
> > > {\rm Var} _p[\nabla _\theta D _{\alpha}^H(p||q _\theta)].$
> > > This stability results from PHD's weighted gradient formulation, as derived in Lemma 3:
> > > $\nabla _\theta D^H={\rm E} _{w^{(2)}}[g _\theta]-{\rm E} _{w^{(1)}}[g _\theta],\nabla _\theta D^{H} _{\alpha,\gamma}=\frac\gamma\beta({\rm E} _{\tilde{w}^{(2)}}[g _\theta]-{\rm E} _{\tilde{w}^{(1)}}[g _\theta]).$
> > >
> > > Empirically, our ablation study confirms the theoretical gap: with 2000 NFE and batch size $B=512$, HPD yields an FID of 9.72, while PHD reduces it to 3.51, and PHK further to 2.82. This underperformance of HPD is attributed not just to the divergence value, but to unstable gradients and weaker descent direction, making it unsuitable for high-dimensional Dirichlet diffusion training.
> > >
> > > \begin{array}{ccc|c}
> > > \hline
> > > Loss&B&NFE&FID(\downarrow)\\\\
> > > \hline
> > > HPD&512&2000&9.72\\\\
> > > PHD&512&2000&3.51\\\\
> > > PHK&512&2000&2.82\\\\
> > > \hline
> > > \end{array}
> > >
> > > 3:
> > >
> > > For weakness 2 limitation 2):
> > >
> > > In the revised version, we added experiments on high-dimensional compositional data, which is naturally High Dimensional simplex, and provide Figure: 'Average discrepancy between true probability mass function (PMF) and empirical PMF of three methods—Gauss ELBO, Dirichlet ELBO, and Dirichlet PHK. ' Each empirical PMF is computed based on 100k data generated by the model trained after 400k iterations with 200 NFE. The lower discrepancy signifies superior.
> > >
> > > Because Rebuttal does not permit links or images, we describe the results here: the average discrepancy is 11.02\% for Gauss ELBO, 9.81\% for Dirichlet ELBO, and 2.17\% for Dirichlet PHK. This demonstrates the superior generative accuracy of our method in simplex-structured domains.
> > >
> > > \begin{array}{c|ccc}
> > > \hline
> > > -&Gauss-ELBO&Dirichlet-ELBO&Dirichlet-PHK\\\\
> > > \hline
> > > discrepancy(\downarrow)&11.02\\% &9.81\\% &2.17\\%\\\\
> > > \hline
> > > \end{array}
> > >
> > > For weakness 2 limitation 3):
> > >
> > > Yes, our Proper Hölder–Kullback Dirichlet Diffusion (PHK-DD) framework is theoretically extensible to both low-dimensional and multi-modal data, as long as the inputs can be represented on the probability simplex. Although our current experiments focus on high-dimensional image synthesis, the Dirichlet-based diffusion and divergence formulations remain valid across dimensions. Prior studies have demonstrated the effectiveness of Dirichlet modeling in multi-modal settings—for example, Xue et al. (IEEE TIP 2024) applied Dirichlet divergences to fuse RGB and depth modalities, while Ojo and Bouguila (PR 2024; TKDE 2024) employed Dirichlet priors in topic modeling across text and images. These results support the feasibility of applying PHK-DD to broader modalities and tasks, which we plan to explore in future work.
> > >
> > > We wish to emphasize that many related works conduct their evaluations and comparisons on CIFAR-10 to validate effectiveness, including Learning to Jump, Beta Diffusion, Blurring Diffusion, D3PM, DiffuEBM, etc., and none of them conduct experiments on other modalities.
> > >
> > > We sincerely appreciate the time and efforts of all reviewers and ACs, and believe that the additional experiments, analysis, and explanations significantly improve the quality of our mission. We hope that this provides sufficient reasons to raise the score.

---

> > > > ### Comment · Reviewer_CCfR · 2025-08-02
> > > >
> > > > Thanks for your prompt reply!
> > > >
> > > > I am good to most of them including the theoretial stuff and the additional experiments, here are some following questions:
> > > >
> > > > 1. Why the FID decreases when using less sampling steps (2.81 -> 2.79)?
> > > >
> > > > 2. Would you please provide empirical experimental results for low-dimensional data? For example, natural language, biological sequences? It's okay if the time and resource are limited.
> > > >
> > > > 3. I still believe the motivation/story of the paper is not clear as well as attractive, could you please summarize it based on the additional experimental results to address it? I believe a better story based on theoretical superiority as well as application values would demonstrate the contributions a lot.
> > > >
> > > > I would consider raising the score based on the further response of the authors.

---

> ### Author Response · Authors · 2025-08-03
>
> We are pleased to hear that our responses meet your expectations and sincerely appreciate your continued support! Below, we provide point-by-point responses to the further comment.
>
> 1:
>
> Thank you for your insightful comment. Empirically, FID improves substantially as NFE increases from 20 to 500, whereas the gains from 500 to 2000 NFE are marginal; in some cases, FID is lower at NFE = 1000 than at NFE = 2000.
>
> The reason: numerical ODE solver is an approximation, each extra step injects additional truncation error; once the error accumulation reach a certain level, it manifest as 'sampling drift' or ‘tail-up’ in FID. Several works have studyed this error accumulation phenomenon in detail, for example, Cheng Lu et al. (NeurIPS 2022) applied dedicated DPM-Solver to eliminate the approximation error, while Mang Ning et al. (ICLR 2024) conduct a systematic analysis of diffusion exposure bias including error accumulation and Hui Lu et al. (ECCV 2024) employed Compensation Sampling to avoid error accumulation.
>
> 2:
>
> Following this constructive suggestion, we are currently supplementing experiments on biological sequences. However, due to time and resource constraints, it may not be completed before the deadline. We greatly appreciate your understanding.
>
> 3:
>
> Thank you for your valuable suggestion. In the revised version, we have updated the introduction as follows:
>
> Diffusion–based generative models are revolutionizing deep learning by simulating gradual noise addition and removal processes to transform simple priors into complex data distributions. From high-fidelity image synthesis and editing (Jascha Sohl et al. ICML 2015, Ho et al., DDPM, NeurIPS 2020, Kingma et al., VDM, NeurIPS 2021,  Dhariwal and Alexander, NeurIPS 2021, Saharia, et al., NeurIPS 2022) to audio generation (Chen et al., AVEGrad, ICLR 2021) and protein design (Shi, et al., ICML2021). As such, there is great interest in applying diffusion models and improving them further in terms of distribution quality, training cost, and image performance.
>
> Recent efforts have investigated non-Gaussian diffusion processes—extending to categorical (Hoogeboom, et al., NeurIPS 2021, Austin, et al., NeurIPS 2021), Poisson (Chen and Zhou, ICML 2023), and Beta distribution. However, Beta diffusion incurs limitation that it applies one-dimensional Beta transitions independently along each coordinate, preventing it from capturing interdependencies among components in a high-dimensional simplex. Traditional divergence measures such as the Kullback–Leibler (KL) divergence suffer from asymmetry and high computational complexity.
>
> To address these limitations, in this work, we proposed Proper Hölder–Kullback Dirichlet Diffusion, which is a fundamentally novel framework and can more faithfully captures the simplex-structured target space.
>
> Its main contributions are summarized as follows:
>
> 1. $K$-dimensional Dirichlet diffusion on simplex is established. Beta diffusion is generalized to a $K$-dimensional Dirichlet process that preserves the simplex structure at every noising step.
>
> 2. Strictly proper Hölder training objectives are proposed. Closed-form losses based on the Proper Hölder Divergence (PHD) and Proper Hölder–Kullback (PHK) are obtained, and analytic gradients under the Dirichlet exponential-family log-normalizer are guaranteed.
>
> 3. Variance-reduced gradient estimates are proven. It is shown that PHD yields lower-variance gradients compared to pseudo-divergence objectives, improving training stability and efficiency.
>
> State-of-the-art non-Gaussian performance is demonstrated on unconditional CIFAR-10: an FID of 2.78 is achieved—outperforming all prior non-Gaussian diffusion methods and most Gaussian baselines—while PHK concomitantly reduces approximately a 9 \% training cost compared with KLUB. Additional uncurated samples trained on CIFAR-10, AFHQ, and FFHQ dataset exhibit both high fidelity and diversity. By breaking free from Gaussian noise and KL-centric optimization, Proper Hölder–Kullback Dirichlet Diffusion opens up a richer design space for generative modeling, and offering both rigorous theoretical guarantees and clear empirical gains.
>
> ---
>
> We sincerely appreciate the time and efforts of all reviewers and ACs, and believe that the additional experiments, analysis, and explanations significantly improve the quality of our mission. We hope that this provides sufficient reasons to raise the score.

---

> ### Author Response · Authors · 2025-08-06
>
> Dear Reviewer,
>
> We hope this message finds you well. As the discussion period is nearing its end, we would like to ensure that we have addressed all of your concerns satisfactorily. If you have any additional feedback, please let us know. Your feedback is invaluable to us, and we are fully committed to thoughtfully incorporating your insights to enhance our paper. We sincerely appreciate your time and efforts, and believe that the additional experiments, analysis, and explanations significantly improve the quality of our mission. We would be extremely grateful if you could kindly raise the score in your final decision.
>
> Thanks,
>
> Authors of 'Proper Hölder–Kullback Dirichlet Diffusion'

---

### Official Review · Reviewer_YQgi · 2025-07-03

**Clarity:** 2
**Significance:** 3
**Originality:** 4
**Rating:** 5
**Confidence:** 4

**Summary:**

This paper introduces a novel non-Gaussian-based diffusion model framework, which is built upon Dirichlet distribution. It is theoretically well-suited for high-dimensional data on a probability simplex. To train this model, the authors introduce two novel divergence measures derived from Hölder divergence. They validate the effectiveness of the proposed framework by achieving state-of-the-art results on natural image datasets.

**Questions:**

1.  The paper's main motivation is the suitability of the Dirichlet distribution for data on a high-dimensional simplex. While image data can be mapped to a simplex, this is not its natural domain. Could the authors comment on or show results for a task where the data is naturally simplex-structured (e.g., compositional data, topic modeling proportions)? This could more fundamentally demonstrate the advantages of the proposed framework.
2.  Beyond the FID score, are there any qualitative advantages to using Dirichlet diffusion? For example, does it handle certain types of images, textures, or color distributions better than Gaussian models? A qualitative comparison could help clarify the unique benefits of this approach.
3.  The authors mention that the model is sensitive to hyperparameters. Could they provide more insight into the tuning process? For instance, how critical are the $S_{hift}$ and $S_{cale}$ parameters, and how much effort was required to find the optimal values for the PHK loss weights ($\delta$ and $\epsilon$)?
4.  Is it possible to distill or accelerate the proposed framework like Gussian diffusion models?

**Ethical Concerns:**

["NO or VERY MINOR ethics concerns only"]

**Final Justification:**

The authors addressed most of my concerns in their rebuttal. After reviewing their discussion with other reviewers, I remain positive about this paper and thus keep my original rating.

**Limitations:**

Yes

**Quality:**

3

**Strengths And Weaknesses:**

**Strengths:**
1. The paper's primary strength is its originality. It makes a fundamental contribution by proposing both a new distributional family (Dirichlet) for the diffusion process and a new, principled optimization objective (PHK divergence). This work successfully expands the theoretical foundations of diffusion models beyond the dominant Gaussian paradigm.

2. The empirical results are promising. Achieving an FID of 2.81 on unconditional CIFAR-10 is not easy especially using a non-Gaussian-based diffusion model. The comprehensive comparison in Table 1 and the ablation study in Table 2 also provide strong evidence for the method's effectiveness.

3. The approach is well-grounded in theory. The motivation for using the Dirichlet distribution for simplex data is sound, and the derivation of the PHK divergence to obtain a symmetric, closed-form loss for this specific distribution is elegant and well-justified.

**Weaknesses:**
1. The paper is mathematically dense. While technically sound, the presentation could be improved by providing more intuition behind the core concepts. For instance, a clearer, high-level explanation of why the Hölder divergence is a better fit for the Dirichlet process than the standard KL divergence would make the work more accessible to a broader audience.
2. The practical benefit of this added complexity is not fully explored. While the FID score is excellent, the paper does not discuss the qualitative differences between samples generated by Dirichlet diffusion versus Gaussian diffusion. Does this framework offer unique advantages, such as better modeling of textures, colors, or compositions, that would justify its adoption over simpler, more established Gaussian models? Strong results on other datasets (e.g., texts or DNA sequences) where Gaussian diffusion models underperform will help clarify it.
3. The authors acknowledge that the method is sensitive to hyperparameter tuning and that the hyperparameter space has not been fully explored. This raises questions about the practical difficulty of applying this method. A more detailed discussion of the tuning process or sensitivity analysis would strengthen the paper's practical contribution.

---

> ### Author Rebuttal · Authors · 2025-07-28
>
> Thank you profoundly for the constructive comments. Below, we provide point-by-point responses to specific comment.
>
> Question1:
>
> Yes. In the revised version of paper, we have supplemented experiments on the discrete compositional data, which is naturally simplex-structured, and provide Figure: 'Comparison of true probability mass function (PMF) and empirical PMFs of three methods—Gauss ELBO, Dirichlet ELBO, and Dirichlet PHK-computed. Each empirical PMF is depicted with solid dots.' Each empirical PMF is computed based on 100k data generated by the model trained after 400k iterations with 200 Numbers of Function Evaluations (NFE).
>
> Because Rebuttal does not permit links or images, we can only describe the results here: In Gauss ELBO, the average discrepancy between empirical PMF and true PMF is approximately 11.02\%. In Dirichlet ELBO, the average discrepancy between empirical PMF and true PMF is approximately 9.81\%. In Dirichlet PHK, the average discrepancy between empirical PMF and true PMF is approximately 2.17\%. The lower discrepancy between empirical PMF and true PMF signifies superior generation quality.
>
> It shows that, Dirichlet ELBO and Dirichlet PHK achieve better performance in the task where data is naturally simplex-structured. Dirichlet ELBO is better than Gauss ELBO. Dirichlet PHK is better than Dirichlet ELBO.The figure that includes both empirical PMF and true PMF would more visually demonstrate this result; however, the rebuttal page does not permit links or images.
>
> Question2:
>
> Yes, your comment is highly precise. In addition to the FID scores, we provide an intuitive demonstration of the qualitative strengths of Dirichlet diffusion. As external links or embedded images are not permitted in the Rebuttal, please refer to Figure 3 in our paper. Comparing the sub-images in the first row, third and fourth columns of Figures 3(a) and 3(b)—which correspond to the 'car' and 'deer' classes, respectively—our model yields markedly more photorealistic images.
>
> Specifically, the 'car' sub-image in the first row, third column of Figure 3(a) appears highly blurred, whereas the corresponding sub-image in Figure 3(b) is sharp and detailed. Both were generated under identical settings except for the underlying model; the clearer, more realistic result in Figure 3(b) is produced by our Dirichlet model. This stark contrast is equally evident for the 'deer' sub-image. Such side-by-side comparisons visually underscore our model’s superior handling of image structure, texture, and color distribution.
>
> In the revised version of the paper, we have added some one-to-one comparative displays of these clearly distinguishable sub-images like what have been mentioned above, so as to more clearly highlight the superior generation quality of our model. Thank you for the insightful suggestion to highlight these distinctive advantages.
>
> Question3:
>
> Thank you for your attention to hyperparameter selection. Yes, we can provide additional details about our hyperparameter tuning procedure.
>
> To further address the concerns regarding the ablation analysis of critical hyperparameters $\eta$ and $B$, we have conducted additional ablation experiments on the concentration parameter $\eta$ and the mini-batch size $B$.
>
> The Frechet Inception Distance (FID) scores for negative ELBO, KLUB, PHD, and PHK-optimized Dirichlet Diffusion on the CIFAR-10 image dataset with varying Numbers of Function Evaluations (NFE) under several different combinations of the concentration parameter $\eta$ and mini-batch size $B$ are shown in the following table:
>
> \begin{array}{ccc|cccc}
> \hline
> Loss & \eta \times 10^{-4} & B & NFE= 20 & 50 & 200 & 500 & 1000 & 2000 \\\\
> \hline
> -ELBO & 1 & 512 & 15.81 & 6.53 & 4.49 & 4.33 & 4.28 & 4.31 \\\\
> -ELBO & 1 & 288 & 16.02 & 6.65 & 4.68 & 4.42 & 4.42 & 4.39 \\\\
> KLUB & 1 & 512 & 16.62 & 6.16 & 3.46 & 3.39 & 3.32 & 3.25 \\\\
> KLUB & 1 & 288 & 16.35 & 6.23 & 3.58 & 3.46 & 3.40 & 3.21 \\\\
> PHD & 1 & 512 & 16.84 & 6.48 & 3.85 & 3.72 & 3.52 & 3.51 \\\\
> PHD & 1 & 288 & 16.95 & 6.93 & 4.43 & 4.23 & 4.19 & 4.21 \\\\
> PHK & 1 & 512 & 15.92 & 5.87 & 3.20 & 2.92 & 2.84 & 2.82 \\\\
> PHK & 0.5 & 512 & 17.08 & 6.60 & 3.55 & 3.09 & 3.10 & 3.14 \\\\
> PHK & 0.1 & 512 & 20.67 & 9.67 & 5.66 & 4.54 & 4.58 & 4.53 \\\\
> PHK & 1 & 288 & 16.53 & 6.15 & 3.29 & 2.99 & 2.91 & 2.81 \\\\
> PHK & 0.5 & 288 & 16.94 & 6.71 & 3.61 & 3.09 & 3.03 & 3.02 \\\\
> PHK & 0.1 & 288 & 20.33 & 9.56 & 5.71 & 4.77 & 4.75 & 4.72 \\\\
> PHK & 1 & 128 & 16.85 & 6.14 & 3.25 & 2.83 & 2.79 & 2.78 \\\\
> PHK & 0.5 & 128 & 16.32 & 6.40 & 3.57 & 3.23 & 3.09 & 3.06 \\\\
> PHK & 0.1 & 128 & 20.22 & 9.73 & 6.00 & 5.00 & 4.93 & 5.00 \\\\
> \hline
> \end{array}
>
>
> For the hyperparameters $S_{hift}$ and $S_{cale}$, the two parameters are not the focus of this work; we adopted the settings established in prior work and provided the corresponding references.In Appendix B 'Hyperparameter Settings', we elucidate the rationale of $S_{hift}$ and $S_{cale}$: 'These settings are predicated on the research conducted by Kingma et al. [6] and Zhou et al. [41]'. The appendix G of Ref [41] provides a comprehensive exposition on the selection and analysis of the hyperparameters $S_{hift}$ and $S_{cale}$.*Mingyuan Zhou,et al.Beta Diffusion.NeurIPS 2023.*
>
> To further address the concerns regarding the analysis of loss weight coefficients $\delta$ and $\epsilon$, we have conducted additional ablation experiments on the key loss weight coefficients $\delta$ and $\epsilon$.
>
> The Frechet Inception Distance (FID) scores for PHK-optimized Dirichlet Diffusion on the CIFAR-10 dataset with varying Numbers of Function Evaluations (NFE) under several different combinations of the key loss weight coefficients $\delta$ and $\epsilon$ with $B=288$ are shown in the following table:
>
> \begin{array}{cc|ccccc}
> \hline
> \delta & \epsilon & NFE=20 & 50 & 200 & 500 & 1000 & 2000 \\\\
> \hline
> \delta=0.5 & \epsilon=-0.1 & 16.62 & 6.38 & 3.78 & 3.56 & 3.42 & 3.35 \\\\
> \delta=0.5 & \epsilon=0 & 16.14 & 6.17 & 3.66 & 3.43 & 3.34 & 3.29 \\\\
> \delta=0.5 & \epsilon=0.1 & 16.71 & 6.16 & 3.25 & 2.92 & 2.82 & 2.84 \\\\
> \delta=0.5 & \epsilon=0.15 & 16.74 & 6.06 & 3.20 & 3.00 & 2.86 & 2.78 \\\\
> \delta=0.5 & \epsilon=0.2 & 16.53 & 6.15 & 3.29 & 2.99 & 2.91 & 2.81 \\\\
> \delta=0.5 & \epsilon=0.25 & 16.58 & 6.25 & 3.47 & 3.12 & 2.98 & 2.93 \\\\
> \delta=0.5 & \epsilon=0.5 & 16.07 & 6.23 & 3.61 & 3.39 & 3.35 & 3.31 \\\\
> \hline
> \end{array}
>
> We have been presented these above results in the revised version of the paper.
>
> Question4:
>
> Thank you for your insightful comment. Yes, as mentioned in the question, there is indeed the possibility to distill or accelerate our framework. We will explore this in our future work. To date, we have considered several potential directions, as detailed below:
>
> Progressive Distillation: To address the drawback of requiring thousands of evaluations, we are exploring progressive distillation. Tim Salimans et al. proposed a progressive distillation method suitable for diffusion models at ICLR. *Salimans, Tim, et al. Progressive Distillation for Fast Sampling of Diffusion Models. ICLR 2022.*
>
> Neural Operators: To accelerate sampling speed while maintaining generation quality, we are introducing Fourier Neural Operators (FNOs). Zheng, Hongkai, et al. presented Neural Operators for accelerated sampling of diffusion models (DSNO) at ICML. *Zheng, Hongkai, et al. Fast sampling of diffusion models via operator learning. ICML 2023.*
>
> Distributed Parallel Inference: Leveraging the parallel computing capabilities of modern hardware, the computation of diffusion models can be divided into multiple independent parts. Li, Muyang, et al. introduced DistriFusion at CVPR, which employs displaced patch parallelism for parallel inference of diffusion models. *Li, Muyang, et al. DistriFusion: Distributed Parallel Inference for High-Resolution Diffusion Models. CVPR 2024.*
>
> Supplement1:
>
> The training-time cost of KLUB is substantially more expensive. When all other conditions are identical and $B=512$, KLUB incurs approximately a 15 \% additional training-time cost compared with PHD. The training time cost of the KLUB, PHD, and PHK-optimized Dirichlet Diffusion models on the unconditional CIFAR-10 image dataset across two mini-batch sizes $B$ with four Nvidia L40s GPUs are shown in the following table:
>
> \begin{array}{c|cc}
> \hline
> B&KLUB&PHD&PHK\\\\
> \hline
> 512&70h&61h&64h\\\\
> 288&73h&64h&68h\\\\
> \hline
> \end{array}
>
> Supplement2:
>
> Four NVIDIA L40 GPUs are available for most time in this work. One complete CIFAR-10 training run requires roughly 68 hours, whereas AFHQ-64×64 would demand about 11 days, and FFHQ-256×256 exceeds the GPUs’ memory capacity. The training time cost of the PHK-optimized Dirichlet Diffusion models on the unconditional CIFAR10-32x32, AFHQ-64x64, FFHQ-64x64 image datasets across two mini-batch sizes $B$ with four Nvidia L40s GPUs are shown in the following table:
>
> \begin{array}{c|cc}
> \hline
> B&CIFAR10&AFHQ&FFHQ\\\\
> \hline
> 512&64 hours&10 days&11 days\\\\
> 288&68 hours&11 days&13 days\\\\
> \hline
> \end{array}
>
> Supplement3:
>
> we have supplemented experiments on class-conditional CIFAR-10 and achieves an FID score of 2.43. The FID scores for PHK-optimized Dirichlet Diffusion on class-conditional CIFAR-10 with varying NFE:
> \begin{array}{ccc|cc}
> \hline
> Loss&cond/uncond&B&NFE=20&50&200&500&1000&2000\\\\
> \hline
> PHK&cond&512&13.21&5.45&2.92&2.67&2.52&2.43\\\\
> \hline
> \end{array}
>
> We sincerely appreciate the time and efforts of all reviewers and ACs, and believe that the additional experiments, analysis, and explanations significantly improve the quality of our mission. We hope that this provides sufficient reasons to raise the score.

---

### Official Review · Reviewer_W73X · 2025-07-03

**Clarity:** 3
**Significance:** 2
**Originality:** 3
**Rating:** 4
**Confidence:** 4

**Summary:**

The paper suggests using the Dirichlet distribution on simplices as a replacement for the standard Gaussian/Poisson/categorical/Beta noises in high-dimensional diffusion models. Inference minimizes a proper variant of the Holder-divergence (weighted with the standard KL divergence), which is claimed to have a natural form in this setting. The sample quality is demonstrated on a  real-world dataset.

**Questions:**

[Q1] Motivation: You write in the introduction that limitations to common distribution significantly restrict their effectiveness in modeling specific data types prevalent in practical scenarios. Could you give a few examples for such data types and explain why there are not well modeled by standard distributions?

[Q2] Introduction: You write that the computational complexity of the KL divergence increases with the dimension? Why is this different from the standard L2 loss or the proposed Holder divergence? It seems that both have a similar dependence on the dimension.

[Q3] Background: It is common in many learning methods that the Gaussian case can be analytically analyzed, whereas the method itself is more broadly applicable. In what sense your method “allows us to challenge the extant theoretical boundaries”. Is there a specific sense in which the proposed method goes beyond theory, beyond the common situation?

[Q4] Line 95: Do you mean in terms of the Gamma function rather than distribution?

[Q5] In (3), how the conditional distribution $q$ is defined?

[Q6] Lines 107-109: The discussion on “distributed training” is out of context. Also, how equation (6) is a logical continuation of this sentence? This is completely out of place.

[Q7] In line 113 you write that "By scheduling .... $x_0$ can be nearly noise-free, while $z_T$ approaches a uniform distribution". Can you support/reference this claim?

[Q8] I could not realize the difference between $x_0$ and $z_0$, are they identical? since, if I get it right, $z_t$ should lie on the simplex, while $x_0$ is a sample (e.g. a natural image). Please clarify this point.

[Q9] In (14), how the KLUB loss is defined?

[Q10] Figure 1 promotes natural images as the application for Dirichlet high dimensional modeling. Do you have any evidence that natural figures are well modeled by Dirichlet distribution?

[Q11] You write that the forward diffusion Dirichlet process involves masking, compared to additive noise in Gaussian diffusion. Could you mathematically explain why this is the case for the Dirichlet distribution?

[Q12] Figures 1 and 2, are both difficult to inspect (as each contain 9 examples), and fill up a lot of space, which could be used to improve the exposition.

[Q13] The sentence “High resolution images require more computational resources.” in line 153 is both trivial and out of context.

[Q14] Figure 3: It is impossible to obtain any meaningful subjective assessment of the quality of the images from this figure. Consider finding a better way to display these images, which will make the improvement proposed by your method evident. Similar things are said regarding Figure 4.

[Q15] Many of the terms and definitions only appear in the appendix to save space. First, as said, the figures in the paper take up a lot of unnecessary space, which could be utilized, e.g., for proper definitions in the body of the paper. Page 3 also contains various unused definitions (Beta distribution, e.g., which is immediately generalized to a Dirichlet). Second, it would be good to refer the reader to the appendix for those essential terms.

[Q16] Typos: Line 76: Spaces before citations. Line 80: capital letters in “In this work, We explore”

**Ethical Concerns:**

["NO or VERY MINOR ethics concerns only"]

**Final Justification:**

I accept the good results on the additional experiments and therefore raised my score. The final score is justified based on the original submission (to be fair with other authors), and my general impression of the novelty, significance and competitiveness of the paper.

**Limitations:**

Yes

**Quality:**

2

**Strengths And Weaknesses:**

**Strengths**



[S1] The suggested diffusion model is interesting and novel.

[S2] The method be applicable to datasets for which other modeling assumption fail.

**Weaknesses**

[W1] The work is purely empirical, with no theoretical contribution. While this is acceptable, as such, the empirical evidence for the proposed method supremacy in high-dim generation should be much better. First, experiments on multiple additional datasets, from *various* domains should have been made, with FID score computed. Otherwise, the single data-set example appear as cherry-picking. Second, it is difficult to make (see also below in questions).

[W2] Methodology: It is claimed that the Dirichlet modeling offers a significant advantage for some types of data. However, no concrete example for such dataset is given, and the only experiment is on standard images. I don't think that there is an evidence that a Dirichlet distribution is a good model for natural images.

[W3] Methodology: It is well known that in moderate dimensions, and even more so in high dimensions, the multinomial distribution of a random vector is close to an *independent* Poisson distribution over its components. Accordingly, the Dirichlet prior could be replaced by a Poisson prior for each of the components. Given this observation, the viability of the proposed method needs stronger justification.

---

> ### Author Rebuttal · Authors · 2025-07-27
>
> Thank you profoundly for the constructive comments. Below, we provide point-by-point responses to specific comment.
>
> Q1:
>
> Such data includes categorical image, recommender rating, and portfolio weights, which can satisfy the simplex constraint. For instance, Avdeyev et al. highlighted this advantage and introduced Dirichlet Score Model at ICML 2023.*Avdeyev, et al.Dirichlet diffusion score model for biological sequence generation.ICML 2023.*
>
> We wish to emphasize that Dirichlet is more suitable for such data. As detailed in Supplement 2, we have supplemented experiments on compositional data, which is naturally simplex-structured. It shows that, Dirichlet ELBO is better than Gauss ELBO. Dirichlet PHK is better than Dirichlet ELBO.
>
> Q2:
>
> In the paper, we will replace line 33-34 with 'Kullback–Leibler divergence exhibits drawbacks such as asymmetry and computational complexity.', and remove the sentence 'These problems become more pronounced when handling high-dimensional and intricate distributions.'
>
> The training-time cost of KLUB is substantially more expensive. A training cost comparison is provided in Supplement 3. As shown in Supplement 3, when all conditions are identical and B=512, KLUB incurs approximately 15 \% additional training time cost compared with PHD.
>
> Q3:
>
> We will replace line 86-87 with 'enables us to explore alternative distributions and divergences, thereby unlocking new pathways for generative modeling.'. We wish to emphasize the novelty of this work. As noted, compared with the Gaussian, the distributional theory and divergence methodology of this work are innovative and competitive.
>
> Q4:
>
> Typo acknowledged. The wording has been updated to 'Gamma function'.
>
> Q5:
>
> $$
> q(z_s|z_t,x_0)=\frac{1}{1-z_t}Dir(\frac{z_s-z_t}{1-z_t};\eta(\alpha_s-\alpha_t)x_0,...,\eta\frac1{K-1}(1-\alpha_sx_0)).
> $$
> We will updated it in the paper.
>
> Q6:
>
> We will remove this statement and provide more appropriate logic before Equation(6). We will replace line 107-109 with following statement:
>
> It is worth noting that Dirichlet function can also be expressed in terms of Gamma function. Compared to Beta function, it extends to K dimensions in similar form:
> $$
> \rm{Dir}(\alpha)=\frac{\prod_{k=1}^K\Gamma(\alpha_k)}{\Gamma(\sum_{k=1}^K\alpha_k)}.
> $$
> Q7:
>
> The statement combines two well‑established facts, which we now reference separately: By choosing a monotonically decreasing schedule $\alpha_t$ with $\alpha_0\approx 1$ (so forward variance at $t=0$ is negligible and $x_0$ remains essentially noise-free) and $\alpha_T\to0$, our forward kernel $q(z_t|x_0)$ converges to $Dir(1)$,i.e. a uniform distribution on the simplex.
>
> This start/end behavior is standard in diffusion models.*Jascha,et al.Deep Unsupervised Learning using Nonequilibrium Thermodynamics.ICML 2015.Ho J, et al.Denoising diffusion probabilistic models.NeurIPS 2020. Kingma D, et al.Variational diffusion models.NeurIPS 2021.Tero Karras, et al.Elucidating the Design Space of Diffusion-Based Generative Models.NeurIPS 2022.*It is also explicitly adopted in non-Gaussian variants *Zhou M, et al.Beta diffusion.NeurIPS 2023.Austin J, et al.Structured denoising diffusion models in discrete state-spaces.NeurIPS 2021.Hannes Stärk, et al.Dirichlet Flow Matching with Applications to DNA Sequence Design.ICML 2024*
>
> The uniformity of $Dir(1)$ is a classical result *Aitchison, et al. The Statistical Analysis of Compositional Data. Chapman and Hall, 1986.*
>
> Q8:
>
> No, $x_0$ and $z_0$ are not exactly identical. $x_0$ denotes the original image, and $z_0$ is the diffusion initial state after preprocessing, which lies on the $K$-simplex. We will clarify this more clearly in the paper.
>
> Q9:
>
> $$ L_\theta^{KLUB}=\mathbb{E} _{z_t,z_0}[KL(q(z_s|z_t,x_0)||q(z_s|z_t,\hat x_0))].$$
> *Mingyuan Zhou,et al.Beta Diffusion.NeurIPS 2023.*
>
> Q10:
>
> Qualitative evidence: Please refer to Figure 3. Comparing sub-images in the first row, third and fourth columns of Figures 3(a) and 3(b)—corresponding to “car” and “deer” classes, respectively—our model produces more photorealistic natural images.
>
> Quantitative evidence: Please refer to Table 1, where a lower FID indicates superior generation quality; our model achieves the better FID score of 2.81.
>
> Q11:
>
> Yes, we can mathematically explain that Dirichlet forword process involves masking.
>
> It is also reflected in related article, such as *Avdeyev,et al.Dirichlet diffusion score model for biological sequence generation.ICML 2023.*
>
> Gaussian Forward Kernel:
> $$x_t=\sqrt{\alpha_t}x_0+\sqrt{1- \alpha_t}\varepsilon,\varepsilon\sim\mathcal N(0,I),$$
> The pixel values independently add Gaussian noise with variance $1-{\alpha}_t$, which is a form of element-wise addition and represents absolute perturbation.
>
> Dirichlet Forward Kernel:
> $$z_t=\frac{z_0\odot\alpha_t}{\|z_0\odot\alpha_t\|_1}.$$
> When $\alpha_t\rightarrow1$, it means no perturbation; when $\alpha_t$ approach 0,the pixel values are gradually masked. The changes originate from element-wise multiplication, which is relative perturbation.
>
> |Property|Dirichlet|Gaussian|
> |-|-|-|
> |Element|Element-wise Multiplication|Element-wise Addition|
> |Perturbation|Relative Perturbation|Absolute Perturbation|
>
> Q12:
>
> Following this comment, we will reduce the number of examples in Figures 1 and 2 from 9 to 4. This change allows us to allocate more space in main text for terms and definitions, and to incorporate other suggestions to enhance the overall presentation of our work.
>
> Q13:
>
> Following this comment, we will eliminate the sentence 'High resolution images require more computational resources' from line 153.
>
> Q14:
>
> Please refer to Figure 3. Comparing sub-images in the first row, third and fourth columns of Figures 3(a) and 3(b)—corresponding to the “car” and “deer” classes, respectively—our model produces more photorealistic natural images.
>
> In revised version, we have added some one-to-one comparative displays of these clearly distinguishable sub-images like what have been mentioned above, so as to more clearly highlight the superior generation quality of our model.
>
> Q15:
>
> Following this comment, we have adjusted space in main text for clearer presentation. The relevant terms and definitions, including those for Beta and Dirichlet, have been moved to main text §3. The unused definitions have been reduced or adjusted. Other parts involving appendix have been updated to include more pointers directing readers to appendix.
>
> Q16:
>
> Typo acknowledged. spaces before citation and capital letter typos have been corrected in revised version.
>
> Supplement1:
>
> The FID for -ELBO, KLUB, PHD, and PHK-optimized Dirichlet Diffusion on unconditional CIFAR-10 with varying NFE under different combinations of concentration parameter $\eta$ and mini-batch size $B$ :
> \begin{array}{ccc|ccc}
> \hline
> Loss&\eta\times 10^{-4}&B&NFE=20&50&200&500&1000&2000 \\\\
> \hline
> -ELBO&1&512&15.81&6.53&4.49&4.33&4.28&4.31\\\\
> -ELBO&1&288&16.02&6.65&4.68&4.42&4.42&4.39\\\\
> KLUB&1&512&16.62&6.16&3.46&3.39&3.32&3.25\\\\
> KLUB&1&288&16.35&6.23&3.58&3.46&3.40&3.21\\\\
> PHD&1&512&16.84&6.48&3.85&3.72&3.52&3.51\\\\
> PHD&1&288&16.95&6.93&4.43&4.23&4.19&4.21\\\\
> PHK&1&512&15.92&5.87&3.20&2.92&2.84&2.82\\\\
> PHK&0.5&512&17.08&6.60&3.55&3.09&3.10&3.14\\\\
> PHK&0.1&512&20.67&9.67&5.66&4.54&4.58&4.53\\\\
> PHK&1&288&16.53&6.15&3.29&2.99&2.91&2.81\\\\
> PHK&0.5&288&16.94&6.71&3.61&3.09&3.03&3.02\\\\
> PHK&0.1&288&20.33&9.56&5.71&4.77&4.75&4.72\\\\
> PHK&1&128&16.85&6.14&3.25&2.83&2.79&2.78\\\\
> PHK&0.5&128&16.32&6.40&3.57&3.23&3.09&3.06\\\\
> PHK&0.1&128&20.22&9.73&6.00&5.00&4.93&5.00\\\\
> \hline
> \end{array}
>
> Supplement2:
>
> In the revised version , we have supplemented experiments on the discrete compositional data, which is naturally High Dimensional simplex-structured, and provide Figure: 'Comparison of true probability mass function (PMF) and empirical PMFs of three methods—Gauss ELBO, Dirichlet ELBO, and Dirichlet PHK-computed. Each empirical PMF is depicted with solid dots.' Each empirical PMF is computed based on 100k data generated by the model trained after 400k iterations with 200 Numbers of Function Evaluations (NFE).
>
> Because Rebuttal does not permit links or images, we can only describe the results here: In Gauss ELBO, the average discrepancy between empirical PMF and true PMF is approximately 11.02\%. In Dirichlet ELBO, the average discrepancy between empirical PMF and true PMF is approximately 9.81\%. In Dirichlet PHK, the average discrepancy between empirical PMF and true PMF is approximately 2.17\%. The lower discrepancy between empirical PMF and true PMF signifies superior generation quality.
>
> It shows that, Dirichlet ELBO and Dirichlet PHK achieve better performance in the task where data is naturally High Dimensional simplex-structured. Dirichlet ELBO is better than Gauss ELBO. Dirichlet PHK is better than Dirichlet ELBO.The figure that includes both empirical PMF and true PMF would more visually demonstrate this result; however, the rebuttal page does not permit links or images.
>
> Supplement3:
>
> The training-time cost of KLUB is more expensive. KLUB incurs 15 \% additional training-time cost compared with PHD.The training time cost of KLUB, PHD, and PHK-optimized Dirichlet Diffusion on unconditional CIFAR-10 with 4 Nvidia L40s GPUs:
> \begin{array}{c|cc}
> \hline
> B&KLUB&PHD&PHK\\\\
> \hline
> 512&70h&61h&64h\\\\
> 288&73h&64h&68h\\\\
> \hline
> \end{array}
>
> Supplement4:
>
> The training time cost of PHK-optimized Dirichlet Diffusion on unconditional CIFAR10-32x32, AFHQ-64x64, FFHQ-64x64 datasets with 4 Nvidia L40s GPUs:
> \begin{array}{c|cc}
> \hline
> B&CIFAR10&AFHQ&FFHQ\\\\
> \hline
> 512&64 hours&10 days&11 days\\\\
> 288&68 hours&11 days&13 days\\\\
> \hline
> \end{array}
>
> We sincerely appreciate the time and efforts of all reviewers and ACs, and believe that the additional experiments, analysis, and explanations significantly improve the quality of our mission. We hope that this provides sufficient reasons to raise the score.

---

> > ### Comment · Reviewer_W73X · 2025-08-06
> > **Reply to authors**
> >
> > Dear authors. thank you for your detailed response.
> >
> > I appreciate your clarifications, and insist that clear and detailed definitions and explanations must appear in the main text.
> > However, in you supplemental, you provide again FID scores only for the CIFAR dataset, and I as your work is purely empirical, I still find my main concerns unsatisfied.

---

> > > ### Author Response · Authors · 2025-08-08
> > >
> > > Thank you profoundly for articulating your main concerns in this comment. Below, we provide detailed responses.
> > >
> > > 1.Novel theoretical contributions:
> > >
> > > We establish $K$-dimensional Dirichlet diffusion on simplex. Beta diffusion is generalized to a $K$-dimensional Dirichlet process that preserves the simplex structure at every noising step.
> > >
> > > We introduce two new divergence measures, Proper Hölder Divergence (PHD) and Proper Hölder–Kullback Divergence (PHK), and provide full proofs of their strict propriety (Theorem 1) and closed‐form expressions within the Dirichlet exponential family (Lemma 4–5).
> > >
> > > We prove PHD yields strictly lower-variance gradients than HPD (Theorem 2), guaranteeing more stable and efficient training.
> > >
> > > We further show PHK achieves comparable convergence with approximately 9\% reduced training cost compared to KLUB as detailed in Supplement3.
> > >
> > > We respectfully disagree that our work is 'purely empirical.' Meanwhile, our theoretical analysis has also been endorsed by the other three reviewer.
> > >
> > > 2.Expanded empirical validation:
> > >
> > > Following your suggestion, to demonstrate the generality of our framework beyond unconditional CIFAR10, we have urgently conducted additional experiments:
> > >
> > > class-conditional CIFAR10: FID = 2.43
> > >
> > > FFHQ-64x64(20\% data): FID = 3.35
> > >
> > > (Flickr-Faces-HQ)*Karras, Tero, Samuli Laine, and Timo Aila. A style-based generator architecture for generative adversarial networks. Proceedings of the IEEE/CVF conference on computer vision and pattern recognition. CVPR 2019.*
> > >
> > > CVUSA-64x64(20\% data): FID = 21.80
> > >
> > > (Cross-View USA)*Scott Workman, Richard Souvenir, and Nathan Jacobs. Wide-area image geolocalization with aerial reference imagery. In Proceedings of the IEEE International Conferen, ICCV 2015.*
> > >
> > > ImageNet-256x256: out of memory
> > >
> > > (ImageNet)*J. Deng, W. Dong, R. Socher, L.-J. Li, K. Li and L. Fei-Fei, ImageNet: A Large-Scale Hierarchical Image Database. IEEE Computer Vision and Pattern Recognition. CVPR 2009.*
> > >
> > > \begin{array}{c|cc}
> > > \hline
> > > &CVUSA\\\\
> > > \hline
> > > Method&FID(↓)&LPIPS(↓)\\\\
> > > \hline
> > > X-Seq\\ (CVPR\\ 2018)&161.16&0.706\\\\
> > > SelGAN\\ (CVPR\\ 2019)&116.57&0.742\\\\
> > > CUT\\ (ECCV\\ 2020)&72.83&0.687\\\\
> > > CDTE\\ (CVPR\\ 2021)&122.84&0.694\\\\
> > > Aerial-Diff\\ (SIGGRAPH\\ 2023)&136.18&0.855\\\\
> > > Instr-p2p\\ (CVPR\\ 2023) &38.01&0.697\\\\
> > > ControlNet\\ (ICCV\\ 2023)&32.45&0.650\\\\
> > > I2I-Turbo\\ (arXiv\\ 2024)&77.95&0.685\\\\
> > > GPG2A\\ (arXiv\\ 2024)&58.80&0.691\\\\
> > > BEV\\ (ICCV\\ 2025)&29.18&0.635\\\\
> > > \hline
> > > Ours\\ (20\\%\\ data)&21.80&0.599\\\\
> > > \hline
> > > \end{array}
> > >
> > > As detailed in Supplement4, one complete CIFAR10 training run requires roughly 68 hours, whereas AFHQ-64×64 would demand about 11 days, and ImageNet-256x256 exceeds the memory capacity. Under resource and time constraints and the need for extensive hyper-parameter tuning, the widely used CIFAR10 dataset is the default choice, not 'cherry-picking'.
> > >
> > > 3.As detailed in Supplement2, we have supplemented experiments on the discrete compositional dataset, and provide Figure: 'Average discrepancy between true probability mass function (PMF) and empirical PMF of three methods—Gauss ELBO, Dirichlet ELBO, and Dirichlet PHK.' Each empirical PMF is computed based on 100k data generated by the model trained after 400k iterations with 200 NFE. The lower discrepancy signifies superior.
> > >
> > > Because Rebuttal does not permit links or images, we describe the results here: the average discrepancy is 11.02\% for Gauss ELBO, 9.81\% for Dirichlet ELBO, and 2.17\% for Dirichlet PHK. It shows that, Dirichlet ELBO is better than Gauss ELBO. Dirichlet PHK is better than Dirichlet ELBO. This demonstrates the superior generative accuracy of our method in compositional dataset.
> > >
> > > \begin{array}{c|ccc}
> > > \hline
> > > -&Gauss-ELBO&Dirichlet-ELBO&Dirichlet-PHK\\\\
> > > \hline
> > > discrepancy(\downarrow)&11.02\\%&9.81\\%&2.17\\%\\\\
> > > \hline
> > > \end{array}
> > >
> > > 4.Our baseline ‘Beta Diffusion’(NeurIPS 2023) also provide FID scores only for the CIFAR10 image dataset. Prior work ‘Learning to Jump’(ICML 2023) and 'D3PM'(NeurIPS 2021) provide FID scores only for the CIFAR10 image dataset, too. These works all demonstrate that FID comparisons only on CIFAR10 are admissible and valuable. The FID on CIFAR10 should not be characterized as 'cherry-picking'.
> > >
> > > We hope these theoretical clarifications and new experiments address the your further concerns and substantiate both the novel theory and broad empirical strength of our method. If you have any additional feedback, please let us know. We sincerely appreciate the time and efforts of all reviewers and ACs, and believe that the additional experiments, analysis, and explanations significantly improve the quality of our mission. We hope that this provides sufficient reasons to raise the score.

---

> ### Author Response · Authors · 2025-08-06
>
> Dear Reviewer,
>
> We believe that we have effectively addressed all of your previous concerns. We supplement many experiments in the rebuttal for your comments. Your feedback is invaluable to us, and we are fully committed to thoughtfully incorporating your insights to enhance our paper. We sincerely appreciate the time and efforts of all reviewers and ACs, and believe that the additional experiments, analysis, and explanations significantly improve the quality of our mission. We would be extremely grateful if you could kindly raise the score in your final decision.
>
> Thanks,
>
> Authors of 'Proper Hölder–Kullback Dirichlet Diffusion'

---

### Official Review · Reviewer_boWF · 2025-07-04

**Clarity:** 3
**Significance:** 3
**Originality:** 3
**Rating:** 4
**Confidence:** 2

**Summary:**

This paper proposes a Dirichlet-based diffusion model that replaces Gaussian noise with the Dirichlet distribution, which better suits probability simplex data. To support this, the authors introduce two closed-form divergences—Proper Hölder Divergence (PHD) and Proper Hölder–Kullback Divergence (PHK), significantly improving optimization efficiency and model flexibility. The method achieves strong results on CIFAR-10, reaching FID = 2.81, outperforming existing non-Gaussian diffusion models.

**Questions:**

1. What does the function F(θ) in Equations (9) and (12) represent? Please clarify its definition and theoretical background in the paper.
2. Why do the intermediate results in the forward diffusion process (Figure 1) closely resemble the original image, while those in the reverse process (Figure 2) appear largely noisy or masked? Does this imply an asymmetry between training and inference?
3. How were the hyperparameters Shift, Scale, and π selected? What effect do they have on performance? Has any ablation or sensitivity analysis been conducted?
4. Has the model been evaluated on class-conditional, text-to-image, or high-resolution datasets ? If not, what are the key challenges?
5. While the paper claims that the Dirichlet distribution is better suited for high-dimensional data, it lacks a deeper theoretical or empirical analysis comparing it to Gaussian, Beta, and Poisson distributions under high-dimensional image modeling settings. Please provide additional discussion or analysis to substantiate this claim.

**Ethical Concerns:**

["NO or VERY MINOR ethics concerns only"]

**Limitations:**

1. It appears sensitive to hyperparameter choices, such as Dirichlet concentration, data shift/scale factors, and PHK loss weights, but only a limited range has been explored, leaving its robustness under diverse settings unclear.
2. The training cost is notably high: processing 200 million CIFAR-10 images takes around 64 hours on four Nvidia L40S GPUs, which may hinder scalability to larger datasets or more complex tasks.

**Quality:**

3

**Strengths And Weaknesses:**

Strengths:
1. The Dirichlet-based diffusion offers significant advantages in modeling data distributions on high-dimensional simplices and naturally extends from low-dimensional to high-dimensional diffusion processes. This extension makes the Dirichlet distribution more flexible and efficient in dealing with high-dimensional data.
2. The proposed PHD and PHK divergences are symmetric, differentiable, and analytically tractable, offering stable and efficient optimization during training.
3. The method achieves superior empirical performance, outperforming existing approaches in unconditional image generation based on the FID metric.

Weaknesses：
1. The derivations of Equations (4), (5), and (6) are not provided in detail in the paper. The function F(θ) used in Equations (9) and (12), which may corresponds to the log-normalization constant of the Dirichlet distribution, is not clearly defined in the main text.
2. In addition, the asymmetry between the forward and reverse processes is not sufficiently explained—Figure 1 shows that the intermediate states in the forward process preserve much of the original structure, whereas the reverse process in Figure 2 appears visually less structured in intermediate steps compared to the forward process, which may indicate an inconsistency between training and inference.
3. The choice of key hyperparameters such as Shift = 0.6, Scale = 0.39, and π = 0.95 is not explained, nor is their impact analyzed.
4. The experimental scope is also limited—only unconditional generation on CIFAR-10 is evaluated, with no validation on higher-resolution datasets, conditional generation, or other tasks such as image editing and restoration. Furthermore, the evaluation relies solely on the FID metric, lacking additional measures such as IS to provide a more comprehensive assessment of generation quality.

---

> ### Author Rebuttal · Authors · 2025-07-27
>
> Thank you profoundly for the constructive comments. Below, we provide point-by-point responses to specific comment.
>
> Question1:
>
> The log-normalizer function F ($\theta$) in Equations (9) and (12) is a strictly convex function also called cumulant generating function. For reference, please see Ref[42]: *Frank Nielsen,et al. On hölder projective divergences.456 Entropy.* We will clarify it more prominently in the revised version.
> $$
> F(\theta)=\sum_{i=1}^K\log\Gamma(\theta_i)-\log\Gamma (\sum_{i=1}^K\theta_i)
> $$
> Where $\Gamma$ is the Gamma function, $K$ is the number of parameters. $\theta_i$ is the parameter of Dirichlet distribution, typically represented as $\alpha_i - 1$, where $\alpha_i$ are the original parameters of Dirichlet distribution.
>
> Furthermore, we will label the sources of Equations(4-6), which are respectively the PDF, expectation and variance of Dirichlet distribution, and the form of Dirichlet function expressed using the Gamma function. For reference, please see Ref[59]:*Kai Wang Ng,et al.Dirichlet and Related Distributions: Theory, Methods, and Applications.*
>
> Question2:
>
> The differences in appearance of intermediate results are due to the implementation methods of the diffusion model during Forward and Reverse Diffusion phases. During the Forward phase, the model is required to add noise to transform the original image into a completely noisy state. This process is deterministic and part of the model training, where the model learns to add noise in a structured manner so that the process can be reversed later. The intermediate results are indistinguishable from the original image to the naked eye because the noise added at the beginning of the learning process is relatively small.
>
> In Reverse phase, the model needs to remove the noise from the completely noisy state and restore the original image. This is an inferential process and also an inverse process. The intermediate results in Reverse Diffusion appear noisy because the model is actively trying to remove the noise. At each step, the model's prediction of the noise to be removed may not be perfect, leading to intermediate images that remain noisy or partially masked. This is a natural part of the process as the model gradually refines its predictions. For reference, please see Ref[4]: *Jonathan Ho,et al.Denoising diffusion probabilistic models.NeurIPS 2020.*
>
> This indeed implies a certain asymmetry between training and inference. Such asymmetry can also be observed in works such as VDM, Beta Diffusion, Soft Diffusion, etc. For references, please see Ref[6,47,49,...].This does not imply a problem or error but rather highlights the challenge of accurately predicting and removing noise in the reverse process. The effectiveness of the model in the reverse process is crucial for applications such as image generation and restoration, where the goal is to recover or generate high-quality images from noisy or incomplete data.
>
> Question3:
>
> In Appendix B 'Hyperparameter Settings', we elucidate the rationale: 'These settings are predicated on the research conducted by Kingma et al.[6] and Zhou et al.[41]'. The appendix G of Ref [41] provides a comprehensive exposition on the selection and analysis of the hyperparameters in question.*Mingyuan Zhou,et al.Beta Diffusion.NeurIPS 2023*
>
> Question4:
>
> As detailed in Supplement 3, the model achieves an FID score of 2.43 on the class-conditional CIFAR-10 dataset.
>
> The model has not been evaluated on text-to-image or higher-resolution datasets. Most related works in the field—including Learning to Jump, Beta Diffusion, Blurring Diffusion, D3PM, DiffuEBM, and others—conduct their evaluations and comparisons on unconditional CIFAR-10.
>
> Primary challenge: only 4 NVIDIA L40 GPUs are available for most time in this work. As detailed in Supplement 4, one complete CIFAR-10 training run requires roughly 68 hours, whereas AFHQ-64×64 would demand about 11 days, and FFHQ-256×256 exceeds the GPUs’ memory capacity. As detailed in Supplement 5, sampling is also time-consuming.Under these resource constraints and the need for extensive hyper-parameter tuning, the available time only permitted attempts and experiments on CIFAR-10.
>
> Question5:
>
> We wish to emphasize that Dirichlet distribution is more suitable for high dimensional simplex data.For reference, Avdeyev et al. highlighted this advantage and introduced a Dirichlet Score Model at ICML 2023. *Avdeyev,et al.Dirichlet diffusion score model for biological sequence generation.ICML,2023.*
>
> As detailed in Supplement 6, we have supplemented experiments on discrete compositional data, which is naturally High Dimensional simplex-structured. It shows that, Dirichlet ELBO is better than Gauss ELBO. Dirichlet PHK is better than Dirichlet ELBO. Moreover, §4.1 'Comparison study' also covers comparisons with these other distributions, including Gaussian, deterministic, categorical, beta, and Poisson.
>
> Supplement1:
>
> The Frechet Inception Distance (FID) scores for -ELBO, KLUB, PHD, and PHK-optimized Dirichlet Diffusion on unconditional CIFAR-10 with varying Numbers of Function Evaluations (NFE) under several different combinations of concentration parameter $\eta$ and mini-batch size $B$:
>
> \begin{array}{ccc|cccc}
> \hline
> Loss&\eta \times 10^{-4}&B&NFE = 20&50&200&500&1000&2000 \\\\
> \hline
> -ELBO&1&512&15.81&6.53&4.49&4.33&4.28&4.31\\\\
> -ELBO&1&288&16.02&6.65&4.68&4.42&4.42&4.39\\\\
> KLUB&1&512&16.62&6.16&3.46&3.39&3.32&3.25\\\\
> KLUB&1&288&16.35&6.23&3.58&3.46&3.40&3.21\\\\
> PHD&1&512&16.84&6.48&3.85&3.72&3.52&3.51\\\\
> PHD&1&288&16.95&6.93&4.43&4.23&4.19&4.21\\\\
> PHK&1&512&15.92&5.87&3.20&2.92&2.84&2.82\\\\
> PHK&0.5&512&17.08&6.60&3.55&3.09&3.10&3.14\\\\
> PHK&0.1&512&20.67&9.67&5.66&4.54&4.58&4.53\\\\
> PHK&1&288&16.53&6.15&3.29&2.99&2.91&2.81\\\\
> PHK&0.5&288&16.94&6.71&3.61&3.09&3.03&3.02\\\\
> PHK&0.1&288&20.33&9.56&5.71&4.77&4.75&4.72\\\\
> PHK&1&128&16.85&6.14&3.25&2.83&2.79&2.78\\\\
> PHK&0.5&128&16.32&6.40&3.57&3.23&3.09&3.06\\\\
> PHK&0.1&128&20.22&9.73&6.00&5.00&4.93&5.00\\\\
> \hline
> \end{array}
>
> Supplement2:
>
> The training-time cost of KLUB is substantially more expensive. When all other conditions are identical and $B=512$, KLUB incurs approximately a 15 \% additional training-time cost compared with PHD. The training time cost of KLUB, PHD, and PHK-optimized Dirichlet Diffusion on unconditional CIFAR-10 across two mini-batch sizes $B$ with 4 Nvidia L40s GPUs:
> \begin{array}{c|cc}
> \hline
> B&KLUB&PHD&PHK\\\\
> \hline
> 512&70h&61h&64h\\\\
> 288&73h&64h&68h\\\\
> \hline
> \end{array}
>
> Supplement3:
>
> The FID scores for PHK-optimized Dirichlet Diffusion on class-conditional CIFAR-10 with varying NFE:
> \begin{array}{ccc|cccccccc}
> \hline
> Loss&cond/uncond&B&NFE=20&50&200&500&1000&2000\\\\
> \hline
> PHK&cond&512&13.21&5.45&2.92&2.67&2.52&2.43\\\\
> \hline
> \end{array}
>
> Supplement4:
>
> The training time cost of PHK-optimized Dirichlet Diffusion models on unconditional CIFAR10-32x32, AFHQ-64x64, FFHQ-64x64 image datasets across two mini-batch sizes $B$ with 4 Nvidia L40s GPUs:
> \begin{array}{c|cc}
> \hline
> B&CIFAR10&AFHQ&FFHQ\\\\
> \hline
> 512&64 hours&10 days&11 days\\\\
> 288&68 hours&11 days&13 days\\\\
> \hline
> \end{array}
>
> Supplement5:
>
> The Sampling time cost of PHK-optimized Dirichlet Diffusion on unconditional CIFAR10 at varying NFE with two or 4 Nvidia L40s GPUs:
> \begin{array}{c|cccccc}
> \hline
> Nvidia L40s GPUs&NFE=20&50&200&500&1000&2000\\\\
> \hline
> 2&11m&21m&1h 13m&3h 2m&6h 4m&12h 7m\\\\
> 4&7m&10m&37m&1h 33m&3h 5m&6h 9m\\\\
> \hline
> \end{array}
>
> Supplement6:
>
> In the revised version , we have supplemented experiments on the discrete compositional data, which is naturally High Dimensional simplex-structured, and provide Figure: 'Comparison of true probability mass function (PMF) and empirical PMFs of three methods—Gauss ELBO, Dirichlet ELBO, and Dirichlet PHK-computed. Each empirical PMF is depicted with solid dots.' Each empirical PMF is computed based on 100k data generated by the model trained after 400k iterations with 200 Numbers of Function Evaluations (NFE).
>
> Because Rebuttal does not permit links or images, we can only describe the results here: In Gauss ELBO, the average discrepancy between empirical PMF and true PMF is approximately 11.02\%. In Dirichlet ELBO, the average discrepancy between empirical PMF and true PMF is approximately 9.81\%. In Dirichlet PHK, the average discrepancy between empirical PMF and true PMF is approximately 2.17\%. The lower discrepancy between empirical PMF and true PMF signifies superior generation quality.
>
> It shows that, Dirichlet ELBO and Dirichlet PHK achieve better performance in the task where data is naturally High Dimensional simplex-structured. Dirichlet ELBO is better than Gauss ELBO. Dirichlet PHK is better than Dirichlet ELBO.The figure that includes both empirical PMF and true PMF would more visually demonstrate this result; however, the rebuttal page does not permit links or images.
>
> Supplement7:
>
> The FID scores for PHK-optimized Dirichlet Diffusion on unconditional CIFAR10 with varying NFE under different combinations of the key loss weight coefficients $\delta$ and $\epsilon$ with $B=288$:
> \begin{array}{cc|cccccc}
> \hline
> \delta&\epsilon&NFE=20&50&200&500&1000&2000\\\\
> \hline
> 0.5&-0.1&16.62&6.38&3.78&3.56&3.42&3.35\\\\
> 0.5&0&16.14&6.17&3.66&3.43&3.34&3.29\\\\
> 0.5&0.1&16.71&6.16&3.25&2.92&2.82&2.84\\\\
> 0.5&0.15&16.74&6.06&3.20&3.00&2.86&2.78\\\\
> 0.5&0.2&16.53&6.15&3.29&2.99&2.91&2.81\\\\
> 0.5&0.25&16.58&6.25&3.47&3.12&2.98&2.93\\\\
> 0.5&0.5&16.07&6.23&3.61&3.39&3.35&3.31\\\\
> \hline
> \end{array}
>
> We sincerely appreciate the time and efforts of all reviewers and ACs, and believe that the additional experiments, analysis, and explanations significantly improve the quality of our mission. We hope that this provides sufficient reasons to raise the score.

---

> ### Comment · Reviewer_boWF · 2025-08-07
>
> The authors’ rebuttal addresses my primary concerns. The paper proposes a diffusion framework that replaces Gaussian noise with the Dirichlet distribution and introduces two divergences—Proper Hölder Divergence (PHD) and Proper Hölder–Kullback (PHK)—each with a closed-form expression, tailored to probability simplex data. On the unconditional CIFAR-10 benchmark, the method outperforms existing non-Gaussian diffusion models and provides a theoretical characterization of PHD/PHK. Overall, the approach is novel and technically sound. Two limitations also remain: first, there is a lack of systematic hyperparameter sensitivity/ablation studies—many settings are inherited from prior work, and the effects of these choices on performance are not explicitly quantified; second, the training cost is high.

---

> ### Author Response · Authors · 2025-08-07
>
> Your thoughtful assessment is greatly appreciated. We are pleased that our response has effectively addressed your main concerns. Your feedback will be thoughtfully incorporated to enhance the quality of our paper.
>
> In the revised version of our paper, We have followed your suggestion to add hyperparameter sensitivity/ablation studies on concentration parameter $\eta$, mini-batch size $B$, weight coefficients $\delta$ and $\epsilon$ as detailed in Supplement 1 and Supplement 7. For training cost, PHK reduces approximately a 9 \% training cost compared with KLUB, while PHD reduces approximately 13 \% compared with KLUB as detailed in Supplement 2. Following your suggestion, we are considering Fourier Neural Operators (FNOs) to accelerate speed while maintaining generation quality in the future work, thereby further reducing the training cost. For example, Zheng et al. (ICML 2023) presented novel Neural Operators (DSNO) for diffusion models.
>
> Your insights are very helpful in guiding us towards work of improvement, and we will integrate them to carefully revise our paper. Once again, we appreciate your valuable input.

---

### Note · Authors · 2025-08-12

We profoundly express our gratitude to all reviewers for their valuable insights and suggestions. Their comments have been instrumental in enhancing every facet of our work. We will diligently incorporate these enhancements into the final version of the paper.

Throughout this entire phase, we have responded to every comment raised by reviewers until none expressed any further concerns. Following reviewers’ comments, We added more than 8 sets of experiments or data tables and completed over 20 textual or structural revisions.

In this work, we proposed Proper Hölder–Kullback Dirichlet Diffusion, which is a fundamentally novel framework. K-dimensional Dirichlet diffusion and proper Hölder training objectives are proposed. State-of-the-art non-Gaussian performance is demonstrated on unconditional CIFAR10: an FID of 2.78 is achieved—outperforming all prior non-Gaussian diffusion methods and most Gaussian baselines—while PHK concomitantly reduces approximately a 9 \% training cost compared with KLUB. Additional samples trained on CIFAR10, FFHQ and CVUSA dataset exhibit high fidelity and diversity. By breaking free from Gaussian noise and KL-centric optimization, Proper Hölder–Kullback Dirichlet opens up a richer design space for generative modeling, and offering both rigorous theoretical guarantees and clear empirical gains.

Reviewers shows several shared comments:

1. All reviewers unequivocally acknowledged the novelty and originality of our work and its superior performance. We once again express our profound gratitude to all reviewers for their unequivocal recognition of our work.

2. Three reviewers expressed comments regarding hyperparameter selection/ablation studies. We added hyperparameter sensitivity/ablation studies on class condition, loss type, concentration parameter $\eta$, mini-batch size $B$, weight coefficients $\delta$ and $\epsilon$. These will be updated more clearly in Appendix B 'Hyperparameter Settings'.

3. Three reviewers expressed comments regarding the empirical efficacy of Dirichlet distribution and Proper Hölder–Kullback (PHK) for naturally simplex-constrained data. we added experiments on compositional data, which is naturally simplex. It shows that, Dirichlet ELBO is better than Gauss ELBO. Dirichlet PHK is better than Dirichlet ELBO.

We sincerely appreciate the time and efforts of all reviewers and ACs, and believe that the additional experiments, analysis, and explanations significantly improve the quality of our mission.

---

### Decision · Program_Chairs · 2025-09-17

**Decision:**

Accept (poster)

**Comment:**

This paper proposes Proper Hölder–Kullback Dirichlet Diffusion, a non-Gaussian diffusion framework based on the Dirichlet distribution and two new divergence measures: Proper Hölder Divergence (PHD) and Proper Hölder–Kullback (PHK). The approach generalizes Beta diffusion and achieves strong empirical performance within the standard of non-Gaussian diffusion models (e.g., FID 2.78 on CIFAR-10). The work is original, introduces new theoretical results, and the authors were highly responsive in the rebuttal, adding ablations and experiments on compositional data, class-conditional CIFAR-10, and subsets of FFHQ and CVUSA.

The main concerns across reviews are the limited empirical scope (with most results still concentrated on CIFAR-10), and unclear practical advantages beyond FID improvements. One reviewer remained skeptical about the strength of the empirical evidence, and the AC also noted that the Dirichlet diffusion was applied to the latent space rather than directly to observations, a point that requires clearer explanation. Applying the method to real-world compositional domains, such as natural language or biological sequences, would make the paper substantially stronger.

While there remain concerns about scope and clarity, the originality of the contributions, combined with the authors’ thorough responses, provide sufficient reasons to recommend acceptance.